# Spatiotemporal mapping of rice acreage and productivity growth in Bangladesh

**Md. Abdullah Al Mamun**[1]\*, **Sheikh Arafat Islam Nihad**[2], **Mou Rani Sarker**[3], **Md Abdur Rouf Sarkar**[4,5]\*, **Md. Ismail Hossain**[1], **Md. Shahjahan Kabir**[6]

1 Agricultural Statistics Division, Bangladesh Rice Research Institute, Gazipur, Bangladesh, 2 Plant Pathology Division, Bangladesh Rice Research Institute, Gazipur, Bangladesh, 3 Sustainable Impact Platform, International Rice Research Institute, Dhaka, Bangladesh, 4 School of Economics, Zhongnan University of Economics and Law, Wuhan, China, 5 Agricultural Economics Division, Bangladesh Rice Research Institute, Gazipur, Bangladesh, 6 Director General, Bangladesh Rice Research Institute, Gazipur, Bangladesh

\* mdrouf_bau@yahoo.com (MARS); mamunru4777@gmail.com (MAAM)

**Data Availability Statement:** All relevant data are within the manuscript and its Supporting Information files.

## Abstract

Technological advancements have long played crucial roles in rice productivity and food security in Bangladesh. Seasonal variation over time and regional differences in rice production, however, pose a threat to agricultural sustainability but remain unexplored. We performed a spatial-temporal mapping of rice cultivation area, production, and yield from 2006–2007 to 2019–2020 using secondary data for disaggregating 64 districts in Bangladesh. Growth and multivariate approaches were employed to analyze time-series data. Results showed that Mymensingh had the highest rice cultivated area and production, while Bandarban had the lowest. The 14 years highest average rice yield was found in Gopalganj and Dhaka (3.63 tons/ha), while Patuakhali (1.73 tons/ha) had the lowest. For the Aus, Aman, and Boro, the rice cultivation area in 19 districts, 11 districts, and 13 districts declined significantly. The overall rice production increased significantly in most districts. For the Aus, Aman, and Boro seasons, the rice yield in 54, 50, and 37 districts demonstrated a significant upward trend, respectively. The adoption rate of modern varieties has risen dramatically. However, there are notable variances between regions and seasons. A significant increasing trend in Aus (0.007% to 0.521%), Aman (0.004% to 0.039%), and Boro (0.013% to 0.584%) were observed in 28, 34, and 36 districts, respectively, with an increase of 1% adaptation of HYV. Predictions revealed that rice cultivation area and production of Aus, Aman, and Boro seasons will be increased in most of the regions of Bangladesh by 2030. Based on spatiotemporal cluster analysis, the five identified cluster groupings illustrated that clusters lack spatial cohesion and vary greatly seasonally. This suggests increasing rice production by expanding cultivable land, adopting high-yielding varieties, and integrating faster technological advancement in research and extension. The findings will assist scientists in developing region-specific production technologies and policymakers in designing decentral region-specific policies to ensure the future sustainability of rice production.

**Funding:** The author(s) received no specific funding for this work.

**Competing interests:** The authors have declared that no competing interests exist.

## 1. Introduction

Rice (*Oryza sativa* L.) holds a preeminent position as a staple commodity for humanity, representing cultural heritage and providing millions of people with a means of subsistence. With over 3.5 billion people are solely rely on rice for more than 20% of their daily calories, and it supplies approximately 62% carbohydrate, 46% protein, 8% fat, 7% calcium, and 44% phosphorus of the recommended dietary allowance [1–3]. Rice's ability to survive in the hill to submergence areas, drought to cold and other stresses makes it a miracle plant of the world. Nearly 787 million metric tons (MT) of milled rice were produced globally in 2021 [4]. China is the leading rice-producing country, followed by India, Bangladesh, and Vietnam [5]. The average paddy (un-milled rice) yield in Bangladesh is 4.81 tons/ha [6], compared to 6.93 tons/ha in China, 3.69 tons/ha in India, 5.41 tons/ha in Indonesia, and 5.58 tons/ha in Vietnam [7].

Rice is at the food security center in Bangladesh [5, 8]. In the country, rice is grown on more than 11.7 million hectares (ha) in three rice growing seasons (namely Aus, Aman, and Boro), accounting for 77% of the total cropped area [6]. Rice is a staple grain for 169 million people, accounting for about 80% of the total cereal food supply [9]. The daily rice consumption per capita is 367 grams [10]. In 2008, the country attained rice self-sufficiency, significantly contributing to national food demand [8]. Nevertheless, according to the latest projection from the Bangladesh Bureau of Statistics, the population is expected to reach 189.9 million by 2030 [11]. Consequently, there will be a pressing need for a substantial rise in unmilled rice production, estimated at approximately 42.5 million metric tons, in order to meet the growing human demand [12]. Additionally, a recent forecast suggests that the total unmilled rice demand in 2030, accounting for both human and non-human utilization, will be 53.8 million metric tons [8]. Therefore, it is imperative that we significantly enhance rice production to ensure the current level of food security is sustained.

Bangladesh has a rich history of the evolution of rice production. The rice sector has witnessed rapid dynamism in its production processes and wide adaptability in diverse ecosystems under varied soil and climate conditions. Put simply, rice cultivation is divided into four ecosystems, i.e., irrigated, rainfed lowland, upland, and deep water [13]. Agriculture is predominantly irrigated in the northern region, whereas the eastern region has rainfed lowland and deepwater ecosystems, and the southwestern region has irrigated and upland ecosystems [13, 14]. Scholars observed significant regional differences in rice production for soil, irrigation, pests and disease infestation, farming practices, infrastructural availability, varietal traits, varietal security, socio-economic characteristics of the farmer, and so on [5, 9, 15–18].

Over time, compared to growers of other food crops, rice farmers in Bangladesh confront various biotic and abiotic challenges that harm the plant's potential for growth and production. For example, more than 30% of the 700 million low-income families in Asia, including Bangladesh, are in rain-fed lowlands that are impacted by environmental triggers [19]. Changes in meteorological variables such as rainfall, temperature, and humidity, as well as natural disasters such as cyclones, floods, drought, etc., cause differences in rice production across the country [20, 21]. Therefore, there is an urgent need to depict the regional rice cultivation scenario to ensure the future sustainability of the rice sector.

We found very few studies that have examined the spatial and temporal changes in rice area, production, and yield dynamics. These studies were primarily focused on major rice-growing countries such as China, India, Indonesia, Vietnam, Thailand, Myanmar, Japan, Philippines, Pakistan, and Brazil, with limited research specifically dedicated to Bangladesh [22–26]. To the best of our knowledge, only one study has investigated the growth and trend analysis of rice in Bangladesh, but its primary focus was on the regional context, examining 14 agricultural regions [5]. However, it is crucial to acknowledge that the rice-growing ecosystem in

Bangladesh is characterized by a high level of diversity driven by various factors, including geographical position, socio-economic conditions, and environmental variations. As a result, the findings of the previous study have limitations in terms of their statistical robustness and generalizability for formulating region-specific evidence and policies. To overcome these limitations, our study contributes to the empirical literature by adopting a more disaggregated approach, analyzing data from 64 regions (currently known as districts). This approach allows for a comprehensive understanding of the dynamics of rice cultivation in Bangladesh. We believe that understanding the variations in rice production among different regions will serve as a valuable reference for determining micro-level growth scenarios in the rice sector and ensuring long-term food security.

In light of this, we analyzed spatiotemporal data on cultivation area, production and yield of rice to examine the trends and growth patterns from 2006–2007 to 2019–2020 in Bangladesh. Our analysis emphasized rice productivity at the disaggregate level (district) and revealed a distinct productivity gap between highly developed and less developed regions. The main objectives of the present study were as follows: (a) to assess the observed performances and future predictions of rice production systems at seasonal and disaggregated levels; (b) to estimate the district and season-wise temporal changes of rice growth and trend of cultivation area, production, and yield; and (c) to characterize the similarities and dissimilarities spatial group of clustering with the distribution of rice production determinants of Bangladesh. The outcomes of this study will assist researchers and policymakers in developing new technology and designing regional policies that will support the food security-environment-sustainability web of Bangladesh.

## 2. Materials and methods

### 2.1. Study area and data

Bangladesh is located in the eastern part of the South Asian sub-continent. The country's geographic coordinates fall between 20˚34' to 26˚38' north latitude and 88˚01' to 92˚41' east longitude, with an average altitude of 30 feet above sea level. In this study, sixty-four districts of Bangladesh were considered to explore the district-wise rice data (**Fig 1**). Secondary data were used to carry out the objective of spatial clustering and temporal trend analysis of cultivation area (ha), production (metric ton), and yield (ton/ha) of rice in Bangladesh (**S1 Data**). Data collection was performed using multiple issues of the Yearbook of Agricultural Statistics. Rice cultivation area, production, and yield data of 14 years covering three rice growing seasons, i.e., Aus, Aman, and Boro for the periods 2006–2007 to 2019–2020 have been used in this study. We were unable to integrate data because of the statistical yearbook's limitation on 64 districts level information; before 2006, there were only 23 regions with data available [11].

### 2.2. Data validation and growth analysis

To test the stationary of time series data, autocorrelation and partial autocorrelation [27, 28] were calculated using R programming. We also used the tau-statistic for testing stationary under the null hypotheses of Augmented Dickey-Fuller (ADF) to determine whether the cultivation area, production, and yield variables were stationary [29]. Shapiro-Wilk normality test was used to identify the distribution of the datasets [30]. The following exponential growth model [31] was performed to estimate the growth rate of cultivation area, production, and yield of Aus, Aman, and Boro rice in Bangladesh.

$$Y_t = ae^{bt}$$

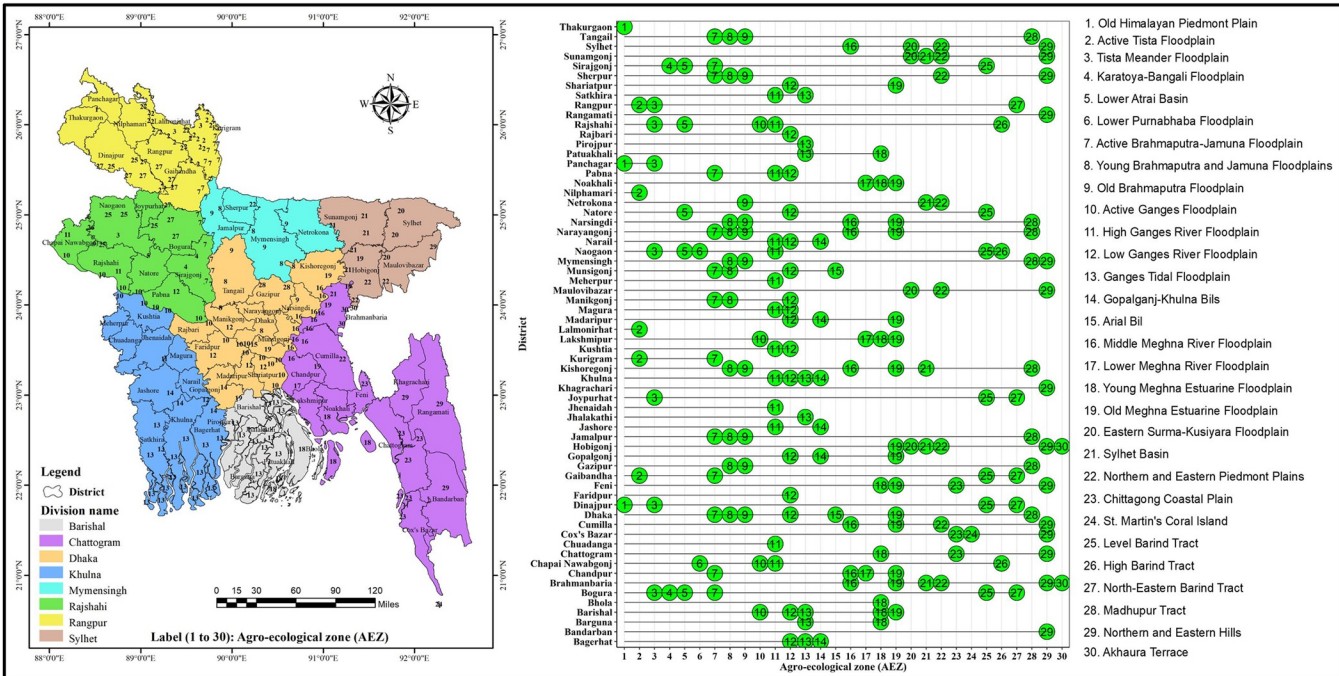

**Fig 1.** Spatial distribution of the study area (left) and inter-district alignment (middle to the right) with the agroecological zone (AEZ) of Bangladesh. GIS map prepared by the authors by using the administrative shapefile of Bangladesh. Shapefile republished from the Bangladesh Agricultural Research Council (BARC) database (http://maps.barcapps.gov.bd/index.php) under a CC BY license, with permission from Computer and GIS unit, BARC, original copyright 2014.

$$or, lnY = lna + bt \qquad (1)$$

Where $Y$ is the cultivation area, production, and yield, $t$ is the time, $b$ is the growth parameter to be estimated, and $ln$ stands for natural logarithm. The estimated parameter of the exponential growth model depicts the nature and magnitude of the trend with significant tests using the student t-test [32]. All our statistical analyses in this study were done at the 5% probability level.

### 2.3. Statistical analysis for a multivariate approach

We analyzed the time series data using multivariate statistical techniques, specifically applying principal component analysis (PCA) and hierarchical cluster analysis (HCA). Because of the existence of different factors, cultivation area, yield, and production may be correlated. In this study, we employed multivariate analysis to account for the spatiotemporal variability present across the 64 districts. Specifically, we utilized a 42-time series attributes matrix (64×42) of cultivation area, production, and yield datasets, which may exhibit significant internal, spatial, and temporal correlations. Following the application of PCA, we obtained eigenvalues (42×1) and eigenvectors (42×42) (referred to as loadings). These eigenvalues indicate the variance explained by the principal components (PCs), and, corresponding to each eigenvalue, the loadings assign weights to the indices, enabling their transformation into PCs. The selection of the number of components is based on a point at which the remaining eigenvalues become relatively small and similar in size [33, 34]. The PCA features the capability to unveil such spatiotemporal complexities among the determinants. In addressing the issue, the R-mode operational input matrix [35], i.e., districts vs. attributes (time series observed values), is

formed such that the rows represent individual districts (64 nos.). The columns represent the long-term observed values for each year of cultivation area, production, and yield data (42 no.) [36]. Another method to determine the number of principal components involved examining a Scree Plot, which displays the ordered eigenvalues from largest to smallest.

Hierarchical cluster analyses were applied to the time series variables to similar group districts (spatial variability) by using Ward's method [37] and dynamic time warping (DTW) [38] for the estimation of the distance matrix. For the cluster analysis, variables such as cultivation area (ha), production (mt), and yield (t/ha) of each rice-growing season (Aus, Aman, and Boro) were utilized, ensuring their independence from one another. Thus, the independent PCs were used as input for the cluster analysis [36]. DTW is a family of algorithms that calculate the optimal local stretch or compression to align two-time series' time axes for the most efficient mapping of one onto the other. DTW generates the cumulative distance between the two series and, if necessary, the mapping or warping function itself. DTW has broad applications in econometrics, chemometrics, and general time series mining for classification and clustering purposes [38]. More precisely, the number of optimal clusters has been identified using the elbow method, which looks at the percentage of explained variance as a function of the number of clusters.

## 2.4. Relationship between adoption rate and rice production using regression analysis

A time series regression model was employed to examine the impact of high-yielding varieties (HYVs) adoption on rice production in Bangladesh. The empirical form of the regression model is presented below:

$$lnY_t = \beta_0 + \beta_1 X_{1t} + \beta_2 X_{2t} + \beta_3 X_{3t} + \cdots + \beta_{64} X_{64t} + \varepsilon_0 \tag{2}$$

Where, $lnY_t$ represents the rice production (in logarithmic form) in period t; $X_{1t}$, $X_{2t}$, $X_{3t}$,...,$X_{64t}$ are the adoption rate (% of area coverage) of HYVs of rice in different districts during period t; $\beta_0$ is the intercept term; $\beta_1$, $\beta_2$, $\beta_3$,...,$\beta_{64}$ are the regression coefficients corresponding to each independent variable; and, $\varepsilon$ represents the error term or residual. We employed robust standard errors to address the issue of heteroscedasticity in the data.

## 2.5. Statistical tools

This study used MS Excel to arrange and compile the secondary datasets. We performed all statistical analyses by using programming R software. The spatial data were organized in a district shape file to better understand the results, and all the outcomes were mapped using the programming R and ArcGIS10.3 software.

## 3. Results

### 3.1. Descriptive statistics of rice cultivation area, yield, and production

The data analysis reveals that, for most districts and across all seasons, the rice cultivation area, production, and yield data demonstrate adherence to the normality assumption. In the case of the cultivation area, data showed a normal distribution for 47 districts in the Aman area, followed by Boro (45) and Aus (38). The production data for the Boro season revealed a normal distribution for the majority of districts (53), followed by the Aman and Aus seasons. The time series data for Fifty-eight districts showed a normal distribution for the Boro, Aus, and Aman yields.

Fig 2 illustrates the fourteen years average of Aus, Aman, and Boro rice cultivation areas, productions, and yields for the 64 districts. The cultivation area of Aus, Aman, and Boro rice showed significant differences among the districts for the average 14 years period (2006–2007 to 2019–2020) (Fig 2A). Bhola (69,122 ha) and Joypurhat (89 ha) had the highest and lowest average Aus cultivation area, while Mymensingh (2,62,827 ha) and Bandarban (9,007 ha) had the highest and lowest Aman cultivation area, respectively. In the Boro season, the highest and lowest average cultivation areas were reported in Mymensingh (2,54,813 ha) and Barguna (491 ha), respectively. The average production of Aus, Aman, and Boro rice differed significantly (*p-value* ≤ 0.01) among the districts of Bangladesh (Fig 2B). The average production of the Aus season was highest in Cumilla (1,48,105 mt) and lowest in Joypurhat (208 mt). Dinajpur (6,36,040 mt) had the highest Aman rice production, while Narayanganj (14,981 mt) had the lowest. In the Boro season, the highest and lowest average rice production was in Mymensingh (9,60,248 mt) and Barguna (1416 mt), respectively. Mymensingh (16,18,040 mt) produced the most rice collectively, while Bandarban (53,535 mt) had the least.

Over the last fourteen years, the average milled rice yield for Aus, Aman, and Boro have significantly differed both spatially and temporarily across the 64 districts of Bangladesh (Fig 2C). The highest average yield was found in Cox's Bazar (2.62 tons/ha), while the lowest was in Faridpur (1.12 tons/ha) in the Aus season. In the Aman season, Khagrachari had the highest average yield (2.75 tons/ha), while Munsiganj had the lowest (1.04 tons/ha). Boro rice yield was highest in Gopalganj (4.60 tons/ha) and the lowest was in Patuakhali (2.26 tons/ha). Gopalganj and Dhaka (3.63 tons/ha) had the highest overall rice yield, while Patuakhali (1.73 tons/ha) had the lowest.

We calculated the standard error of the mean (SEM) (in percentage) to compare seasonal fluctuations in rice cultivation area, production, and yield in both temporal and spatial aspects (Figs 2 and 3). The results suggest that the Aus season had the largest SEM for the cultivation area, production, and yield at 7.09%, 8.56%, and 4.99%, while the Boro season had the lowest at 2.07%, 2.93%, and 1.76% for the cultivation area, production, and yield, respectively. The Aman season exhibited SEM values of 2.17% for cultivation area, 3.87% for production, and 2.91% for yield.

## 3.2. District-wise trend and growth rates for rice cultivation area, production, and yield

We have estimated the exponential growth rates of all districts in Bangladesh to identify the actual nature of rice cultivation areas, production and yield pattern based on both temporal and spatial aspects.

**3.2.1. Trend and growth rate of rice cultivation area.** The spatiotemporal variations in rice cultivation areas during the Aus, Aman, and Boro seasons in Bangladesh are illustrated in Fig 4. The exponential growth rate with its significance at a 5% level is also presented in Table 1. During the last 14 years (2006–2007 to 2019–2020), the annual growth rate of rice cultivation area in different districts of Bangladesh ranged from -24.91 to 87.01% for the Aus season, -3.47 to 5.06% for the Aman season and -11.22 to 7.67% for Boro season. Aus rice cultivation area decreased significantly (p ≤ 0.05 and negative correlation) over the year in Barishal, Chattogram, Dhaka, Feni, Gazipur, Gopalganj, Jashore, Jhalokathi, Khulna, Madaripur, Munsiganj, Mymensingh, Norail, Natore, Netrokona, Pabna, Patuakhali, Rajbari, and Sherpur districts. Aman rice cultivation area showed a significant negative correlation (p ≤ 0.05) with time for Bagerhat, Chattogram, Chuadanga, Cumilla, Jashore, Madaripur, Narayanganj, Narsingdi, Pabna, Pirojpur, and Satkhira districts. In the Boro season, rice cultivation in Barishal, Chattogram, Cumilla, Dhaka, Faridpur, Lalmonirhat, Madaripur, Meherpur,

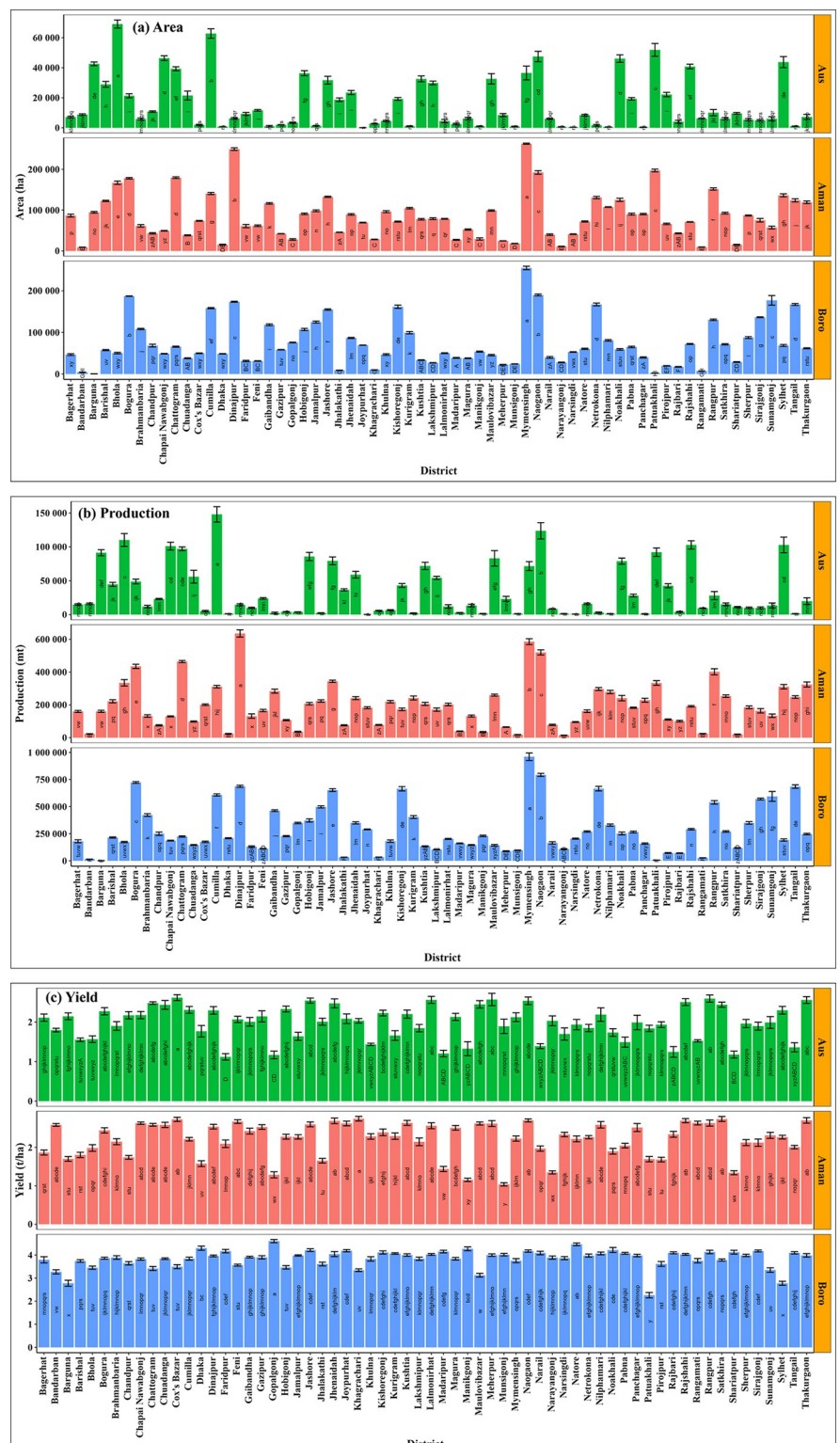

**Fig 2.** District-wise average (a) cultivation area, (b) production, and (c) yield of rice in Bangladesh for different seasons from 2006–2007 to 2019–2020.

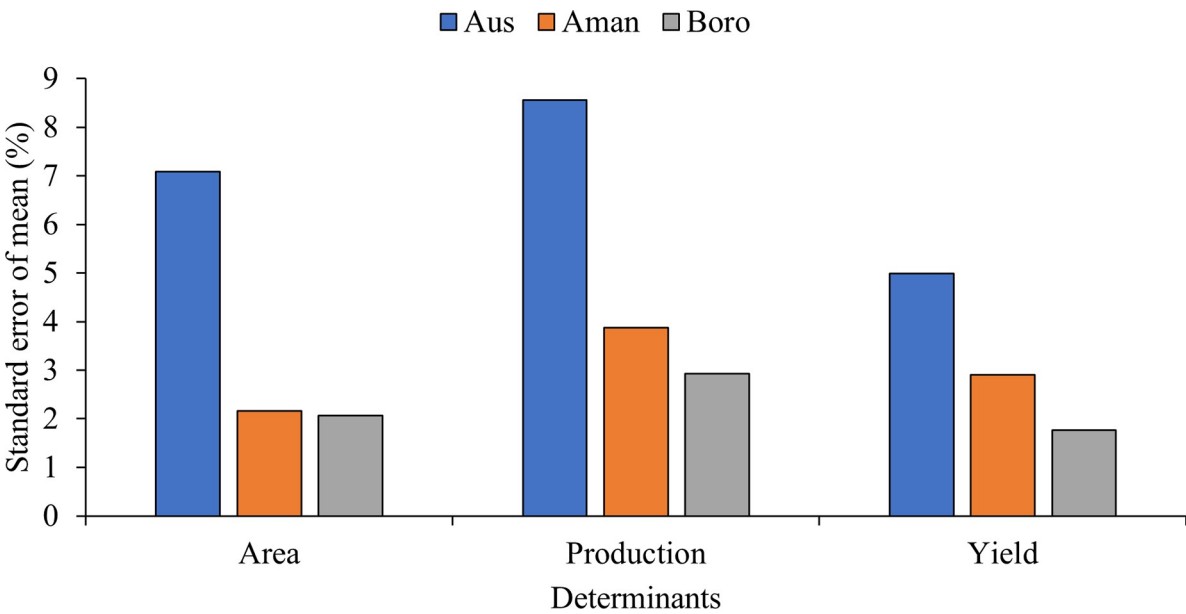

**Fig 3. Variation of the standard error of the mean (%) for rice production determinants of different seasons.**

Narayanganj, Natore, Pabna, Patuakhali, Rajbari, Rajshahi, and Shariatpur decreased significantly ($p \leq 0.05$) over time.

**3.2.2. Trend and growth rate of rice production.** From 2006–2007 to 2019–2020, the annual growth rate of rice production in different districts of Bangladesh (Fig 5) ranged from -20.14 to 99.47% for the Aus season, -1.82 to 9.23% for the Aman season and -7.49 to 10.79% for Boro season (Table 1). The results showed that rice production in the Aus season decreased significantly ($p \leq 0.05$) in Barishal, Jashore, Khulna, Madaripur, Munsiganj, Mymensingh, Netrokona, Patuakhali, Rajbari, and Sherpur. In Table 1, the Aman rice production showed no significant negative correlation with time among 64 districts. Aman rice production in the majority of districts has increased significantly. In the Boro season, the production growth rates of 22 districts decreased by between 0.1 and 7.49%, with Patuakhali, Rajbari, Faridpur, Meherpur, Pabna, Narayanganj, Rajshahi, and Natore exhibiting the most substantial declines. Out of the 64 districts in Bangladesh, about 45 districts showed a significant positive growth rate in rice production.

**3.2.3. Trend and growth rate of rice yield.** The spatiotemporal changing patterns of Aus, Aman, and Boro rice yield of the last 14 years of Bangladesh are presented in Fig 6. The annual growth rate of rice yield in different districts of Bangladesh ranged from -0.54 to 14.62% in the Aus season, -0.61 to 4.93% in the Aman season, and -0.49 to 3.73% in the Boro season during 2006–2007 to 2019–2020 (Table 1). Except for Rangamati and Cox's Bazar, the majority of the districts have found a positive association between Aus rice yield and time change. In the Aman season, except for Cox's Bazar, most districts have shown significant positive yield growth rates. Excluding the districts, Thakurgaon, Pabna, Rajshahi, Gaibandha, Lalmonirhat, Joypurhat, Bogura, Chuadanga, and Meherpur, the annual growth rate of rice yield has been increasing throughout the country during the Boro season.

### 3.3. Multivariate analysis of time series

**3.3.1. PCA interpretations.** The analysis of multivariate statistical techniques PCA is obtained to discover sets of attributes whose trends covariate similarly across districts. As all

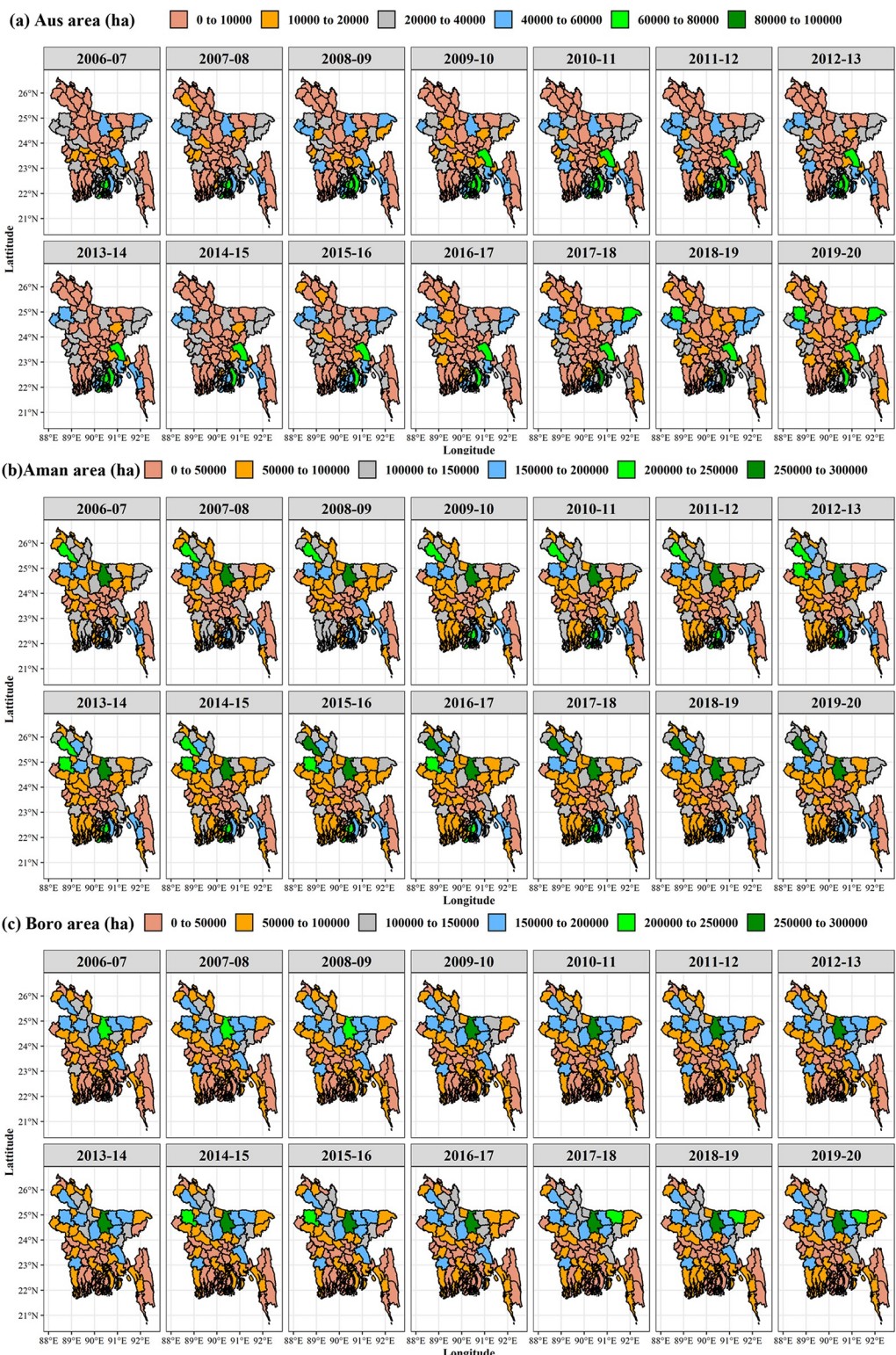

**Fig 4.** Spatial distribution of rice cultivation area (ha) in Bangladesh of (a) Aus, (b) Aman, and (c) Boro season from 2006–2007 to 2019–2020. GIS map prepared by the authors by using the administrative shapefile of Bangladesh. Shapefile republished from the Bangladesh Agricultural Research Council (BARC) database (http://maps.barcapps.gov.bd/index.php) under a CC BY license, with permission from Computer and GIS unit, BARC, original copyright 2014.

**Table 1. District-wise growth rate (%) of rice cultivation area, production, and yield in Bangladesh from 2006–07 to 2019–20.**

| Sl. | Districts | Cultivation area | | | | Production | | | | Yield | | | |
|---|---|---|---|---|---|---|---|---|---|---|---|---|---|
| | | Aus | Aman | Boro | Total | Aus | Aman | Boro | Total | Aus | Aman | Boro | Total |
| 1. | Bagerhat | -1.90 | -3.47* | 3.82* | -1.05* | 1.79 | -1.82 | 6.81* | 2.62* | 3.70* | 1.65 | 2.99* | 3.68* |
| 2. | Bandarban | 4.04* | 2.51* | 5.03* | 3.64* | 5.15* | 2.87* | 6.59* | 4.54* | 1.10 | 0.37 | 1.56* | 0.91* |
| 3. | Barguna | 0.55 | -0.97 | 7.67* | -0.42 | 3.57* | 1.20 | 10.79* | 2.07* | 3.02* | 2.17* | 3.12* | 2.49* |
| 4. | Barishal | -5.35* | 0.39 | -1.51* | -0.82* | -4.45* | 1.53 | -0.67 | -0.06 | 0.90 | 1.14 | 0.83* | 0.76 |
| 5. | Bhola | 2.91* | 1.48* | -1.25 | 1.38* | 6.47* | 5.17* | -0.35 | 3.88* | 3.56* | 3.69* | 0.90 | 2.49* |
| 6. | Bogura | 2.44 | 0.12 | -0.09 | 0.10 | 5.60* | 2.43* | -0.17 | 0.93* | 3.16* | 2.31* | -0.09 | 0.84* |
| 7. | Brahmanbaria | 7.30* | 1.10 | 0.60 | 0.94 | 12.30* | 4.14* | 1.85* | 2.50* | 5.00* | 3.03* | 1.25* | 1.56* |
| 8. | Chandpur | -2.40* | -1.34 | 0.30 | -0.50 | 1.09 | 0.27 | 0.66 | 0.58 | 3.49* | 1.62* | 0.36 | 1.07* |
| 9. | Chapai Nawabgonj | 1.13 | 0.73 | -0.47 | 0.43 | 4.59* | 0.82 | -0.41 | 1.16* | 3.46* | 0.10 | 0.06 | 0.73* |
| 10. | Chattogram | -1.75* | -0.89* | -1.45* | -1.14* | -0.68 | -0.41 | -0.10 | -0.35 | 1.07* | 0.48 | 1.35* | 0.78* |
| 11. | Chuadanga | 12.31* | -1.43* | -0.81 | 1.82* | 16.17* | 0.31 | -0.89 | 2.47* | 3.85* | 1.74* | -0.08 | 0.65* |
| 12. | Cox's Bazar | 7.78 | 0.89* | 0.80 | 0.91* | 7.24 | 0.28 | 2.79* | 1.47* | -0.54 | -0.61 | 2.00* | 0.56 |
| 13. | Cumilla | 3.88* | -1.39* | -0.58* | -0.20 | 6.92* | -0.46 | 0.15 | 0.79 | 3.04* | 0.93 | 0.73 | 0.99* |
| 14. | Dhaka | -4.68* | -1.59 | -0.77* | -1.05* | 2.78 | 0.62 | 0.04 | 0.05 | 7.47* | 2.20* | 0.80 | 1.11 |
| 15. | Dinajpur | -0.86 | 0.98* | 0.25 | 0.65* | 1.79 | 2.80* | 0.89* | 1.81* | 2.65* | 1.81* | 0.64* | 1.16* |
| 16. | Faridpur | -4.54 | 5.06* | -3.68* | 1.41 | 1.01 | 9.23* | -3.17* | 2.83* | 5.54* | 4.17* | 0.51 | 1.42* |
| 17. | Feni | -2.99* | 1.84* | -0.80 | 0.48 | -0.31 | 2.21 | -0.50 | 0.91 | 2.68* | 0.36 | 0.30 | 0.43 |
| 18. | Gaibandha | 39.75* | 0.97* | 1.53* | 1.38* | 41.13* | 3.57* | 1.22* | 2.17* | 1.37 | 2.59* | -0.31 | 0.80* |
| 19. | Gazipur | -5.21* | 0.42 | 0.02 | 0.08 | 0.14 | 0.81 | 1.08* | 0.99 | 5.35* | 0.40 | 1.06* | 0.90* |
| 20. | Gopalgonj | -5.94* | -0.86 | 0.13 | -0.38 | -0.40 | 4.08 | 1.27* | 1.50* | 5.54* | 4.93* | 1.13* | 1.88* |
| 21. | Hobigonj | 3.91* | 1.18* | 0.96 | 1.49* | 6.16* | 2.30* | 1.56 | 2.32* | 2.26* | 1.12 | 0.60 | 0.84 |
| 22. | Jamalpur | 8.63* | 0.66 | 1.17* | 0.99* | 13.28* | 2.33* | 1.29* | 1.63* | 4.65* | 1.66* | 0.12 | 0.63* |
| 23. | Jashore | -5.80* | -0.78* | 0.65* | -0.51 | -4.25* | 0.85 | 1.55* | 0.96* | 1.55* | 1.63* | 0.90* | 1.46* |
| 24. | Jhalakathi | -5.36* | 0.66* | 0.88 | -0.72* | -2.11 | 1.77 | 2.13* | 0.92 | 3.25* | 1.11 | 1.25* | 1.64* |
| 25. | Jhenaidah | 3.92* | 1.03* | -0.07 | 0.87* | 7.69* | 3.19* | 1.96* | 2.88* | 3.77* | 2.16* | 2.03* | 2.01* |
| 26. | Joypurhat | 6.43 | 0.77* | 0.06 | 0.42* | 10.48 | 2.77* | -0.12 | 1.02* | 4.06* | 2.00* | -0.17 | 0.60* |
| 27. | Khagrachari | 2.54* | -0.33 | 0.69 | 0.14 | 3.77* | 0.83 | 1.57* | 1.21 | 1.23 | 1.16* | 0.88* | 1.08* |
| 28. | Khulna | -6.12* | -0.52 | 3.23* | 0.43 | -5.62* | 1.38 | 4.83* | 2.78* | 0.50 | 1.90* | 1.60* | 2.34* |
| 29. | Kishoregonj | 1.76 | 0.99* | -0.30 | 0.26 | 4.01* | 3.98* | 0.75 | 1.52* | 2.25* | 2.98* | 1.05* | 1.27* |
| 30. | Kurigram | 11.69 | 1.04* | 2.34* | 1.72* | 17.90* | 4.05* | 2.73* | 3.24* | 6.21* | 3.00* | 0.40* | 1.52* |
| 31. | Kushtia | -1.37 | 1.88* | 0.56 | 0.89 | 2.60 | 3.66* | 0.94 | 2.58* | 3.97* | 1.77* | 0.38 | 1.70* |
| 32. | Lakshmipur | -1.96 | 0.87 | -0.31 | 0.05 | 1.37 | 4.94* | 0.79 | 2.98* | 3.33* | 4.07* | 1.11* | 2.93* |
| 33. | Lalmonirhat | 43.56* | 0.88* | -0.93* | 0.64* | 46.07* | 3.44* | -1.11 | 1.55* | 2.51 | 2.56* | -0.18 | 0.91* |
| 34. | Madaripur | -10.27* | -2.08* | -1.71* | -2.24* | -5.27* | 1.16 | -0.68 | -0.44 | 5.00* | 3.25* | 1.03* | 1.80* |
| 35. | Magura | -5.26 | 2.36* | 0.36 | 1.17* | -2.92 | 4.07* | 0.81 | 2.18* | 2.35* | 1.71* | 0.45 | 1.01* |
| 36. | Manikgonj | 5.32 | 4.01 | -1.04 | 0.41 | 18.05* | 7.25* | 0.44 | 1.24 | 12.73* | 3.25* | 1.48* | 0.82 |
| 37. | Maulovibazar | 8.32* | 0.27 | 2.47* | 2.32* | 11.60* | 1.02 | 4.38* | 3.72* | 3.28* | 0.75 | 1.90* | 1.40* |
| 38. | Meherpur | 10.45* | 0.03 | -2.43* | 0.65 | 15.59* | 2.06* | -2.50* | 1.38* | 5.14* | 2.04* | -0.08 | 0.73* |
| 39. | Munsigonj | -24.91* | 1.46 | -0.28 | 0.04 | -16.04* | 1.88 | 0.59 | 0.64 | 8.87* | 0.43 | 0.87* | 0.60 |
| 40. | Mymensingh | -12.06* | 0.01 | 1.08* | -0.22 | -7.77* | 2.47* | 2.90* | 2.27* | 4.29* | 2.46* | 1.82* | 2.49* |
| 41. | Naogaon | 6.56* | -0.88 | 0.53* | 0.51 | 9.49* | 0.10 | 1.08* | 1.39* | 2.93* | 0.97* | 0.56 | 0.88* |
| 42. | Narail | -2.76* | -2.00 | 3.21* | 0.29 | 0.77 | 0.63 | 3.75* | 2.57* | 3.53* | 2.63* | 0.54 | 2.28* |
| 43. | Narayangonj | -2.81 | -3.15* | -2.59* | -2.61* | -0.29 | -1.81 | -1.79* | -1.71* | 2.51 | 1.34 | 0.79 | 0.91 |
| 44. | Narsingdi | -1.73 | -1.35* | -0.67 | -0.96* | 5.84 | -0.18 | -0.27 | -0.23 | 7.57* | 1.17* | 0.39 | 0.73 |
| 45. | Natore | -3.31* | 1.53* | -1.51* | -0.08 | 2.54 | 4.15* | -1.09* | 0.86* | 5.85* | 2.62* | 0.42 | 0.94* |
| 46. | Netrokona | -22.82* | 0.39 | 0.88 | 0.52 | -20.14* | 1.71* | 2.00* | 1.81* | 2.67* | 1.32* | 1.12* | 1.29* |

*(Continued)*

**Table 1.** (Continued)

| Sl. | Districts | Cultivation area | | | | Production | | | | Yield | | | |
|---|---|---|---|---|---|---|---|---|---|---|---|---|---|
| | | Aus | Aman | Boro | Total | Aus | Aman | Boro | Total | Aus | Aman | Boro | Total |
| 47. | Nilphamari | 35.75 | 0.14 | 1.43* | 0.73* | 50.36 | 2.50* | 2.40* | 2.45* | 14.62 | 2.36* | 0.96* | 1.72* |
| 48. | Noakhali | -0.44 | 2.50* | 2.15* | 1.87* | 3.60* | 5.66* | 4.27* | 4.69* | 4.05* | 3.16* | 2.11* | 2.82* |
| 49. | Pabna | -2.18* | -1.38* | -1.77* | -1.59* | 5.59* | 0.83 | -2.21* | -0.55 | 7.77* | 2.21* | -0.44 | 1.04* |
| 50. | Panchagar | 87.01* | 1.46* | -1.77 | 0.56* | 99.47* | 4.66* | -0.93 | 2.41* | 12.46* | 3.20* | 0.83* | 1.86* |
| 51. | Patuakhali | -7.49* | -0.56 | -11.22* | -1.94* | -3.83* | 2.76* | -7.49* | 1.30 | 3.67* | 3.31* | 3.73* | 3.25* |
| 52. | Pirojpur | -2.87 | -1.52* | 2.30* | -1.07* | 0.42 | 0.41 | 4.49* | 1.77* | 3.29* | 1.93* | 2.19* | 2.84* |
| 53. | Rajbari | -18.99* | 0.58 | -3.89* | -1.59* | -13.93* | 2.92* | -3.46* | 0.09 | 5.07* | 2.34* | 0.43 | 1.68* |
| 54. | Rajshahi | 1.56 | 0.19 | -1.04* | 0.00 | 4.45* | 1.53* | -1.36* | 0.58 | 2.89* | 1.34* | -0.32 | 0.58* |
| 55. | Rangamati | 0.72 | 0.74 | 0.35 | 0.58 | 0.50 | 2.13* | 2.18 | 1.87* | -0.22 | 1.39* | 1.83* | 1.29* |
| 56. | Rangpur | 53.59* | 1.54* | 1.05* | 1.84* | 57.80* | 4.09* | 2.47* | 3.55* | 4.20* | 2.55* | 1.42* | 1.71* |
| 57. | Satkhira | 10.81* | -1.16* | 1.37* | 0.13 | 12.28* | 0.36 | 2.17* | 1.43* | 1.47* | 1.52* | 0.80* | 1.30* |
| 58. | Shariatpur | -1.69 | -0.08 | -2.63* | -1.79* | 4.18* | 3.07 | -1.25 | -0.33 | 5.87* | 3.15* | 1.38* | 1.46* |
| 59. | Sherpur | -10.86* | 0.13 | 1.89* | 0.67* | -6.53* | 2.55* | 2.49* | 2.29* | 4.33* | 2.42* | 0.61 | 1.61* |
| 60. | Sirajgonj | 10.28* | 4.55* | 0.57* | 1.82* | 14.41* | 8.30* | 1.07* | 2.39* | 4.13* | 3.75* | 0.50* | 0.57 |
| 61. | Sunamgonj | 13.01* | 2.80* | 0.17 | 1.32 | 19.06* | 4.48* | 2.60 | 3.29 | 6.04* | 1.68* | 2.43* | 1.97* |
| 62. | Sylhet | 4.80* | 0.99 | 2.11* | 1.97* | 8.31* | 1.83 | 4.27* | 3.65* | 3.51* | 0.84 | 2.17* | 1.68* |
| 63. | Tangail | -2.60 | -1.26 | 0.74* | -0.12 | 2.84 | 0.49 | 1.49* | 1.19* | 5.44* | 1.75* | 0.76* | 1.31* |
| 64. | Thakurgaon | 47.71* | 2.33* | 0.24 | 2.29* | 50.55* | 3.78* | -0.25 | 2.60* | 2.84* | 1.45* | -0.49 | 0.32 |

Note: * indicate the significance at a 5% probability level

obtained principal components (PCs) are not statistically significant in the scree plot, we may consider stopping at the fifth principal component. The result found that the first five PCs have been explained about 93.21%, 97.14%, and 96.02% of the total variances for the Aus, Aman, and Boro seasons, respectively (**Fig 7A–7C**). Those mentioned first two PCs explain the maximum variation of the total variance of input attributes.

**3.3.2. Spatial distribution and clustering.** This section described the spatial distribution and cluster analysis of the district-level rice cultivation area, production, and yield in Bangladesh. The characterization of the cluster according to the performance of yield, production, and cultivated area in different seasons in different cluster groups is presented in the boxplot (**Figs 8–10**). The cluster analysis was carried out to identify spatial groups of different districts based on their similarities of long-term rice cultivation scenarios, i.e., cultivation area, production, and yield performance trends. The number of clusters was identified at this point, so the "elbow criterion" (*The elbow criterion is a method used in cluster analysis to determine the optimal number of clusters in a dataset. It involves plotting the variance explained by the clusters against the number of clusters. The plot typically resembles an arm, and the "elbow" or bend in the plot represents the point where the addition of more clusters does not significantly reduce the variance. This point is considered the optimal number of clusters for the given dataset. The elbow criterion helps in selecting a reasonable number of clusters that balance capturing meaningful patterns in the data while avoiding overfitting.*) in the scree plot depicted the number of cluster groups: five for Aus, seven for Aman, and six for Boro season (**Fig 7A–7C**).

Based on the historical data of the Aus season, five cluster groups are depicted in **Fig 8(A)**: cluster 1 (C1), cluster 2 (C2), cluster 3 (C3), cluster 4 (C4), and cluster 5 (C5). The analysis indicates that the highly populated cluster 1 comprises a significant number of districts from the Rangpur, Mymensingh, and Dhaka divisions (**Fig 8B**). Clusters 1 and 2 have the smallest

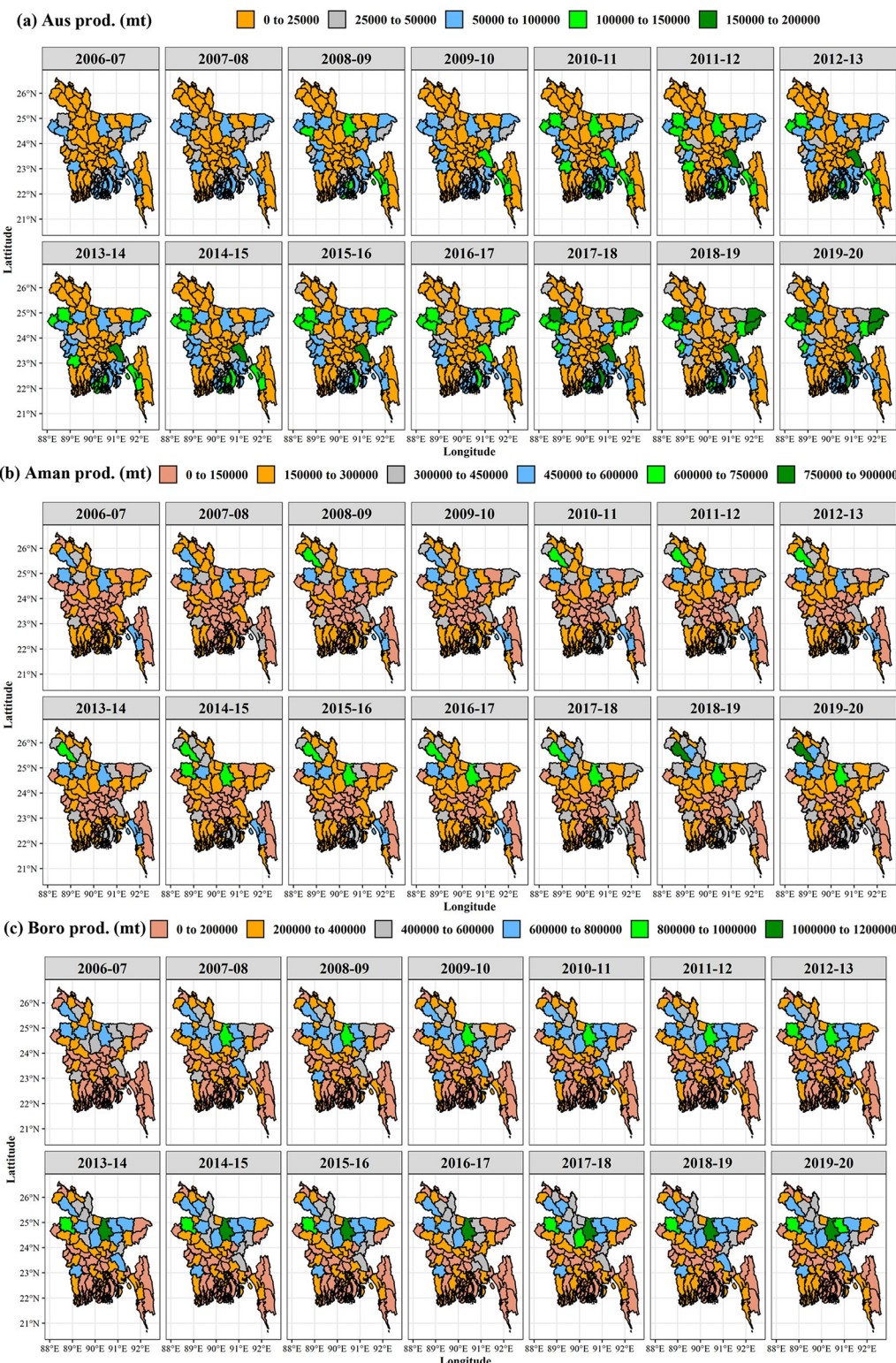

**Fig 5.** Spatial distribution of rice production (mt) in Bangladesh of Aus (a) Aus, (b) Aman, and (c) Boro season from 2006–2007 to 2019–2020. GIS map prepared by the authors by using the administrative shapefile of Bangladesh. Shapefile republished from the Bangladesh Agricultural Research Council (BARC) database (http://maps.barcapps.gov.bd/index.php) under a CC BY license, with permission from Computer and GIS unit, BARC, original copyright 2014.

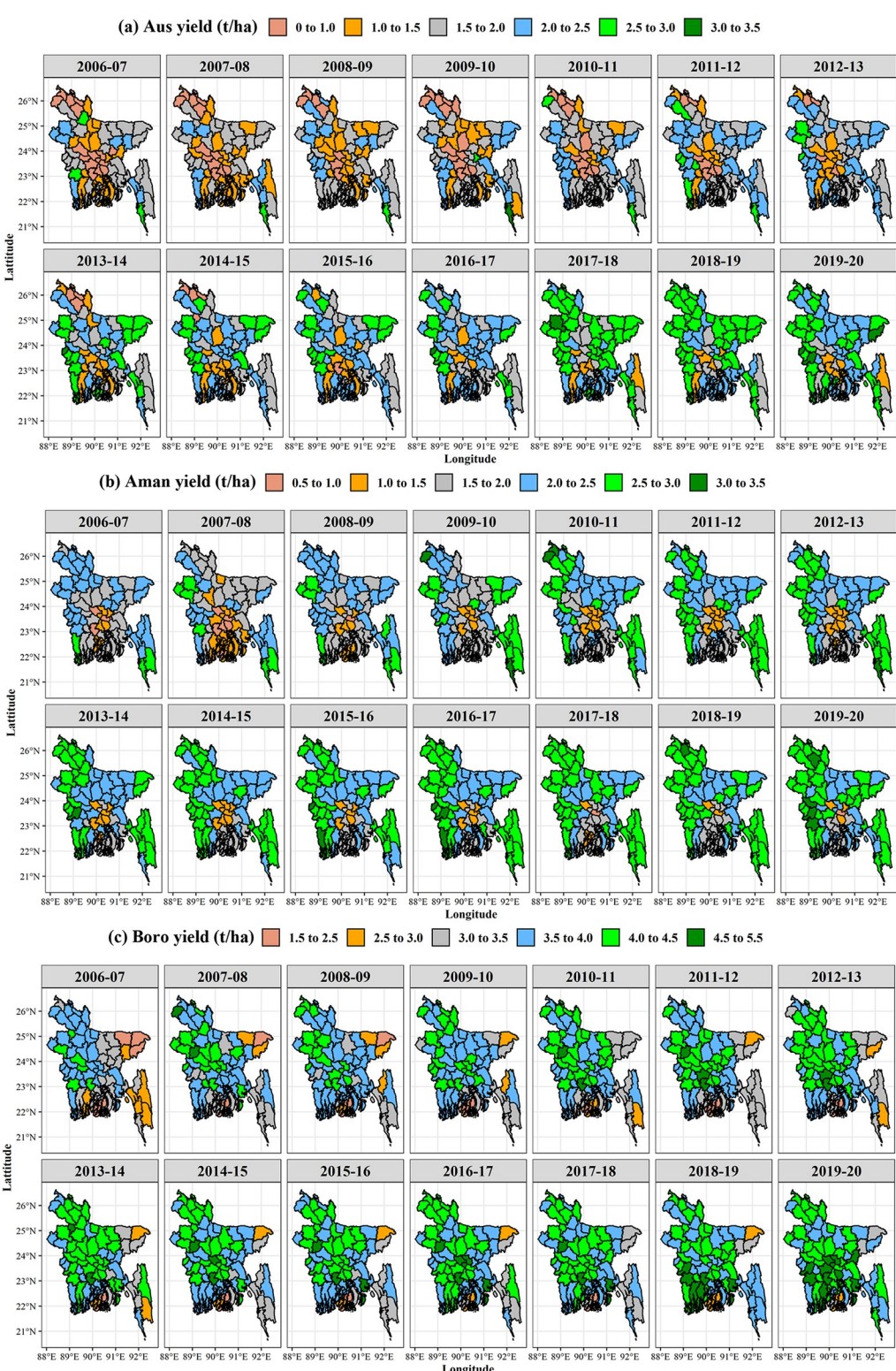

**Fig 6.** Spatial distribution of rice yield (ton/ha) in Bangladesh of (a) Aus, (b) Aman, and (c) Boro season from 2006–2007 to 2019–2020. GIS map prepared by the authors by using the administrative shapefile of Bangladesh. Shapefile republished from the Bangladesh Agricultural Research Council (BARC) database (http://maps.barcapps.gov.bd/index.php) under a CC BY license, with permission from Computer and GIS unit, BARC, original copyright 2014.

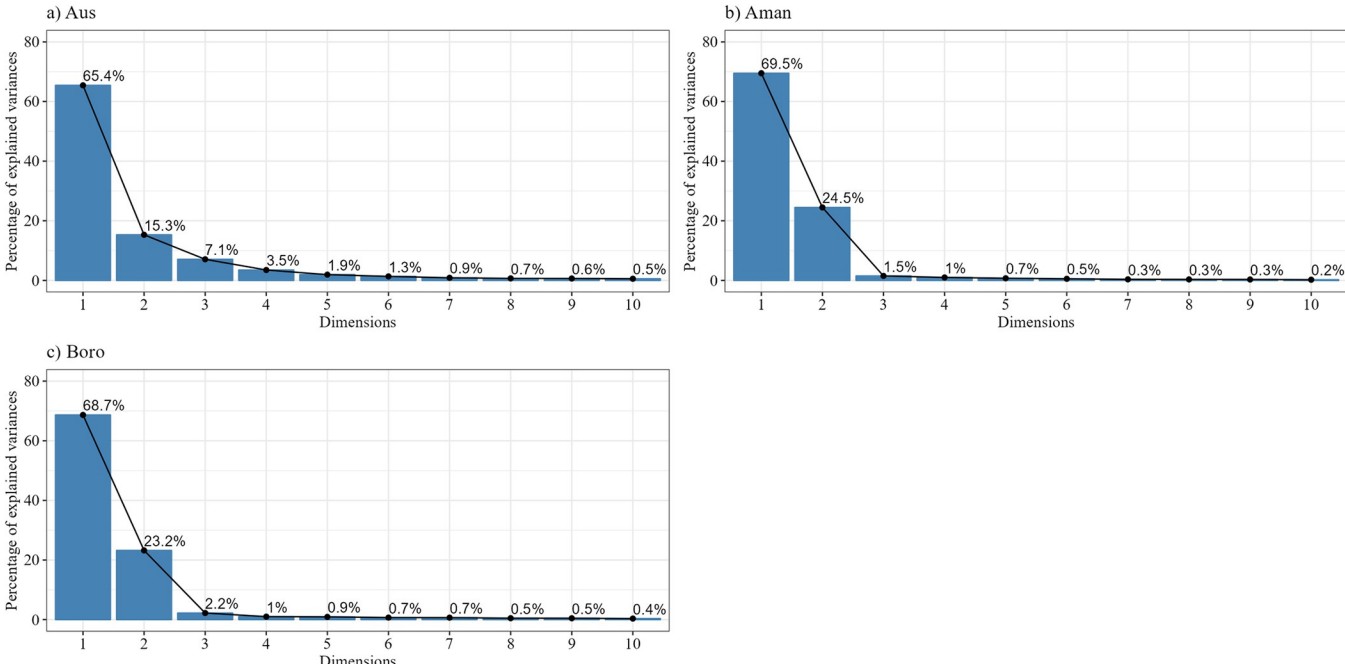

**Fig 7.** Percentage of explained variance for ten principal components of different seasons (a) Aus, (b) Aman, and (c) Boro rice.

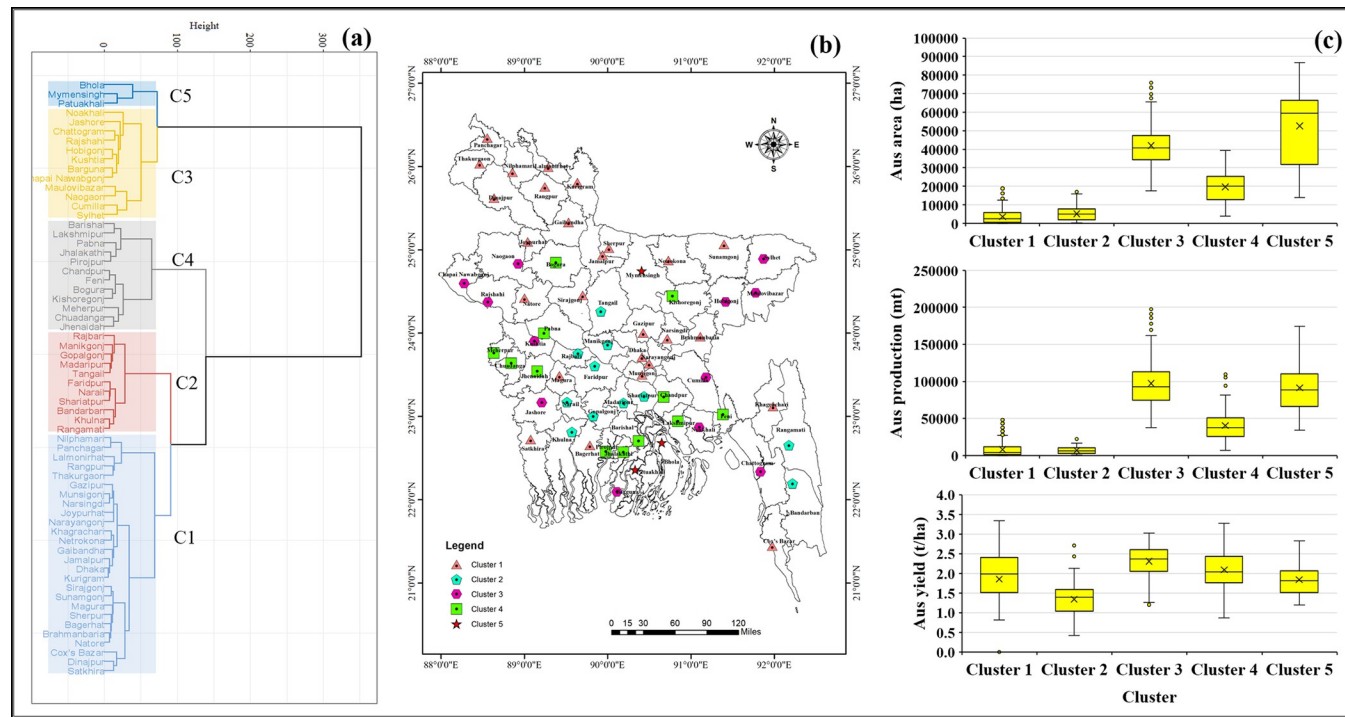

**Fig 8. Spatial clustering and classification of rice-growing districts based on cultivation area, production, and yield in the Aus season.** (a) dendrograms based on their spatial similarity; (b) spatial representation of the clusters on a GIS map; and (c) distribution of clusters. GIS map prepared by the authors by using the administrative shapefile of Bangladesh. Shapefile republished from the Bangladesh Agricultural Research Council (BARC) database (http://maps.barcapps.gov.bd/index.php) under a CC BY license, with permission from Computer and GIS unit, BARC, original copyright 2014.

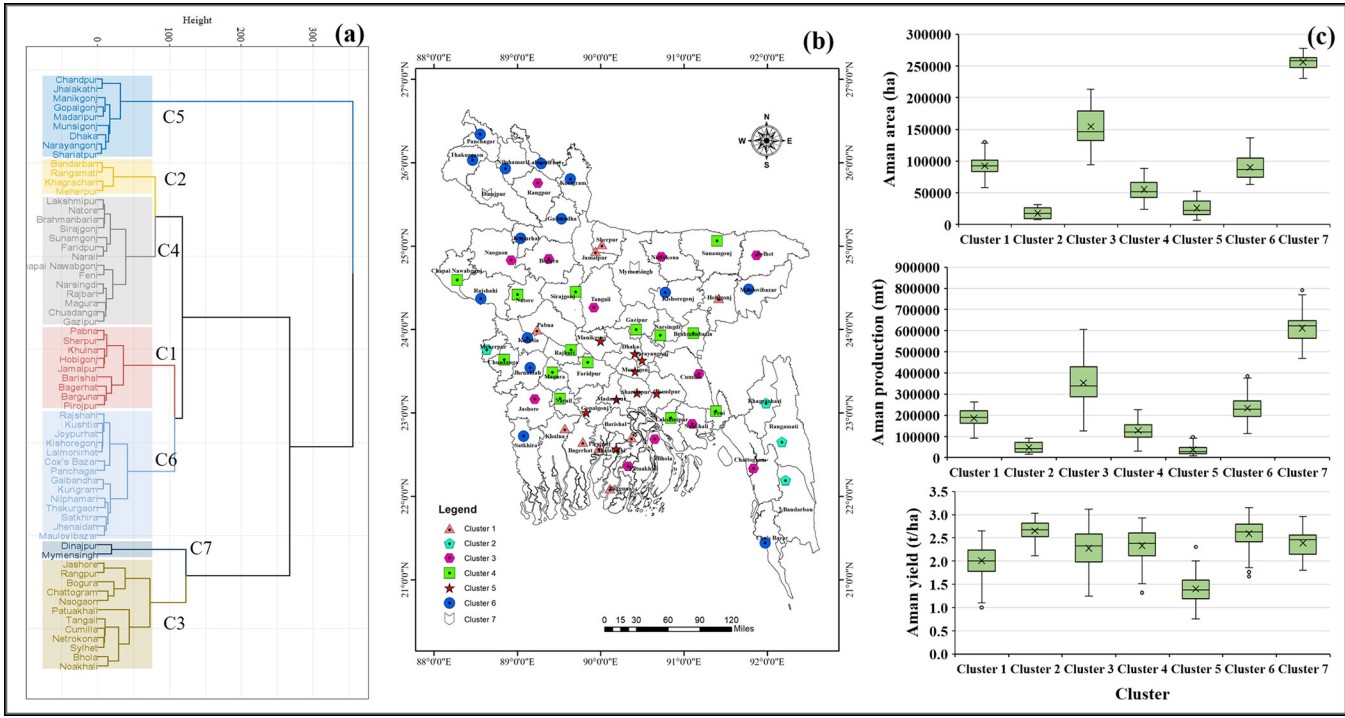

**Fig 9. Spatial clustering and classification of rice-growing districts based on cultivation area, production, and yield in Aman season.** (a) dendrograms based on their spatial similarity; (b) spatial representation of the clusters on a GIS map; and (c) distribution of clusters. GIS map prepared by the authors by using the administrative shapefile of Bangladesh. Shapefile republished from the Bangladesh Agricultural Research Council (BARC) database (http://maps.barcapps.gov.bd/index.php) under a CC BY license, with permission from Computer and GIS unit, BARC, original copyright 2014.

rice cultivation area and produce the lowest rice production compared to all other cluster groups. Cluster groups 3 and 5 also captured the districts with the highest land-use patterns and production contribution during the Aus season in Bangladesh. It was observed that most of the districts within all cluster groups had similar average yield performance except cluster 2 (Fig 8C).

Spatial groups corresponding to the Aman season were observed to be different compared to the historical rice cultivation area, production, and yield data. Seven clusters, i.e., C1, C2, C3, C4, C5, C6, and C7 (Fig 9A), were found in the Aman season based on the historical data of rice cultivation area, production, and yield. Clusters 2 and 5 had the minimum Aman area coverage and minimum production. The majority of the districts of C2 were located in the Northern and Eastern hills regions. Cluster 5 is mainly covered by the districts of Dhaka division (Fig 9B). Cluster 7 was the least populated cluster covering the maximum cultivated area and the highest rice production in the Aman season. Among the seven clusters, C3 had the second-highest average rice production and area cultivation in the Aman season (Fig 9C).

The spatial groups for the Boro season were observed in six clusters based on the historical data on rice cultivation area, production, and yield (Fig 10). Cluster 1 consisted of the highest number (18) of districts spatially distributed in the different divisions of Bangladesh. The second highest number (14) of districts was captured by cluster 6 (Fig 10A). Most districts are spatially distributed in Khulna and Dhaka divisions and covered by active, high, and low Ganges River floodplain regions (Fig 10B). Cluster 4 had the highest average Boro rice cultivated area and production among the cluster group, followed by C5, C1, C3, C6, and C2 (Fig 10C). Four spatial groups, C1, C4, C5, and C6 clusters, showed comparatively similar average yield performance, while C2 and C3 had the lowest average yield.

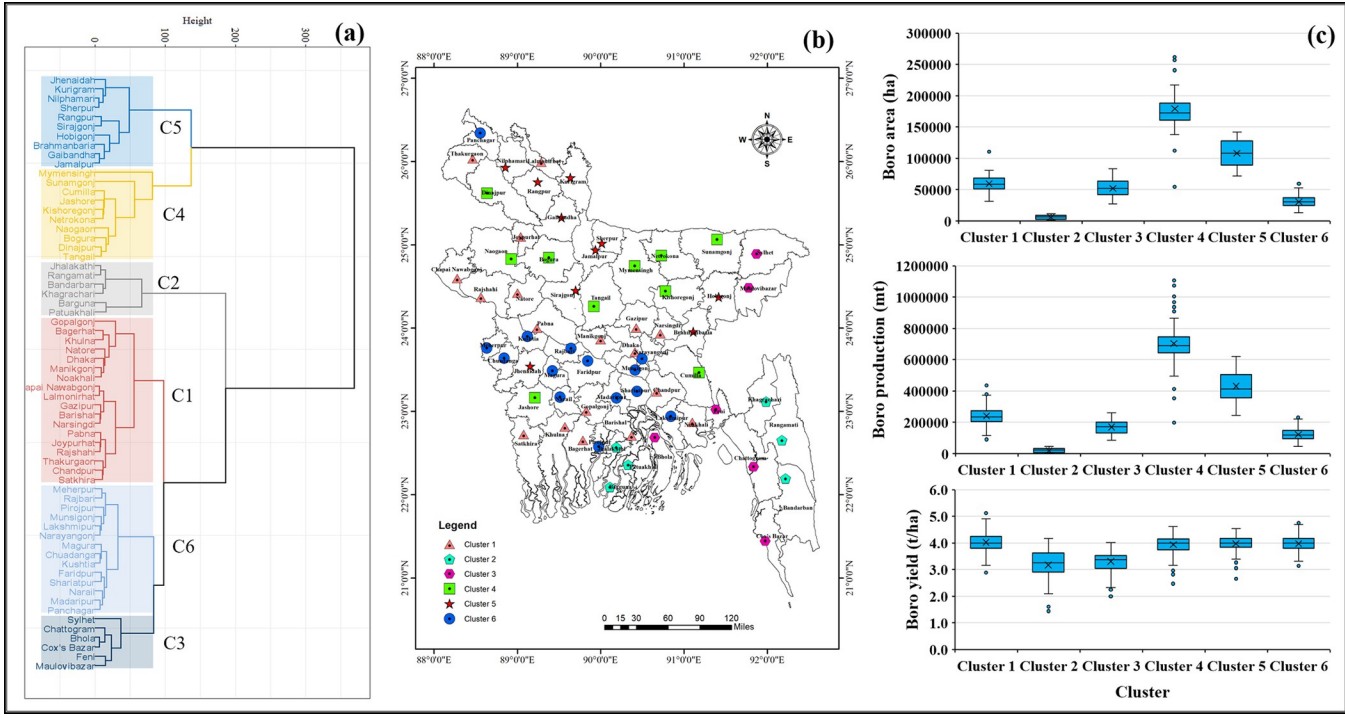

**Fig 10. Spatial clustering and classification of rice-growing districts based on cultivation area, production, and yield in the Boro season.** (a) dendrograms based on their spatial similarity; (b) spatial representation of the clusters on a GIS map; and (c) distribution of clusters. GIS map prepared by the authors by using the administrative shapefile of Bangladesh. Shapefile republished from the Bangladesh Agricultural Research Council (BARC) database (http://maps.barcapps.gov.bd/index.php) under a CC BY license, with permission from Computer and GIS unit, BARC, original copyright 2014.

### 3.4. Spatiotemporal variation of high-yielding variety adoption in Bangladesh

The adoption rate (percentage of area covered) of high-yielding variety (HYV) by season at the country level is illustrated in **Fig 11**. Over the past 14 years (2006–2007 to 2019–2020), Bangladesh's annual HYV adoption of rice has gradually increased. Among the total cultivated areas, significant ($p \leq 0.05$) increasing trends of HYV adoption were observed in Aus (1.97%), Aman (1.06%), and Boro (0.29%) seasons, with standard error (SE) representing the range of regional variability.

The spatial distribution and temporal variation of the HYV adoption rate of Aus, Aman, and Boro rice is shown in **Fig 12**. For the spatial distribution, we have employed the mean value of the 14-year data for each district, which is represented in the figure as 'HYV adoption (%)'. The temporal variability of HYV adoption was measured by a coefficient of variation, i.e., CV (%). The result showed that a low CV (%) value in the districts with a high HYV adoption rate increased yield trend, while a high CV in the districts with a low HYV adoption rate showed mild increasing yield trends for all seasons. In Aus season (**Fig 12A**), except Shariat-pur, Madaripur, Faridpur, Rajbari, Gopalganj, Rangamati, Tangail, Khulna, and Narail, the majority of the districts have the highest increasing HYV adoption rate and the lowest adoption variability. In Aman season (**Fig 12B**), the districts of Munsiganj, Manikganj, Gopalganj, Narayanganj, Shariatpur, Madaripur, Pirojpur and Jhalakathi in the Dhaka division and all districts of Barishal divisions had the lowest HYV adoption rate (less than 25%) and the highest adoption variability (CV greater than 35%). Most districts throughout the Boro season (**Fig 12C**) reported high adoption of high-yielding varieties and the lowest adoption variability. Only Patuakhali, Barguna, and Sylhet had the lowest HYV adoption rate and the highest adoption variations.

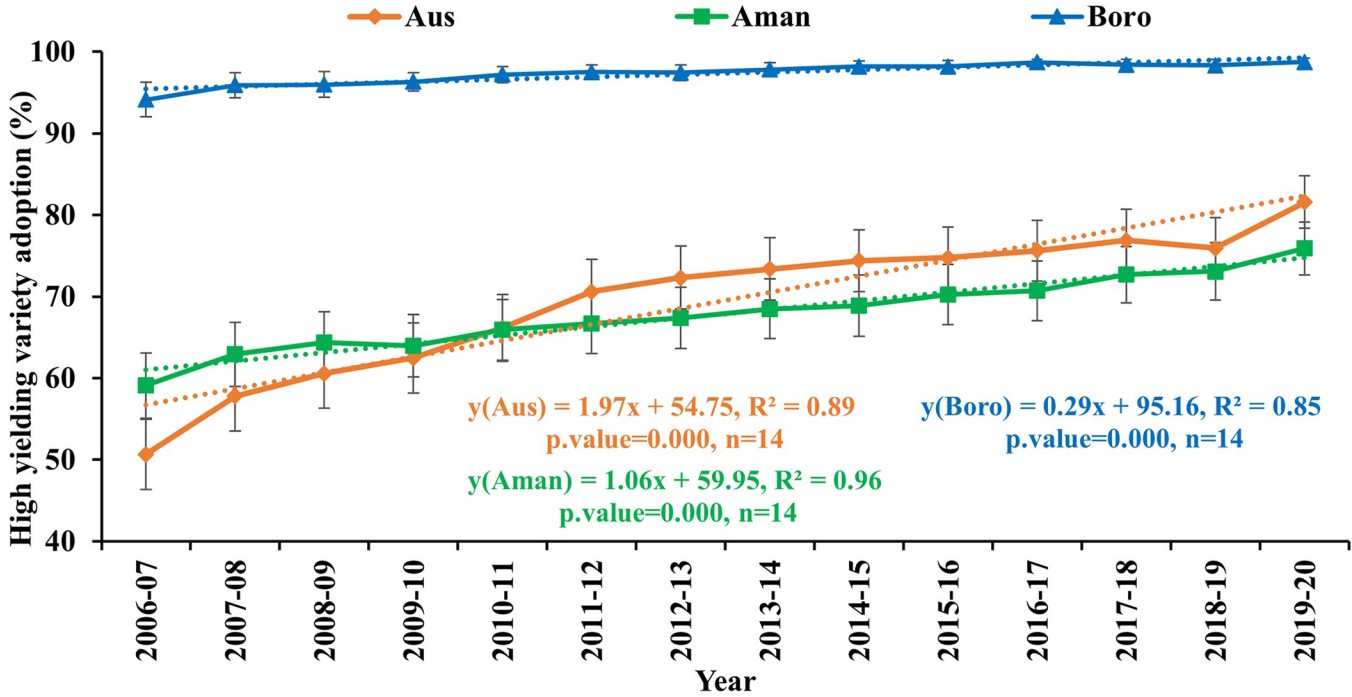

**Fig 11. Seasonal trend and growth rate of high-yielding variety adoption (% of area coverage) in Bangladesh from 2006–2007 to 2019–2020.**

## 3.5. Impact of high-yielding variety adoption on rice production

This section shows the impact of technological advancement, more specifically focusing on examining the impact of HYV adoption on rice production in various districts of Bangladesh, assuming all other factors remained constant. In this analysis, HYV adoption served as a proxy for technological advancement. To the best of our knowledge, the extent to which this technological advancement (HYVs) has influenced rice production in specific seasons and districts of Bangladesh has not been previously revealed. This study provides novel evidence in this regard, shedding light on the relationship between HYV adoption and rice production at the district level (**Fig 13**). Our findings revealed that a 10% increase in HYV adoption led to a varying range of rice production increase, ranging from 0.04% to 5.8% across different seasons, with a few exceptions. In the Aus season, a significant increasing trend (0.007% to 0.521%) was observed in 28 districts, with an increase of 1% adoption of HYVs, while a significant decrease (0.005% to 0.070%) in production was seen in five districts. In Aman season, the trend was significant and positive (0.004% to 0.039%) for 34 districts, whereas a significant negative trend (0.006% to 0.021%) was found in four districts. Boro season also followed the same trend, where 36 districts exhibited a significant increasing trend (0.013% to 0.584%), and six districts showed significant negative trends (0.090% to 0.426%). Overall, the production advantage due to HYV adoption was higher in Aus and Boro compared to the Aman season.

## 3.6. Future prediction of rice cultivation area and production change in Bangladesh (2020–2030)

Here, our aim was to identify the future challenges related to rice cultivation area and production changes in Bangladesh. We accomplished this by analyzing the existing trends from 2006–2007 to 2019–2020, and subsequently predicting and presenting the rice production and cultivation area in **Fig 14.** In this study, our prediction time was restricted to align with the

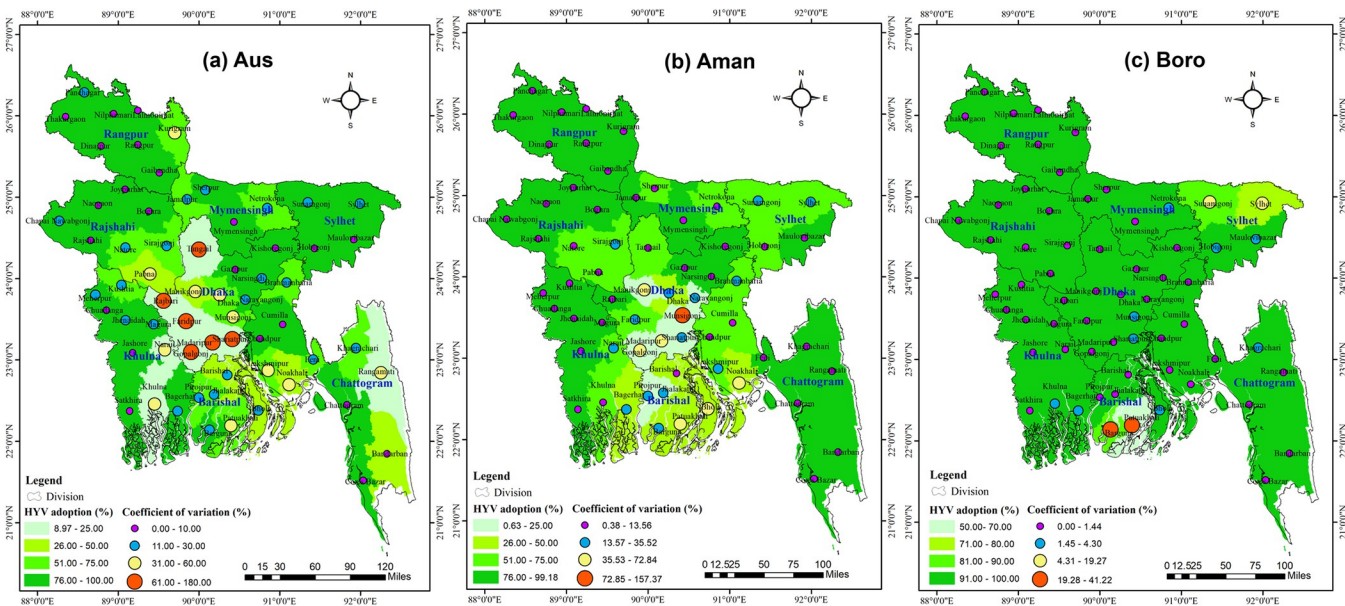

**Fig 12.** The district-wise average adoption rate (% of area coverage) of high-yielding varieties (HYVs) and its temporal variation in Bangladesh during 2006–2007 to 2019–2020 of (a) Aus, (b) Aman, and (c) Boro rice. GIS map prepared by the authors by using the administrative shapefile of Bangladesh. Shapefile republished from the Bangladesh Agricultural Research Council (BARC) database (http://maps.barcapps.gov.bd/index.php) under a CC BY license, with permission from Computer and GIS unit, BARC, original copyright 2014.

global development agenda timeframe, which extends until 2030. It is worth noting that shorter prediction periods may yield higher accuracy and precision in forecasting outcomes. In the Aus season, the highest decreasing production rate (>5%) will be found in Mymensingh, Patuakhali, Pirujpur, Rangamati, Barishal, Jessore, Madaripur and Sherpur, whereas

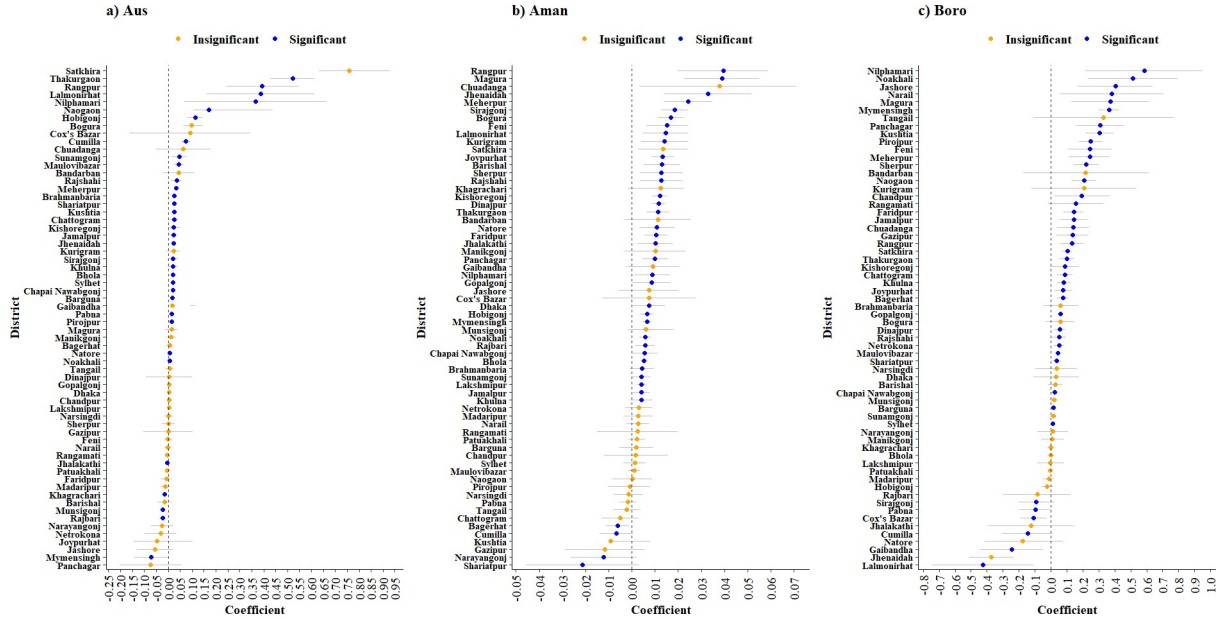

**Fig 13.** Impact of high-yielding variety adoption on (a) Aus, (b) Aman, and (c) Boro rice production in Bangladesh. Here, the positive coefficient indicates the increasing trend, the negative coefficient indicates the decreasing trend, and the coefficient value indicates the percentage change.

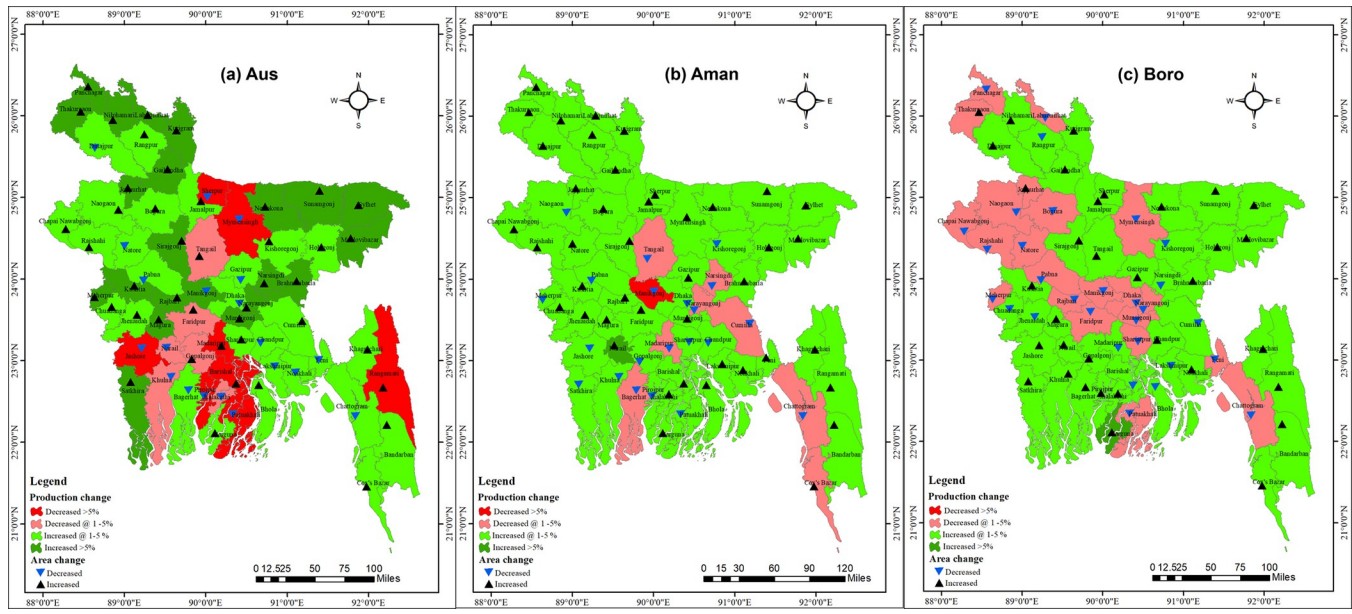

**Fig 14.** Prediction (2020–2030) of rice cultivation area and production change in (a) Aus, (b) Aman, and (c) Boro season of Bangladesh. GIS map prepared by the authors by using the administrative shapefile of Bangladesh. Shapefile republished from the Bangladesh Agricultural Research Council (BARC) database (http://maps.barcapps.gov.bd/index.php) under a CC BY license, with permission from Computer and GIS unit, BARC, original copyright 2014.

1–5% will be found in Faridpur, Gopalganj, Jhalokathi, Khulna, Narail and Tangial. On the other hand, >5% and 1–5% increasing production trends will be found in 22 and 28 districts, respectively. Aus cultivation area will be increased and decreased in 44 and 20 districts, respectively. During Aman, a more than 5% increase in production will be found only in Narail, where 1–5% will be noticed in 54 districts. Conversely, a more than 5% decrease in production has been projected in Manikganj and 1–5% in Bagerhat, Chattagram, Cumilla, Cox's Bazar, Madaripur, Narayanganj, Narsingdi, and Tangail. In the case of the rice cultivation area, the amount will be increased and decreased in 43 and 21 districts, respectively. For the Boro season, a more than 5% increase in production has been predicted in Borguna, whereas 1–5% in 41 districts. Oppositely, a 1–5% decreasing trend will be found in 22 districts, and no districts will show a decreasing trend of more than 5%. For area, it was forecasted that area would increase in 34 districts while it will be decreased in 30 districts.

## 4. Discussion

The rice-growing ecosystem in Bangladesh exhibits significant diversity, and several factors profoundly impact rice cultivation practices and productivity growth. The following are the main points of discussion.

*Land changing nature and geographical diversity*: Urbanization and human settlements reduce the thousands of agricultural lands in worldwide [39, 40] which consequences a threat to the environment and food security. Agricultural lands have been declining in Bangladesh for the past 30 to 40 years at a pace of 1% per year, while the percentage of urban areas has dramatically expanded [41]. The total amount of agricultural land decreased from 57.27% in 1992 to 42.82% in 2018, indicating an annual loss of 1.8% [42]. In the last 26 years, 25.6% of agricultural lands have been changed to other land cover types while there is a high conversion between agriculture and vegetation with rural settlements [42]. However, significant increase of per unit production of rice playing a vital role to ensure the food security of Bangladesh [5,

8]. The land type of the ecosystem mostly determines the rice cultivation area, cropping pattern, and cropping intensity [43]. Also, the favorable and unfavorable environment, farmers' selection of crops, and crop profitability influence the rice cultivation area [44].

According to our findings, Mymensingh district's dominance in rice production and area coverage can be attributed to a favorable environment for rice cultivation, including suitable topography, stagnant water in 80% of its area, low water flow recession, and high clay soil content [5]. Additionally, Dinajpur is the second largest rice cultivation area in the Aman season because of favorable environment, market potential for aromatic rice, and well-structured rice processing and distribution network. The bulk of the aromatic rice, along with other high-yielding varieties grown during the Aman season may lead to the second largest rice producing season in the country [9, 45]. Sunamgonj, Kishorgonj, Sylhet, Brahmanbaria, Habiganj, Moulvibazar, and Netrokona districts belong to the haor area (a wetland ecosystem). The *Haor* region is a significant contributor to the overall rice production in the country, especially Boro rice; however, there exist various challenges and limitations that impede the production of rice in this region, such as inadequate transportation facilities, improper usage of fertilizers, a shortage of quality seeds, deficiency of potassium nutrients, various rice diseases and insects such as *blast*, *bakanae*, and *brown plant hopper*, flash floods, and cold injury [46–50]. Developing short-duration with high-yielding rice varieties, improving transportation facilities, and implementing farmers' training programs on nutrient and disease management are essential steps to overcome the existing problems and increase rice production in the region. On the contrary, Bandarban, Khagrachari, and Rangamati districts have the lowest rice cultivation area and production. The location of these districts in a hilly region, coupled with the absence of flat terrain, agricultural inputs, new technologies, and adequate infrastructure, accounts for the challenges encountered in rice cultivation [5, 51]. In districts such as Jhalokati, Meherpur, Munsigonj, Shariatpur, and Moulvibazar, the cultivation of local rice varieties has resulted in a relatively lower production contribution to the national food basket. However, from an economic perspective, these local varieties exhibit higher profitability compared to HYVs, making a significant contribution to increasing farmers' income [45] and bolstering the national economy.

*Technological advancements and adoption*: Sustainable agriculture heavily relies on technology. In the 1970s in Bangladesh, the introduction of irrigation systems, chemical fertilizers, and the modern rice cultivars IR5, IR8, and IR20, collectively referred to as "green revolution technologies," have long played major roles in rice production [52]. After that, gradual improvement occurs in domestic rice productivity by developing and adopting new technologies for rice cultivation. In 1990, the government's substantial subsidies on irrigation facilities and the release of two high-yield varieties of rice, BRRI dhan28 and BRRI dhan29 in 1994, paved the way for a second green revolution in Bangladesh [5]. These facilities and the high demand for rice act as a catalyst to show the positive association of 45 districts and the trend of rice production over the last 14 years. Our study revealed that Bangladesh's bulk of rice growing land is devoted to high-yielding varieties. As evidence, High Yielding Varieties (HYVs) adoption has expanded dramatically throughout the country over the past 36 years, with observed variations in adoption rates of 72.0% for Aus, 73.5% for Aman, and 98.5% for Boro season in 2020–21 [5]. Our result stated that the coverage is almost saturated in the majority of districts in Boro season compared to Aus and Aman. In several regions, adoption of improved crop technology was low (less than 50%) for Aus and Aman seasons. Perceptible differences exist in the spread of HYVs across districts for different seasons. For example, in the central and southern districts of Bangladesh, the adoption rate of HYVs is very low (less than 25%). Availability and accessibility frequently influence farmers' selection of rice varieties, hence increasing the adoption heterogeneity of HYV in some regions. Moreover, a new variety might

not be suitable for every district and may result in low yield due to environmental and edaphic factors and poor management. This situation sometimes convinces farmers to cultivate their old and traditional varieties instead of modern ones, leading to significant variation in adoption in some districts. However, the widespread cultivation of HYVs has the potential to displace many traditional rice varieties from cultivation, posing a threat to their preservation. Likewise, this is of grave concern to the cultural heritage of Bangladesh.

The adoption of advanced technology by farmers plays a crucial role in increasing rice production. Regarding yield, the Gopalganj district has the highest rice yield in Bangladesh, attributed to the high organic matter content in the soil, the dominance of the high-yielding Boro cultivation area [5], and the single cropping pattern of Boro-Fallow-Fallow, which makes Boro season the highest yielding rice season [43]. Afterward, technological advancements and the dissemination of modern high-yielding rice varieties in Dhaka district have resulted in its potential for higher rice yields compared to other districts. The Bangladesh Rice Research Institute (BRRI) and the Bangladesh Agricultural Research Institute (BARI) are located in the Gazipur district, closer to Dhaka. So, the dissemination of new agricultural inventions and the adoption of modern high-yielding crop varieties from institutions are facilitated for the surrounding regions of Gazipur. High yield is a significant challenge in Patuakhali and other coastal regions due to the prevalent issue of salinity in the soil. In response to this, BRRI has developed high-yield salt-tolerant rice varieties such as BRRI dhan47, BRRI dhan67, BRRI dhan76, and BRRI dhan77. However, farmers still prefer cultivating traditional low-yield salt-tolerant and tidal submergence varieties [53, 54]. Therefore, extensive campaigns and input provisions are required to encourage farmers to adopt these varieties to increase yield.

*Changes in production practices*: The overall rice production trend in Bangladesh is increasing. Unfortunately, rice production in some regions fell short of expectations and declined over time. In the coastal districts of Bagerhat and Barishal, and the hilly district Chattogram, rice cultivation areas and production are falling for irrigation constraints. Shrimp farming (known as *gher*) is more profitable than rice in coastal regions, so farmers switch rice fields for *gher* farms [55]. In the northwestern and hilly regions of Bangladesh, farmers have embraced a transformation in agricultural patterns by adopting diverse fruit-dominated farming systems instead of relying solely on rice monoculture. This shift is driven by factors such as erratic rainfall, fluctuations in temperature, a shortage of labourers, limited economic benefits, and difficulties in accessing agricultural inputs [56]. Relative profitability, labour shortages, and low land topography are the key determinants of converting rice fields into aquaculture farming in the districts of Mymensingh and Khulna [57–59]. Shortage of labour and low price of rice also discourage farmers from growing rice instead of other crops. Mechanization tools like introducing the combined harvester, reaper machine, and processing unit could be an alternate option for labours. Policymakers should develop marketing strategies so farmers can get profitable prices by cultivating rice. However, the rising male outmigration left women to manage the farm with limited access to resources. Participatory water governance and capacity building of women through various training programs can be beneficial in reviving rice production [60].

*Shifts in crop and varietal preferences*: Location-wise crop selection and varietal adoption depends on the farmers' choice, market value, and environmental factors. People of south western part prefer to eat bold grain rice (like BRRI dhan76, BRRI dhan77, and local varieties), where slender grain rice (BRRI dhan28, BRRI dhan34, BRRI dhan49, BRRI dhan50, and BRRI dhan63) are preferred by the people of northern and central regions of Bangladesh [61]. BRRI dhan51 and BRRI dhan52 are popular in submergence prone areas like Lalmonirhat, Kurigram, Sylhet, Cumilla, Faridpur, Rangpur, etc., while drought tolerant varieties (BRRI dhan33, BRRI dhan56, BRRI dhan57, and BRRI dhan71) are preferable in north western, central, and

part of southern regions [12]. Recently BRRI dhan89, BRRI dhan92 and Bangabandhu dhan100 are promising Boro varieties of Bangladesh and gaining popularity because of their high yielding capacity. BR11, BR21, BR22, BR23, BRRI dhan49, BRRI dhan71, and BRRI dhan87 are also very popular Aman rice varieties throughout the country. BRRI dhan48 is one of the top choice varieties to the farmers in Kharif-I season compared to local Aus varieties. BRRI developed hybrid varieties (BRRI hybrid dhan3, BRRI hybrid dhan5, and BRRI hybrid dhan7) gaining popularities throughout the country because of high yield and profitable price. In sum, the popularity of different varieties, reflects the diverse agricultural landscape and the importance of selecting suitable rice varieties for different agro-ecological conditions in the country. Additionally varietals adoption also depends on the farmers taste, grain structure, economic value, and even in gender who lead the households [62].

Farmers in Narsingdi, Pabna, Pirojpur, Rajbari, Madaripur, and Shariatpur districts have preferred cultivating vegetables and spices over rice production. This can be attributed to the higher profitability and demand for these crops, as well as the insufficient supply of rice seeds. To increase rice production in these regions, developing short-duration Aman varieties and the seed supply chain can enable farmers to cultivate winter vegetables after Aman rice (Experts from the Department of Agricultural Extension, personal communication, March 10, 2023). Urban areas, such as Dhaka, Chattogram, Narayonganj, and Narsingdi, which are centers of industrialization, have experienced a decline in rice production and cultivation areas. Industrialization reduces crop farming and induces people to engage in high-paying industrial work instead of laborious rice cultivation.

*Environmental challenges and adaptation*: Bangladesh is one of the climate vulnerable countries of the world [63]. Salinity, drought, flood and cold are becoming major challenges for rice cultivation in the country. Around two million ha of land in the south and southwestern regions of Bangladesh are already affected by salinity. For the adaptation, BRRI-developed saline-tolerant varieties like BRRI dhan47 and BRRI dhan67 are gaining popularity in the saline regions, but to cover more area, we need to develop higher saline (>16 dsm$^{-1}$) tolerant varieties at vegetative and reproductive stages. The northern regions of Bangladesh are prone to drought stress due to factors such as reduced rainfall, groundwater depletion, and inadequate water drainage [64]. Many cultivable lands are remained fallow due to lack of water and drought conditions. BRRI dhan33, BRRI dhan56, BRRI dhan57, and BRRI dhan71 are drought-tolerant varieties gaining popularity in northern drought-prone areas of Bangladesh. But to increase cultivation area coverage of rice in the drought-prone region, high drought-tolerant varieties with precision management systems like drip or sprinkler irrigation, direct seeded planting, and rainwater harvesting could be possible solutions. In the Boro season, most of the land of southern and northeastern regions remain fallow due to lack of irrigation water. The establishment of deep water and shallow water tubewell, and irrigation channels to use natural resources like rivers and canals could increase the rice cultivation area in Bangladesh. Flood is not a regular phenomenon in Bangladesh, but heavy rainfall and flash flood causes significant crop losses in the low-lying and flood-prone areas. Forecasts of heavy rainfall, dam making, and river digging could solve the flood problem in the country. Moreover, flood-tolerant varieties like quick regeneration capacity after the recession of water, sustaining ability under flood water for 21–25 days, stronger culm having lodging tolerant could be another way to withstand the challenge of flood problems.

*Rice crop zoning*: Effective rice crop zoning encourages the preservation of agricultural land, can stimulate locally grown crops, and aims to assist the overall agrarian economy. According to the spatial clustering of our analysis, districts in clusters 1 and 2 in the Aus season have low cultivable areas, which means area expansion is limited. However, expanding high-yielding varieties and other management practices can improve the production of these

areas. Districts in clusters 3, 4, and 5 also can enhance their rice production by expanding cultivable land, using high-yielding varieties, and adopting new technologies. In the case of Aman season, the focus should be given to clusters 1, 2, and 3 to increase rice cultivation area and production. For the Boro season, regions belonging to clusters 1, 4, and 5 have the potential to expand rice cultivation area and production.

## 4.1. Practical implications

The research findings have significant practical implications for informing and guiding future agricultural practices and policies in rice-intensive areas of Bangladesh. The study provides valuable insights into the dynamic changes in the area and production of rice cultivation at a more disaggregated level, representing diverse ecosystems and management constraints. Policymakers can utilize this information to develop targeted strategies and interventions aimed at enhancing rice production in specific regions. For instance, regions with favorable environmental conditions and fertile soil content can be encouraged to prioritize expanding rice cultivation. Efforts can be directed towards providing farmers in these regions with access to modern high-yielding varieties and advanced agricultural technologies to improve productivity. Conversely, areas facing challenges such as salinity or hilly terrain require tailored approaches to overcome these limitations, such as the development of stress-tolerant rice varieties and the implementation of appropriate irrigation systems. Additionally, identifying regions where rice production is declining, or farmers are shifting to alternative crops can guide initiatives aimed at revitalizing rice cultivation through improved irrigation facilities, training programs, and the development of short-duration varieties. The study highlights the areas that require closer attention to overcome challenges and enhance productivity, ultimately contributing to achieving Sustainable Development Goal 2.3 of doubling agricultural productivity and income for smallholder farmers by 2030. In summary, these practical implications can contribute to sustainable agricultural practices, enhance food security, and improve the livelihoods of farmers in Bangladesh.

## 4.2. Limitations of this study and future research scope

Despite the valuable insights provided by this study, there are certain limitations that should be acknowledged. First, this research focused primarily on the quantitative analysis of rice cultivation area and production trends, and did not delve into the underlying socio-economic factors that may influence these trends. Additionally, this study relied on secondary data sources, which may be subject to limitations such as data accuracy and availability. Conducting primary data collection through field surveys and interviews could enhance the accuracy and reliability of the findings. Furthermore, the study's scope was limited to a specific timeframe (2006–2019), and it is important to recognize that future changes in climate, technology, and agricultural practices could impact rice cultivation in unforeseen ways.

Building on the findings and limitations of this study, several areas for future research can further contribute to the understanding of rice cultivation in Bangladesh. Firstly, investigating the socio-economic factors that influence farmers' decision-making processes regarding rice cultivation, including factors such as market conditions, government policies, and farmers' preferences, would provide valuable insights into the drivers of rice production. Secondly, exploring the impact of climate change on rice cultivation, including the effects of rising temperatures, changing rainfall patterns, and increased occurrences of extreme weather events, is crucial for developing climate-resilient agricultural strategies. Additionally, examining the adoption and effectiveness of specific interventions aimed at improving rice productivity, such as the dissemination of high-yielding varieties, the implementation of irrigation systems, and

the provision of training and extension services, would help in identifying best practices and areas for improvement. Finally, integrating remote sensing and geospatial analysis techniques can enhance the accuracy and timeliness of monitoring rice cultivation dynamics, allowing for more precise and up-to-date assessments of cultivation area and production trends.

## 5. Conclusions and policy recommendations

To ensure the sustainability of rice production in Bangladesh, it is imperative to have a comprehensive understanding of the spatiotemporal distribution of rice cultivation area, production, and yield. This knowledge will enable the formulation of region-specific policies tailored to the specific needs of different areas. The study has introduced novel methodological approaches for trend analysis and spatial clustering. Our findings showed that 14 years averages of rice cultivation area, production, and yield for three major seasons, Aus, Aman, and Boro, differ significantly among the study districts in Bangladesh. The Aus season has the highest temporal variability of rice production determinants, followed by the Aman and Boro seasons. Regional disparities in production were revealed in five cluster groups for the Aus season, seven for the Aman, and six for the Boro season. The share of HYV adoption significantly increased for most of the season. A significant increasing trend in Aus (0.007–0.521%), Aman (0.004–0.039%), and Boro (0.013–0.584%) were observed in 28, 34, and 36 districts, respectively, with an increase of 1% adoption of HYV. Predictions revealed that more than 5% of rice production would be increased in 28 districts in the Aus season, and for Aman and Boro, more than 5% would be increased in Narail and Bogura, respectively. Moreover, a 1–5% increase will be found in 50, 54, and 41 districts in Aus, Aman, and Boro seasons, respectively. These findings underscore the importance of formulating tailored and targeted policies at the regional level to effectively enhance rice productivity in Bangladesh.

The following policy recommendations are crucial for addressing the challenges and maximizing the potential of rice production in the country.

i. Measures should be implemented to safeguard existing rice cultivation areas as the availability of arable land decreases. This includes preserving fertile lands, preventing land conversions, and implementing land-use planning strategies that prioritize agricultural purposes.

ii. In regions where there is an increasing trend in rice cultivation area, the adoption of high-yielding varieties should be promoted.

iii. Areas with low production can benefit from the adoption of precision management systems. These systems utilize advanced technologies to optimize resource utilization, enhance efficiency, and improve crop yields.

iv. To overcome the challenges faced in certain regions, such as salinity, drought, or flood-prone areas, tactical management approaches should be implemented.

v. In regions with high rice production, there should be a focus on building adequate storage facilities to prevent post-harvest losses. This will ensure food security and stabilize prices in the market.

vi. Providing farmers with access to quality inputs such as seeds, fertilizers, and pesticides, along with financial support in the form of subsidies and crop insurance, will help mitigate risks and encourage increased rice production.

vii. Strengthening water management systems and promoting the adoption of modern farming technologies are essential steps toward achieving sustainable increases in rice production.

viii. Policies should be implemented to ensure fair and remunerative prices for rice farmers.

ix. Last but not least, the government should prioritize investment in agricultural infrastructure, research and development, and extension services.

By implementing these policy recommendations, the government can create an enabling environment for sustainable rice production, ensure food security, and enhance the income and livelihoods of rice farmers in Bangladesh.

## Supporting information

**S1 Data.**
(XLSX)

## Acknowledgments

The authors express their sincere thanks to the Bangladesh Bureau of Statistics (BBS) for making available the relevant rice data. The authors also acknowledge several scientists of the Bangladesh Rice Research Institute for participating in the discussion at various stages of preparing the manuscript.

## Author Contributions

**Conceptualization:** Md. Abdullah Al Mamun, Sheikh Arafat Islam Nihad, Md Abdur Rouf Sarkar.

**Data curation:** Md. Abdullah Al Mamun.

**Formal analysis:** Md. Abdullah Al Mamun.

**Investigation:** Md. Abdullah Al Mamun, Mou Rani Sarker, Md Abdur Rouf Sarkar, Md. Ismail Hossain, Md. Shahjahan Kabir.

**Methodology:** Md. Abdullah Al Mamun, Sheikh Arafat Islam Nihad, Md Abdur Rouf Sarkar, Md. Ismail Hossain, Md. Shahjahan Kabir.

**Project administration:** Md. Abdullah Al Mamun.

**Resources:** Md. Abdullah Al Mamun, Mou Rani Sarker, Md Abdur Rouf Sarkar.

**Software:** Md. Abdullah Al Mamun.

**Supervision:** Md. Abdullah Al Mamun, Mou Rani Sarker, Md Abdur Rouf Sarkar, Md. Ismail Hossain, Md. Shahjahan Kabir.

**Validation:** Md. Abdullah Al Mamun.

**Visualization:** Md. Abdullah Al Mamun, Sheikh Arafat Islam Nihad.

**Writing – original draft:** Md. Abdullah Al Mamun, Sheikh Arafat Islam Nihad.

**Writing – review & editing:** Md. Abdullah Al Mamun, Sheikh Arafat Islam Nihad, Mou Rani Sarker, Md Abdur Rouf Sarkar, Md. Ismail Hossain, Md. Shahjahan Kabir.

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
