## [Decision Letter · Decision Letter 0]

9 May 2023

PONE-D-23-09867Spatiotemporal mapping of rice acreage and productivity growth in Bangladesh

PLOS ONE

Dear Dr. Al Mamun,

Thank you for submitting your manuscript to PLOS ONE. After careful consideration, we feel that it has merit but does not fully meet PLOS ONE’s publication criteria as it currently stands. Therefore, we invite you to submit a revised version of the manuscript that addresses the points raised during the review process.

We look forward to receiving your revised manuscript.

Kind regards,

Abul Khayer Mohammad Golam Sarwar

Academic Editor

PLOS ONE

Journal Requirements:

2. Thank you for submitting the above manuscript to PLOS ONE. During our internal evaluation of the manuscript, we found significant text overlap between your submission and previous work in the [introduction, conclusion, etc.].

Please revise the manuscript to rephrase the duplicated text, cite your sources, and provide details as to how the current manuscript advances on previous work. Please note that further consideration is dependent on the submission of a manuscript that addresses these concerns about the overlap in text with published work.

[If the overlap is with the authors’ own works: Moreover, upon submission, authors must confirm that the manuscript, or any related manuscript, is not currently under consideration or accepted elsewhere. If related work has been submitted to PLOS ONE or elsewhere, authors must include a copy with the submitted article. Reviewers will be asked to comment on the overlap between related submissions (http://journals.plos.org/plosone/s/submission-guidelines#loc-related-manuscripts).]

We will carefully review your manuscript upon resubmission and further consideration of the manuscript is dependent on the text overlap being addressed in full. Please ensure that your revision is thorough as failure to address the concerns to our satisfaction may result in your submission not being considered further

3. We note that Figure 1, 4, 8, 9, 10, 11, 12, 14 in your submission contain [map/satellite] images which may be copyrighted. All PLOS content is published under the Creative Commons Attribution License (CC BY 4.0), which means that the manuscript, images, and Supporting Information files will be freely available online, and any third party is permitted to access, download, copy, distribute, and use these materials in any way, even commercially, with proper attribution. For these reasons, we cannot publish previously copyrighted maps or satellite images created using proprietary data, such as Google software (Google Maps, Street View, and Earth). For more information, see our copyright guidelines: http://journals.plos.org/plosone/s/licenses-and-copyright.

1. You may seek permission from the original copyright holder of Figure(s) [#] to publish the content specifically under the CC BY 4.0 license.  

Reviewers' comments:

Reviewer's Responses to Questions

**Comments to the Author**

1. Is the manuscript technically sound, and do the data support the conclusions?

Reviewer #1: No

Reviewer #2: Yes

2. Has the statistical analysis been performed appropriately and rigorously? 

Reviewer #1: Yes

Reviewer #2: Yes

3. Have the authors made all data underlying the findings in their manuscript fully available?

Reviewer #1: No

Reviewer #2: Yes

4. Is the manuscript presented in an intelligible fashion and written in standard English?

Reviewer #1: No

Reviewer #2: No

5. Review Comments to the Author

Reviewer #1: In this paper, the authors performed a spatio-temporal mapping of the area, production, and yield of rice from 2006-2007 to 2019-2020 using secondary data for disaggregating 64 districts in Bangladesh. They also looked at the adoption rate of high-yielding varieties of rice and did cluster analysis. The authors of this paper has also published a similar paper (same lead author) in PLOS One with the title “Growth and trend analysis of area, production and yield of rice: A scenario of rice security in Bangladesh (Al Mamun M.A, Nihad SAI, Sarkar M.AR, Aziz M.A, Qayum M.A, Ahmed R, et al. 2021. Growth and trend analysis of area, production and yield of rice: A scenario of rice security in Bangladesh. PLoS ONE 16(12): e0261128. https://doi.org/10.1371/journal.pone.0261128”). The current manuscript is a subset of the other published paper. In the published paper, they used data for the period of 1969–70 to 2019–20 at the region level (old district level, there were 20 old districts which are currently named as region, the statistical data until early 2005 were available only at that level). In this paper they are using the current 64 districts (20 regions were divided into 64 districts) for which data are available for the period of 2006 to 2020 which they used in this manuscript). Many things are common between these two papers (such as maps of area, production, yield and adoption rate, statistical parameters, cluster analysis, etc.). There is no new message in this manuscript. So, what is the novel aspect of this paper? The paper is mostly the presentation of the secondary data in maps and charts though they are hardly readable. The quality of the figures and graphs are very poor as I mentioned in detail in my specific comments below. The discussion section of the paper is mostly not much relevant. Many significant factors such as rapid urbanization and industrialization on the agricultural lands, varieties of rice grown in different districts, problems of flood, drought and salinity in the coastal region are significant factors in future rice cultivation which should have been discussed. Please see also my specific comments given below.

Abstract and introduction and in other places in the manuscript: Spatial – temporal should be spatio-temporal

Lines 59-68: The word rice is used and the statistics for milled-rice is given for the world (787 tons) in lines 59-65. However, in line 66 the authors mentioned about the average paddy yield. Please clarify the average paddy yield. Is this milled-rice or yield at the farm after the harvest, un-milled rice? This always create confusion and it is not clear in the Bangladesh statistics whether the yield reported in for milled rice or un-milled paddy rice.

Line 74: Projected population growth to 189.9 – provide reference for that.

Line 76: Reference 9 given in the list does not have journal name.

Fig.1 What does inter-district alignment mean? The text in the figure particularly in the right on is not readable.

Line 193: Showing area in has up to 2 decimals is unnecessary. Please remove decimal places in all.

Lines 205 to 212: Please explain whether this is the yield of milled rice or unmilled rice.

Lines 217-218: Among the three seasons Aman season …., respectively – not clear. Please rephrase.

Line 229-230: “We examined season-wise assessments and their aggregated aspects to determine the impact of the leading season on national rice security in Bangladesh” - Not clear.

Fig.7: What does this % numbers mean? It is not clear in the text. What are the 10 principal components as mentioned in the fig caption?

Figs 8-10: What additional information do they provide? The information presented here can be easily presented in the Fig.2 district wise and then arranging them in ascending or descending order to show the cluster. In addition, there is spatial maps which shows the variations. So new information do they provide? Just another figures. The text in the figures are not readable at all.

Lines 314: What are the characteristics of different clusters? In Fig. 7, C1 to C5 were used as clusters? What C1 to C5 indicates in terms of area, yield, and production?

Lines 319: Elbow criterion – not readable so difficult to understand.

Lines: 321- 329: Please see my comments above.

Line 325: Poorest rice production – what does it mean? I think you wanted to mean lowest total production

Lines 337 – 365: Comments related to cluster analysis mentioned above applies here as well.

Lines 366 – 370: Increasing trend of what? I understand this could be the area but there is no mention of it in the text or in the figure 11. This should be clear that of the total cultivated area, xx% were on HYV cultivation. This is also country level data. Up to this point, the data was presented at the district level. Here, there is no mention of whether the data and the figure is at country level or district level.

Fig 11: Adoption (%) - % of what? Should be mentioned. Same applies to Fig. 12.

Fig 12: Spatial distribution of adoption rate presented here is very confusing. Adoption rate for each district for 14 years were classified into different groups. But this data should have two dimensions on is temporal and the other is spatial. How the authors presented these in these maps? What is the point of presenting the previous years? I think the authors can take the adoption rate of the last year and then classify them in different group (64 district into different group) which then can be presented in a map.

Line 395 -396: Districtwide adaptation scenario – is it adaptation or adoption? In the whole section of 3.5, adaptation was used in the text whether in the title it is adoption.

Fig 13: Again it is hard to understand the message from the figure caption or the axis headings. What does influence mean here? Increase or decrease of yield or production?

Lines 432-441: Mostly already described in the introduction (lines 78 to 86).

Lines 448-452: Repetition of results

Line 452-453: “The bulk of the aromatic rice is grown during the Aman season which may lead to high Aman rice production in Dinajpur” what about area. Aromatic rice nothing to do with the high production it is the area or the yield. There is no discussion on the yield differences and on varieties. In some areas, local varieties or HYV with lower yield are grown which has impact on overall production.

Lines 454- 462: Any references for this?

Lines 491 – 496: Any data or references?

Lines 499-502: Vertical and hydroponic farming for rice? Any references?

Lines 529-532: Repetitive.

Reviewer #2: The research topic “Spatiotemporal mapping of rice acreage and productivity growth in Bangladesh” is substantive and is within the scope of the journal. In general, the paper is thoroughly researched and well-written. By the way, I have some minor comments for the authors that would help to improve the quality of the manuscript.

Abstract:

1. Line # 35: Authors wrote “… performed a spatial-temporal mapping of the area, production, and yield….”. Is the word “area” sufficient or it is “cultivated/cultivation area”?

2. Line # 37 – 38: Replace “Results show that …” with “Results showed that….”.

3. Line # 41: “…the rice area in 19 districts, 11 districts, and 13 districts declined significantly”. The word “rice area” is not a suitable wording. Please replace it with “rice-cultivation-area” or “cultivation area”.

Introduction:

1. Line # 60-63: Please add the information “over 3.5 billion people are solely dependent upon rice for at least 20% of their daily required calories” from Introduction section of “Alam, M. J., Alamin, M., Sultana, M. H., Ahsan, M. A., Hossain, M. R., Islam, S. S., & Mollah, M. N. H. (2020). Bioinformatics studies on structures, functions and diversifications of rolling leaf related genes in rice (Oryza sativa L.). Plant Genetic Resources, 18(5), 382-395” with the sentence “With over half of the world's population depending on rice for their daily energy, and it supplies approximately 62% carbohydrate, 46% protein, 8% fat, 7% calcium, and 44% phosphorus of the recommended dietary allowance” and cite Alam et al. 2020.

2. Line # 65-66: Please add citation for the sentence “China is the leading rice-producing country, followed by India, Bangladesh, and Vietnam”.

3. Line #103-104: “This paper analyzed spatiotemporal” is not a standard wording. Please reshape the sentence as “In this study, we analyzed spatiotemporal data on cultivation area, and production and yield of rice to examine/investigate the trends and growth patterns from 2006-2007 to 2019-2020 in Bangladesh. Do not mention the data sources under Introduction section, rather write it under Materials and Methods section.

Materials and Methods:

1. Line # 120: Replace the word “area” with “cultivation area”. Also, this is applicable for whole body of this manuscript.

2. Please add a sentence to mention the level of significance considered for different statistical tests (e.g., normality test, t-test, etc.) under “Statistical analysis” subsection.

Results:

1. Line # 233: “… seasons in Bangladesh is illustrated in Fig 4”. Replace “is” with “are”.

2. Line # 293: Add unit of measurement for area, production and yield within brackets.

3. Line # 425: Replace “will” with “would” in line # 425.

Discussion:

1. Line # 484: Add an “and” before the phrase “declined over time”.

2. Line # 485: Replace “rice area” with “rice cultivation areas”.

3. Line # 555: Add “were” verb before the phrase “revealed in five ….” in line no. 555.

4. Line # 561: Replace “will” with “would” in line no. 561.

5. I would suggest to add practical implications of this research under the Discussion section.

6. In the Discussion section limitation of this study and future scope could be added.

Conclusion:

1. Line 545-550: First four sentences of Conclusion section are repetition that are already told in objective and method section. Rewrite these into one introductory sentence of Conclusion and write the conclusion with more focus on main finding and recommendation and policy making for government.

6. PLOS authors have the option to publish the peer review history of their article (what does this mean?). If published, this will include your full peer review and any attached files.

Reviewer #1: No

Reviewer #2: **Yes: **Dr. Md. Jahangir Alam

---

## [Author Response · Author response to Decision Letter 0]

21 Jul 2023

Manuscript number: PONE-D-23-09867

Title: Spatiotemporal mapping of rice acreage and productivity growth in Bangladesh

Journal: PLOS ONE

Thank you for the comments concerning our manuscript. We deeply appreciate your positive evaluation of our work. Those comments are valuable and very helpful. We have read through comments carefully and have made corrections. Please see below; all tasks and revisions taken are shown point-by-point.

Response to Academic Editor comments

Comments #1: Please ensure that your manuscript meets PLOS ONE's style requirements, including those for file naming.

Response to comment #1: We tried to meet the PLOS ONE’s style requirements.

Comments #2: Thank you for submitting the above manuscript to PLOS ONE. During our internal evaluation of the manuscript, we found significant text overlap between your submission and previous work in the [introduction, conclusion, etc.]. Please revise the manuscript to rephrase the duplicated text, cite your sources, and provide details as to how the current manuscript advances on previous work.

Response to comment #2: Thank you for this comment. We made an attempt to rephrase the manuscript.

Comments #3: We note that you have indicated that data from this study are available upon request. PLOS only allows data to be available upon request if there are legal or ethical restrictions on sharing data publicly. In your revised cover letter, please address the following prompts: a) If there are ethical or legal restrictions on sharing a de-identified data set, please explain them in detail (e.g., data contain potentially sensitive information, data are owned by a third-party organization, etc.) and who has imposed them (e.g., an ethics committee). Please also provide contact information for a data access committee, ethics committee, or other institutional body to which data requests may be sent. b) If there are no restrictions, please upload the minimal anonymized data set necessary to replicate your study findings as either Supporting Information files or to a stable, public repository and provide us with the relevant URLs, DOIs, or accession numbers. We will update your Data Availability statement on your behalf to reflect the information you provide.

Response to comment #3: We have uploaded the minimal anonymized dataset and subsequently adjusted the data availability statement accordingly.

Comments #4: We note that Figure 1, 4, 8, 9, 10, 11, 12, 14 in your submission contain [map/satellite] images which may be copyrighted. All PLOS content is published under the Creative Commons Attribution License (CC BY 4.0), which means that the manuscript, images, and Supporting Information files will be freely available online, and any third party is permitted to access, download, copy, distribute, and use these materials in any way, even commercially, with proper attribution. For these reasons, we cannot publish previously copyrighted maps or satellite images created using proprietary data, such as Google software (Google Maps, Street View, and Earth). For more information, see our copyright guidelines: http://journals.plos.org/plosone/s/licenses-and-copyright.

Response to comment #4: We obtained the necessary permission from the authority to use the shape file, and a copy of the permission letter has been attached.

Thank you once again for your precious comments and advice. Those comments are all valuable and very helpful for revising and improving our manuscript. We have revised the manuscript accordingly, and our point-by-point responses are presented above. We hope you are satisfied with our answers and the new data we have provided. Our deepest gratitude goes to you for your careful work and thoughtful suggestions that have helped improve this paper substantially.

Sincerely yours

All Authors’

Response to Reviewer’s comments

Reviewer #1:

Manuscript number: PONE-D-23-09867

Title: Spatiotemporal mapping of rice acreage and productivity growth in Bangladesh

Journal: PLOS ONE

Dear Reviewer,

Thank you for the comments concerning our manuscript. We deeply appreciate your posi-tive evaluation of our work. Those comments are valuable and very helpful. We have read through comments carefully and have made corrections. Please see below; all tasks and re-visions taken are shown point-by-point.

Comment #1: In this paper, the authors performed a spatio-temporal mapping of the area, production, and yield of rice from 2006-2007 to 2019-2020 using secondary data for dis-aggregating 64 districts in Bangladesh. They also looked at the adoption rate of high-yielding varieties of rice and did cluster analysis. The authors of this paper has also pub-lished a similar paper (same lead author) in PLOS One with the title “Growth and trend analysis of area, production and yield of rice: A scenario of rice security in Bangladesh (Al Mamun M.A, Nihad SAI, Sarkar M.AR, Aziz M.A, Qayum M.A, Ahmed R, et al. 2021. Growth and trend analysis of area, production and yield of rice: A scenario of rice security in Bangladesh. PLoS ONE 16(12): e0261128. https://doi.org/10.1371/journal.pone.0261128”). The current manuscript is a sub-set of the other published paper. In the published paper, they used data for the period of 1969–70 to 2019–20 at the region level (old district level, there were 20 old districts which are currently named as region, the statistical data until early 2005 were available only at that level). In this paper they are using the current 64 districts (20 regions were divided into 64 districts) for which data are available for the period of 2006 to 2020 which they used in this manuscript). Many things are common between these two papers (such as maps of area, production, yield and adoption rate, statistical parameters, cluster analysis, etc.). There is no new message in this manuscript. So, what is the novel aspect of this paper? 

Response to comment #1: Thank you for your comment. Yes, our research paper on the growth and trend analysis of rice area, production, and yield has been published in PLOS ONE. In the aforementioned publication, our primary focus was on the regional context, aiming to derive meaningful findings within the framework of 14 specific locations. How-ever, it is important to recognize that the rice-growing ecosystem in Bangladesh is charac-terized by a high degree of diversity, driven by factors such as geographical position, socio-economic conditions, and environmental variations. So, the findings of the previous study have limitations in terms of their statistical robustness and generalizability for the formula-tion of region-specific policies. To address these limitations, our current study takes a more disaggregated approach, analyzing data from 64 locations, in order to provide a comprehen-sive understanding of the dynamics of rice cultivation in Bangladesh. We believe that this shift in focus not only enhances the statistical validity of our research but also facilitates the formulation of more effective policies tailored to specific locations. However, we greatly appreciate your insightful comments, and as a result, we have made efforts to incorporate the research novelty into the introduction section of our manuscript. For a detailed elabora-tion of the novelty and distinctiveness of our current research, please refer to the table pro-vided below.

Table 1. Comparison of two manuscripts

Particulars Previous pa-per Current paper Remarks

Study area 14 agricultural regions 64 districts The rice growing ecosystem in Bangladesh exhibits significant diversity, making it essen-tial to initiate strategies and policy formula-tion at the grassroots level. In our previous study, we focused on 14 agricultural regions to gather information at the regional level. However, in our current study, we analyzed data at a disaggregated level, specifically ex-amining the 64 districts of Bangladesh. Ex-panding the study from 14 locations to 64 locations in Bangladesh was justified for sev-eral reasons. Firstly, by including a larger number of locations, we were able to capture a more comprehensive and representative pic-ture of rice cultivation practices and produc-tion across the country. This broader scope allowed us to identify regional variations, un-derstand diverse agricultural practices, and uncover unique challenges specific to each location. More specifically, by studying 64 districts, we gained a deeper understanding of the spatial distribution and dynamics of rice cultivation, which would have been limited in a study confined to only 14 regions. Second-ly, the inclusion of additional locations en-hanced the statistical robustness and generali-zability of our findings. With a larger sample study location, we were able to obtain more reliable estimates and draw more meaningful conclusions about rice cultivation patterns in Bangladesh. The increased geographical cov-erage provided a more accurate representation of the overall situation, minimizing potential biases that could arise from studying a small-er subset of regions. Furthermore, the expan-sion to 64 districts enabled us to better tailor our recommendations and policy implications to the specific needs and challenges faced by a wider range of regions, ensuring that the findings have practical relevance and applica-bility at a national level.

Data period 1969-70 to 2019-20 2006-07 to 2019-20 In order to align with the research objectives of our present study, we utilized 14 years of data (from 2006-07 to 2019-20) pertaining to rice area, production, and yield at the district level. For this purpose, we relied on the Bang-ladesh Bureau of Statistics (BBS) as the sole national statistical data repository in the country. The availability of data at the district level spans from 2006 onwards, while data prior to 2006 was only available at the region-al level. Therefore, in our previous study, we conducted a regional-level analysis using data ranging from 1969-70 to 2019-20. In that study, we utilized national-level data from 1969-70 to 2019-20 and regional-level data from 1984-85 to 2019-20, taking into consid-eration the availability of relevant data for each level of analysis. Thus, the choice of data period in our studies is primarily guided by the study objectives and the availability of data.

Statistical method Durbin-Watson test, Exponential growth

model, Cochrane-Orcutt itera-tion method, and k-means cluster analy-sis Augmented Dickey-Fuller test, Shapiro-Wilk normality test, Exponential growth model, Principal compo-nent analysis, and Hierarchical clus-ter analysis This study nature is a partial continuation of the previous one, employing a similar growth model to estimate growth and conduct trend analysis in both cases. However, there are notable differences in the methodology. In the previous study, we utilized the k-means cluster method, where only average data points were considered for clustering. In con-trast, for the present study, we applied robust multivariate time series clustering techniques, specifically dynamic time warping, which enabled us to use time series data points for more precise cluster identification. Addition-ally, we employed principal component analy-sis and optimal clustering methods to identify the most suitable clusters in our current study.

Study ob-jective Region map-ping via growth, trend and cluster analysis District mapping via growth, trend and cluster analy-sis

Nexus between adoption and pro-duction growth

District-level area and production forecasting The present study aims to provide a compre-hensive understanding of rice cultivation at the district level through growth, trend, and cluster analysis. Specifically, we focus on mapping the growth patterns and trends in rice production within each district, while also exploring the relationship between adop-tion rates of new technologies and the corre-sponding production growth. Additionally, we aim to forecast rice cultivation area and pro-duction at the district level. In contrast, our previous study focused on regional mapping, analyzing growth, trends, and clusters at a broader regional level rather than the district level. By shifting our focus to the district lev-el in the present study, we can provide more granular insights into the spatial dynamics of rice cultivation and better inform policy and decision-making processes.

Study con-text-1 - District-wise de-scriptive statistics of area, produc-tion and yield In the current paper, we employed descriptive statistics to offer a comprehensive overview of rice cultivation area, production, and yield at the district level over a 14-year period. This analysis enables us to summarize data, ex-plore patterns, facilitate comparisons, and support arguments. Moreover, it provides a crucial foundation for subsequent analysis and interpretation while enhancing the clarity and communicability of our research findings.

Study con-text-2 Production contribution by seasons and regions - While the current paper includes comprehen-sive descriptive statistics, it does not present the results regarding production contribution.

Study con-text-3 Periodic trend and growth assessment by regions and seasons Yearly trend and growth by dis-tricts and seasons In the previous study, our focus was on con-ducting trend and growth analyses within a regional context, limited by specific periods, which hindered our ability to uncover region-al heterogeneity. In contrast, in the present study, we expanded our examination to the district level, enabling us to conduct trend and growth analyses that capture a greater degree of regional heterogeneity.

Study con-text-4 Periodic adop-tion of modern varieties by regions and seasons The average adoption rate (%) of modern varie-ties and its tem-poral variation via GIS map In the earlier study, the adoption rate (%) of high-yielding variety (HYV) of rice was pre-sented across three distinct periods; however, the spatial distribution of this adoption was not analyzed. In contrast, our current paper offers a more comprehensive analysis by ex-amining the spatiotemporal variation of HYV adoption at the district level and representing it through GIS mapping. This approach pro-vides a valuable understanding of the geo-graphical patterns and temporal changes in HYV adoption, enhancing our knowledge of this important factor (technological adoption) in rice cultivation.

Study con-text-5 Clustering rice growing re-gions based on the production growth and HYV adoption Spatial clustering and classification of rice-growing districts based on cultivation area, production, and yield In our previous paper, we employed the k-means clustering technique to cluster 14 re-gions based on the growth rates of rice pro-duction and high-yielding variety adoption. However, this approach had limitations in identifying regional heterogeneity as it fo-cused on mean values and had a limited scope. Considering the highly diverse and regionally specific factors influencing the rice growing ecosystem in Bangladesh, we sought to address this limitation in our current study. To achieve more comprehensive and precise results, we utilized time series data of rice cultivation area, production, and yield at the district level as input parameters for cluster-ing. We employed robust multivariate cluster-ing techniques, specifically dynamic time warping (DTW), which takes into account technological advancements, farmers' prefer-ences, government initiatives, and environ-mental interactions associated with rice culti-vation. The cluster analysis resulted in group-ing similar districts, facilitating the creation of a rice zoning map and providing valuable insights for policy implications.

Study con-text-6 - Effect of modern varieties adoption on rice produc-tion by districts and seasons This section represents a new addition to our current paper, focusing on examining the im-pact of high-yielding variety (HYV) adoption on rice production in various districts of Bangladesh, assuming all other factors re-mained constant. Our findings revealed that a 10% increase in HYV adoption led to a vary-ing range of rice production increase, ranging from 0.04% to 5.8% across different seasons, with a few exceptions. In this analysis, HYV adoption served as a proxy for technological advancement. To the best of our knowledge, the extent to which this technological ad-vancement has influenced rice production in specific seasons and districts of Bangladesh has not been previously revealed. This study provides novel evidence in this regard, shed-ding light on the relationship between HYV adoption and rice production at the district level.

Study con-text-7 - District-level pro-jections of rice area and produc-tion changes an-ticipated by the year 2030 This section introduces a new addition to our current paper, with a focus on identifying future challenges that may hinder the achievement of doubling agricultural produc-tivity by 2030, specifically in sustaining rice production. While previous studies have of-fered forecasts for national-level rice area and production, our study takes a step further by providing projections for changes in district-level area and production from 2020 to 2030, visualized through GIS mapping. This ap-proach enables a comprehensive situational analysis, offering valuable insights into the required actions and strategies to effectively implement the 2030 global development agenda.

Comment #2: The paper is mostly the presentation of the secondary data in maps and charts though they are hardly readable. The quality of the figures and graphs are very poor as I mentioned in detail in my specific comments below. 

Response to comment #2: We appreciate your feedback regarding the presentation of the secondary data in maps and charts. We acknowledge that the figures and graphs may have been difficult to read due to their low resolution in the PDF copy. However, we would like to inform you that we have uploaded the figures in TIFF format, ensuring higher resolution and larger file sizes. We kindly request you to refer to the original figures for better clarity and readability.

Comment #3: The discussion section of the paper is mostly not much relevant. Many sig-nificant factors such as rapid urbanization and industrialization on the agricultural lands, varieties of rice grown in different districts, problems of flood, drought and salinity in the coastal region are significant factors in future rice cultivation which should have been dis-cussed. 

Response to comment #3: Thank you for your comment and valuable suggestions. We have taken your feedback into consideration and made revisions to the discussion section accord-ingly. We have included a more comprehensive discussion on significant factors such as rapid urbanization and industrialization affecting agricultural lands, the diversity of rice varieties grown in different districts, and the challenges posed by issues like flood, drought, and salinity in the coastal region. We believe these additions have enhanced the relevance and completeness of our discussion. We appreciate your input and the opportunity to im-prove the paper.

Comment #4: Abstract and introduction and in other places in the manuscript: Spatial – temporal should be spatio-temporal

Response to comment #4: Thank you for your comment. We have made the necessary changes throughout the manuscript by replacing the term "Spatial - temporal" with "Spatio-temporal" to ensure consistency and accuracy.

Comment #5: Lines 59-68: The word rice is used and the statistics for milled-rice is given for the world (787 tons) in lines 59-65. However, in line 66 the authors mentioned about the average paddy yield. Please clarify the average paddy yield. Is this milled-rice or yield at the farm after the harvest, un-milled rice? This always create confusion and it is not clear in the Bangladesh statistics whether the yield reported in for milled rice or un-milled paddy rice.

Response to comment #5: Thank you for bringing this to our attention. Here, paddy means un-milled rice. We have made the necessary clarification by adding the definition of "pad-dy" in the brackets.

Comment #6: Line 74: Projected population growth to 189.9 – provide reference for that.

Response to comment #6: Thank you for bringing this to our attention. We have added a reference to support this projection.

Comment #7: Line 76: Reference 9 given in the list does not have journal name.

Response to comment #7: Thank you for bringing this to our attention. We have made the necessary correction to the reference.

Comment #8: Fig.1 What does inter-district alignment mean? The text in the figure partic-ularly in the right on is not readable.

Response to comment #8: Thank you for your comment. In this context, inter-district alignment refers to the occurrence of agro-ecological zones (AEZs) overlapping in different districts. In Bangladesh, there are 30 AEZs distributed across 64 districts with overlapping zones. We acknowledge that the figures in the PDF copy may have had low resolution, mak-ing them difficult to read. However, we would like to inform you that we have uploaded the figures in TIFF format, ensuring higher resolution and larger file sizes. We kindly request you to refer to the original figures for better clarity and readability.

Comment #9: Line 193: Showing area in has up to 2 decimals is unnecessary. Please re-move decimal places in all.

Response to comment #9: Thank you for your suggestion. We have made the necessary re-visions to this section accordingly.

Comment #10: Lines 205 to 212: Please explain whether this is the yield of milled rice or unmilled rice.

Response to comment #10: Thank you for bringing this to our attention. In fact, this refers to milled rice, and we have already mentioned milled rice in the first line of the paragraph.

Comment #11: Lines 217-218: Among the three seasons Aman season …., respectively – not clear. Please rephrase.

Response to comment #11: Thank you for your suggestion. We have revised the sentence as per your recommendation. Please see the text, “The Aman season exhibited standard error of the mean (SEM) values of 2.17% for cultivation area, 3.87% for production, and 2.91% for yield”.

Comment #12: Line 229-230: “We examined season-wise assessments and their aggregated aspects to determine the impact of the leading season on national rice security in Bangla-desh” - Not clear.

Response to comment #12: Thank you for bringing this to our attention. We have revised the sentence as per your recommendation.

Comment #13: Fig.7: What does this % numbers mean? It is not clear in the text. What are the 10 principal components as mentioned in the fig caption?

Response to comment #13: Thank you for your comment. Regarding the principal compo-nent analysis (PCA), the percentages represent the amount of explained variation by each individual principal component (PC). The PCA technique allows us to specify the number of components to consider. In our analysis, we utilized 10 PCs to capture the variation in the multivariate data. These components serve as indicators of the explanatory power of the da-ta, with the first few components explaining the majority of the variation. Specifically, we observed that the first five PCs accounted for approximately 93.21%, 97.14%, and 96.02% of the total variances in the Aus, Aman, and Boro seasons, respectively.

Comment #14: Figs 8-10: What additional information do they provide? The information presented here can be easily presented in the Fig.2 district wise and then arranging them in ascending or descending order to show the cluster. In addition, there is spatial maps which shows the variations. So new information do they provide? Just another figures. The text in the figures are not readable at all.

Response to comment #14: Thank you for your comment. In our analysis, Figure 2 illus-trates the average performance (mean and standard error) of area, production, and yield for different seasons spanning from 2006-2007 to 2019-2020. It provides an overview of the trends without considering the clustering aspect. Our research focuses on three parameters: area, production, and yield. When dealing with a single parameter, clustering is relatively straightforward. However, when multiple parameters are involved, clustering becomes more challenging, requiring the application of statistical techniques.

For example, if we were to cluster based on ascending order of area using the mean value, the time variation effect would be disregarded, resulting in a regional area cluster. On the other hand, clustering based on production or yield could potentially result a different re-gional cluster. With multiple parameters, it becomes challenging to establish fixed regional clusters as the cluster of area may not be similar to the production cluster or yield cluster. To address this issue, we rely on multivariate statistical techniques to identify unique clus-ters based on the three parameters: area, production, and yield.

To achieve more comprehensive and precise results, we employed robust multivariate time series clustering techniques, specifically dynamic time warping, which enables us to utilize time series data points for more accurate cluster identification. Furthermore, we utilized principal component analysis and optimal clustering methods to identify the most suitable clusters for our study. By incorporating time series data of rice cultivation area, production, and yield at the district level as input parameters, we captured the heterogeneity in patterns across districts, attributable to geographical positions, technological dissemination, farmers' responsiveness, and climatic and edaphic factors. So, the cluster analysis resulted in the grouping of similar districts, facilitating the creation of a rice zoning map and providing valuable insights for policy implications. 

We acknowledge that the text in the figures may have been difficult to read in the PDF copy due to their low resolution. However, we have uploaded the figures in TIFF format, ensuring higher resolution and larger file sizes. We kindly request you to refer to the original figures for better clarity and readability.

Comment #15: Lines 314: What are the characteristics of different clusters? In Fig. 8, C1 to C5 were used as clusters? What C1 to C5 indicates in terms of area, yield, and production?

Response to comment #15: Thank you for your comments. In our study, we utilized long-term data of area, production, and yield as input parameters for identifying clusters using a multivariate clustering technique. The logic behind cluster analysis is to uncover hidden structures or patterns within a dataset and group similar observations together. More specif-ically it helps to identify groups or clusters within a dataset based on similarities or dissimi-larities between observations. In the case of the Aus season (Fig. 8), we identified five clus-ters out of the 64 districts. Each cluster represents a group of districts with similar charac-teristics in terms of cultivation area, production, and yield. Additionally, these clusters cap-ture regional heterogeneity and agro-ecological conditions, enabling effective policy formu-lation and implementation.

For example, the characteristics of Cluster 5 in Fig. 8 includes three districts (Bhola, My-mensingh, and Patuakhali) with similar dynamics in Aus rice cultivation. These districts are located in two coastal and one central region of Bangladesh. The rice cultivation area (high-est among the clusters) in this cluster ranges from 15,000 to 88,000 hectares, with more sta-ble rice production ranging from 40,000 to 170,000 tons and the highest mean rice produc-tion. The average yield is 1.8 tons per hectare, with a minimum of 1.2 tons per hectare and a maximum of 2.8 tons per hectare. Furthermore, six agro-ecological zones (AEZs) namely 8, 9, 13, 18, 28, and 29 have been identified in terms of Aus rice cultivation area, production, and yield within this cluster, indicating similar agro-ecological conditions in those regions for Aus cultivation.

Similarly, each cluster exhibits unique characteristics that provide valuable insights for pol-icymakers and researchers in formulating effective policies and taking immediate initia-tives to sustain rice production in Bangladesh.

Comment #16: Lines 319: Elbow criterion – not readable so difficult to understand.

Response to comment #16: Thank you for your comments. We have included a footnote to provide a clear explanation of the 'elbow criterion'. Please refer to the following text: “The elbow criterion is a method used in cluster analysis to determine the optimal number of clusters in a dataset. It involves plotting the variance explained by the clusters against the number of clusters. The plot typically resembles an arm, and the "elbow" or bend in the plot represents the point where the addition of more clusters does not significantly reduce the variance. This point is considered the optimal number of clusters for the given dataset. The elbow criterion helps in selecting a reasonable number of clusters that balance capturing meaningful patterns in the data while avoiding overfitting.”

Comment #17: Lines: 321- 329: Please see my comments above.

Response to comment #17: Thank you for your comments. We have addressed the im-portance of clustering in response to comment #14 and provided an explanation of the key characteristics of each cluster in response to comment #15. Please refer to the aforemen-tioned responses for more details.

Comment #18: Line 325: Poorest rice production – what does it mean? I think you wanted to mean lowest total production.

Response to comment #18: Thank you for bringing this to our attention. We have revised the sentence as per your recommendation.

Comment #19: Lines 337 – 365: Comments related to cluster analysis mentioned above applies here as well.

Response to comment #19: Thank you for your comments. We have addressed the im-portance of clustering in response to comment #14 and provided an explanation of the key characteristics of each cluster in response to comment #15. Please refer to the aforemen-tioned responses for more details.

Comment #20: Lines 366 – 370: Increasing trend of what? I understand this could be the area but there is no mention of it in the text or in the figure 11. This should be clear that of the total cultivated area, xx% were on HYV cultivation. This is also country level data. Up to this point, the data was presented at the district level. Here, there is no mention of wheth-er the data and the figure are at country level or district level.

Response to comment #20: Thank you for your comments. We have made the necessary re-visions to this section accordingly.

Comment #21: Fig 11: Adoption (%) - % of what? Should be mentioned. Same applies to Fig. 12.

Response to comment #21: Thank you for your comments. We have made the necessary re-visions to these figures accordingly.

Comment #22: Fig 12: Spatial distribution of adoption rate presented here is very confus-ing. Adoption rate for each district for 14 years were classified into different groups. But this data should have two dimensions on is temporal and the other is spatial. How the au-thors presented these in these maps? What is the point of presenting the previous years? I think the authors can take the adoption rate of the last year and then classify them in differ-ent group (64 district into different group) which then can be presented in a map.

Response to comment #22: Thank you for your comments. We sincerely apologize for any confusion caused by Fig 12 and appreciate the opportunity to clarify it. In Fig 12, we have utilized 14 years of high-yielding variety (HYV) adoption data from 64 districts. This data encompasses two dimensions: temporal and spatial. Our aim is to illustrate the spatial dis-tribution and temporal variations in the adoption rate of HYVs in rice cultivation across Bangladesh.

If we were solely interested in the spatial distribution, it would be sufficient to present the adoption rate of the current year. However, this approach would not capture the dynamic changes and fluctuations in HYV adoption over time. As varietal adoption is subject to dy-namic shifts, relying solely on the current year's data might overlook important temporal variations. Moreover, cultivating a particular variety in a single year does not guarantee its cultivation in the following year.

Given the two-dimensional nature of the data and our objective to track both spatiotemporal variations, a statistical technique is required. For the spatial distribution, we have employed the mean value of the 14-year data for each district, which is represented in the map as one of the legends "HYV adoption (%)." Additionally, we have included another legend in the form of "coefficient of variation (%)" to depict the temporal variability of HYV adoption in each district. The coefficient of variation (CV) was calculated using the 14-year data points for each district and visualized as circles on the map.

Therefore, to fulfill our research objective and ensure clarity, it is essential to consider the multi-year data. We have revised the title of Fig 12 accordingly. We hope that our explana-tion satisfactorily addresses your concerns.

Comment #23: Line 395 -396: Districtwide adaptation scenario – is it adaptation or adop-tion? In the whole section of 3.5, adaptation was used in the text whether in the title it is adoption.

Response to comment #23: Thank you for bringing this to our attention. We sincerely apol-ogize for the typo mistake. The correct term should be "adoption" instead of "adaptation" in section 3.5. We appreciate your feedback, and we have made the necessary revisions to en-sure consistency throughout the section.

Comment #24: Fig 13: Again it is hard to understand the message from the figure caption or the axis headings. What does influence mean here? Increase or decrease of yield or pro-duction?

Response to comment #24: Thank you for your comments. We deeply regret any confusion caused by Fig 13 and we are grateful for the chance to provide clarification. We have re-vised Section 3.5 and included the methodology in Section 2.4 to ensure a clear explanation and improve understanding. We kindly request you to review these sections, and we hope that the revisions will address your concerns adequately.

Comment #25: Lines 432-441: Mostly already described in the introduction (lines 78 to 86).

Response to comment #25: Thank you for your comments. We have taken note of your sug-gestion and have removed the redundant information as per your recommendation. We ap-preciate your feedback in streamlining the content of the manuscript.

Comment #26: Lines 448-452: Repetition of results

Response to comment #26: Thank you for your continued feedback. We have carefully con-sidered your comment and have made the appropriate changes to ensure that the content is concise and avoids repetition. We appreciate your diligence in reviewing our manuscript and helping us enhance its quality.

Comment #27: Line 452-453: “The bulk of the aromatic rice is grown during the Aman season which may lead to high Aman rice production in Dinajpur” what about area. Aro-matic rice nothing to do with the high production it is the area or the yield. There is no dis-cussion on the yield differences and on varieties. In some areas, local varieties or HYV with lower yield are grown which has impact on overall production.

Response to comment #27: Thank you for your comments. We have made the necessary re-visions to this section accordingly.

Comment #28: Lines 454- 462: Any references for this?

Response to comment #28: Thank you for pointing out the need for references in lines 454-462. We apologize for the oversight and have now included the relevant citations as per your suggestion.

Comment #29: Lines 491 – 496: Any data or references?

Response to comment #29: Thank you for pointing out the need for references in lines 491-496. We apologize for the oversight and have now included the relevant citations as per your suggestion. 

Comment #30: Lines 499-502: Vertical and hydroponic farming for rice? Any references?

Response to comment #30: Thank you for bringing this to our attention. We appreciate your comment and have removed the reference line to vertical and hydroponic farming for rice from the manuscript.

Comment #31: Lines 529-532: Repetitive.

Response to comment #31: Thank you for your comments. We have taken them into consid-eration and have removed the mentioned lines from the manuscript.

Thank you once again for your precious comments and advice. Those comments are all val-uable and very helpful for revising and improving our manuscript. We have revised the manuscript accordingly, and our point-by-point responses are presented above. We hope you are satisfied with our answers and the new data we have provided. Our deepest gratitude goes to you for your careful work and thoughtful suggestions that have helped improve this paper substantially.

Sincerely yours

All Authors’

Response to Reviewer’s comments

Reviewer #2:

Manuscript number: PONE-D-23-09867

Title: Spatiotemporal mapping of rice acreage and productivity growth in Bangladesh

Journal: PLOS ONE

Dear Reviewer,

Thank you for the comments concerning our manuscript. We deeply appreciate your posi-tive evaluation of our work. Those comments are valuable and very helpful. We have read through comments carefully and have made corrections. Please see below; all tasks and re-visions taken are shown point-by-point.

Comment #1: The research topic “Spatiotemporal mapping of rice acreage and productivity growth in Bangladesh” is substantive and is within the scope of the journal. In general, the paper is thoroughly researched and well-written. By the way, I have some minor comments for the authors that would help to improve the quality of the manuscript.

Response to comment #1: We would like to express our gratitude for your valuable and in-sightful suggestions and comments regarding the improvement of our manuscript. We firm-ly believe that they have significantly enhanced the scientific value of the manuscript, and we sincerely appreciate your contributions.

Abstract

Comment #2: Line # 35: Authors wrote “… performed a spatial-temporal mapping of the area, production, and yield….”. Is the word “area” sufficient or it is “cultivated/cultivation area”?

Response to comment #2: Thank you for the suggestion. The word “cultivation” has been added.

Comment #3: Line # 37 – 38: Replace “Results show that …” with “Results showed that….”.

Response to comment #3: The word 'show' has been replaced with 'showed'.

Comment #4: Line # 41: “…the rice area in 19 districts, 11 districts, and 13 districts de-clined significantly”. The word “rice area” is not a suitable wording. Please replace it with “rice-cultivation-area” or “cultivation area”.

Response to comment #4: The word "cultivation" has been added before the word "area".

Introduction

Comment #5: Line # 60-63: Please add the information “over 3.5 billion people are solely dependent upon rice for at least 20% of their daily required calories” from Introduction sec-tion of “Alam, M. J., Alamin, M., Sultana, M. H., Ahsan, M. A., Hossain, M. R., Islam, S. S., & Mollah, M. N. H. (2020). Bioinformatics studies on structures, functions and diversifica-tions of rolling leaf related genes in rice (Oryza sativa L.). Plant Genetic Resources, 18(5), 382-395” with the sentence “With over half of the world's population depending on rice for their daily energy, and it supplies approximately 62% carbohydrate, 46% protein, 8% fat, 7% calcium, and 44% phosphorus of the recommended dietary allowance” and cite Alam et al. 2020.

Response to comment #5: We appreciate your suggestion, and we have included the provid-ed information and incorporated the reference you suggested. Thank you for your valuable input.

Comment #6: Line # 65-66: Please add citation for the sentence “China is the leading rice-producing country, followed by India, Bangladesh, and Vietnam”.

Response to comment #6: We appreciate your observation. The citation for the sentence "China is the leading rice-producing country, followed by India, Bangladesh, and Vietnam" has been duly included.

Comment #7: Line #103-104: “This paper analyzed spatiotemporal” is not a standard word-ing. Please reshape the sentence as “In this study, we analyzed spatiotemporal data on culti-vation area, and production and yield of rice to examine/investigate the trends and growth patterns from 2006-2007 to 2019-2020 in Bangladesh. Do not mention the data sources un-der Introduction section, rather write it under Materials and Methods section.

Response to comment #7: Thank you for the suggestion. We have revised the sentence ac-cordingly.

Materials and Methods

Comment #8: Line # 120: Replace the word “area” with “cultivation area”. Also, this is ap-plicable for whole body of this manuscript.

Response to comment #8: We have made the necessary adjustment by replacing the term "area" with "cultivation area" throughout the entire manuscript, as suggested.

Comment #9: Please add a sentence to mention the level of significance considered for dif-ferent statistical tests (e.g., normality test, t-test, etc.) under “Statistical analysis” subsec-tion.

Response to comment #9: As per your recommendation, we have included the significance level of 5% in the analysis. Thank you for the suggestion.

Results

Comment #10: 1. Line # 233: “… seasons in Bangladesh is illustrated in Fig 4”. Replace “is” with “are”.

Response to comment #10: The word 'is' has been replaced with 'are'.

Comment #11: 2. Line # 293: Add unit of measurement for area, production and yield with-in brackets.

Response to comment #11: We have incorporated the percentage as a unit of measurement for the growth rate.

Comment #12: 3. Line # 425: Replace “will” with “would” in line # 425.

Response to comment #12: The word 'will' have been replaced with 'would'.

Discussion

Comment #13: Line # 484: Add an “and” before the phrase “declined over time”.

Response to comment #13: We have corrected it accordingly.

Comment #14: Line # 485: Replace “rice area” with “rice cultivation areas”.

Response to comment #14: We have corrected it accordingly.

Comment #15: Line # 555: Add “were” verb before the phrase “revealed in five ….” in line no. 555.

Response to comment #15: We have added ‘were’ verb before the phrase “revealed in five ….” accordingly.

Comment #16: Line # 561: Replace “will” with “would” in line no. 561.

Response to comment #16: The word 'will' have been replaced with 'would'.

Comment #17: I would suggest to add practical implications of this research under the Dis-cussion section.

Response to comment #17: Thank you for the suggestion. We have made an effort to incor-porate the practical implications of this research into the discussion section. Please see the text, “The research findings have significant practical implications for informing and guid-ing future agricultural practices and policies in rice-intensive areas of Bangladesh. The study provides valuable insights into the dynamic changes in the area and production of rice cultivation at a more disaggregated level, representing diverse ecosystems and man-agement constraints. Policymakers can utilize this information to develop targeted strate-gies and interventions aimed at enhancing rice production in specific regions. For instance, regions with favorable environmental conditions and fertile soil content can be encouraged to prioritize expanding rice cultivation. Efforts can be directed towards providing farmers in these regions with access to modern high-yielding varieties and advanced agricultural technologies to improve productivity. Conversely, areas facing challenges such as salinity or hilly terrain require tailored approaches to overcome these limitations, such as the de-velopment of stress-tolerant rice varieties and the implementation of appropriate irrigation systems. Additionally, identifying regions where rice production is declining, or farmers are shifting to alternative crops can guide initiatives aimed at revitalizing rice cultivation through improved irrigation facilities, training programs, and the development of short-duration varieties. The study highlights the areas that require closer attention to overcome challenges and enhance productivity, ultimately contributing to achieving Sustainable De-velopment Goal 2.3 of doubling agricultural productivity and income for smallholder farm-ers by 2030. In summary, these practical implications can contribute to sustainable agricul-tural practices, enhance food security, and improve the livelihoods of farmers in Bangla-desh.”

Comment #18: In the Discussion section limitation of this study and future scope could be added.

Response to comment #18: Thank you for the suggestion. We have made an effort to incor-porate the limitation and future research scope into the discussion section. Please see the text, “Despite the valuable insights provided by this study, there are certain limitations that should be acknowledged. First, this research focused primarily on the quantitative analysis of rice cultivation area and production trends, and did not delve into the underlying socio-economic factors that may influence these trends. Additionally, this study relied on second-ary data sources, which may be subject to limitations such as data accuracy and availabil-ity. Conducting primary data collection through field surveys and interviews could enhance the accuracy and reliability of the findings. Furthermore, the study's scope was limited to a specific timeframe (2006-2019), and it is important to recognize that future changes in cli-mate, technology, and agricultural practices could impact rice cultivation in unforeseen ways.

Building on the findings and limitations of this study, there are several areas for future re-search that can further contribute to the understanding of rice cultivation in Bangladesh. Firstly, investigating the socio-economic factors that influence farmers' decision-making processes regarding rice cultivation, including factors such as market conditions, govern-ment policies, and farmers' preferences, would provide valuable insights into the drivers of rice production. Secondly, exploring the impact of climate change on rice cultivation, in-cluding the effects of rising temperatures, changing rainfall patterns, and increased occur-rences of extreme weather events, is crucial for developing climate-resilient agricultural strategies. Additionally, examining the adoption and effectiveness of specific interventions aimed at improving rice productivity, such as the dissemination of high-yielding varieties, the implementation of irrigation systems, and the provision of training and extension ser-vices, would help in identifying best practices and areas for improvement. Finally, integrat-ing remote sensing and geospatial analysis techniques can enhance the accuracy and time-liness of monitoring rice cultivation dynamics, allowing for more precise and up-to-date assessments of cultivation area and production trends.”

Conclusion

Comment #19: Line 545-550: First four sentences of Conclusion section are repetition that are already told in objective and method section. Rewrite these into one introductory sen-tence of Conclusion and write the conclusion with more focus on main finding and recom-mendation and policy making for government. 

Response to comment #19: We greatly appreciate your valuable suggestion, which has prompted us to thoroughly revise the conclusion section and incorporate essential policy recommendations. 

Thank you once again for your precious comments and advice. Those comments are all val-uable and very helpful for revising and improving our manuscript. We have revised the manuscript accordingly, and our point-by-point responses are presented above. We hope you are satisfied with our answers and the new data we have provided. Our deepest gratitude goes to you for your careful work and thoughtful suggestions that have helped improve this paper substantially.

Sincerely yours

All Authors’

---

## [Decision Letter · Decision Letter 1]

9 Jan 2024

PONE-D-23-09867R1

Spatiotemporal mapping of rice acreage and productivity growth in Bangladesh

PLOS ONE

Dear Dr. Al Mamun,

Thank you for submitting your manuscript to PLOS ONE. After careful consideration, we have decided that your manuscript does not meet our criteria for publication and must therefore be rejected.

I am sorry that we cannot be more positive on this occasion, but hope that you appreciate the reasons for this decision.

Kind regards,

Abul Khayer Mohammad Golam Sarwar

Academic Editor

PLOS ONE

Additional Editor Comments:

fails to show any novelty even after the major revision. Please see the reviewer's comment.

Reviewers' comments:

Reviewer's Responses to Questions

**Comments to the Author**

1. If the authors have adequately addressed your comments raised in a previous round of review and you feel that this manuscript is now acceptable for publication, you may indicate that here to bypass the “Comments to the Author” section, enter your conflict of interest statement in the “Confidential to Editor” section, and submit your "Accept" recommendation.

Reviewer #3: All comments have been addressed

Reviewer #4: All comments have been addressed

2. Is the manuscript technically sound, and do the data support the conclusions?

Reviewer #3: Yes

Reviewer #4: No

3. Has the statistical analysis been performed appropriately and rigorously? 

Reviewer #3: Yes

Reviewer #4: No

4. Have the authors made all data underlying the findings in their manuscript fully available?

Reviewer #3: Yes

Reviewer #4: No

5. Is the manuscript presented in an intelligible fashion and written in standard English?

Reviewer #3: Yes

Reviewer #4: No

6. Review Comments to the Author

Reviewer #3: The authors have fairly addressed all the comments. In this regard, this article can be published in this journal. Good luck!

Reviewer #4: Sorry to say that there is nothing new in the manuscript. A lot of similar studies have been performed. These cited studies are missing in this paper. I suggest rejecting this paper due to lack of novelty.

7. PLOS authors have the option to publish the peer review history of their article (what does this mean?). If published, this will include your full peer review and any attached files.

Reviewer #3: No

Reviewer #4: No

- - - - -

---

## [Author Response · Author response to Decision Letter 1]

30 Jan 2024

Manuscript number: PONE-D-23-09867

Title: Spatiotemporal mapping of rice acreage and productivity growth in Bangladesh

Journal: PLOS ONE

Thank you for the comments concerning our manuscript. We deeply appreciate your positive evaluation of our work. Those comments are valuable and very helpful. We have read through comments carefully and have made corrections. Please see below; all tasks and revisions taken are shown point-by-point.

Response to Academic Editor comments

Comments #1: Please ensure that your manuscript meets PLOS ONE's style requirements, including those for file naming.

Response to comment #1: We tried to meet the PLOS ONE’s style requirements.

Comments #2: Thank you for submitting the above manuscript to PLOS ONE. During our internal evaluation of the manuscript, we found significant text overlap between your submission and previous work in the [introduction, conclusion, etc.]. Please revise the manuscript to rephrase the duplicated text, cite your sources, and provide details as to how the current manuscript advances on previous work.

Response to comment #2: Thank you for this comment. We made an attempt to rephrase the manuscript.

Comments #3: We note that you have indicated that data from this study are available upon request. PLOS only allows data to be available upon request if there are legal or ethical restrictions on sharing data publicly. In your revised cover letter, please address the following prompts: a) If there are ethical or legal restrictions on sharing a de-identified data set, please explain them in detail (e.g., data contain potentially sensitive information, data are owned by a third-party organization, etc.) and who has imposed them (e.g., an ethics committee). Please also provide contact information for a data access committee, ethics committee, or other institutional body to which data requests may be sent. b) If there are no restrictions, please upload the minimal anonymized data set necessary to replicate your study findings as either Supporting Information files or to a stable, public repository and provide us with the relevant URLs, DOIs, or accession numbers. We will update your Data Availability statement on your behalf to reflect the information you provide.

Response to comment #3: We have uploaded the minimal anonymized dataset and subsequently adjusted the data availability statement accordingly.

Comments #4: We note that Figure 1, 4, 8, 9, 10, 11, 12, 14 in your submission contain [map/satellite] images which may be copyrighted. All PLOS content is published under the Creative Commons Attribution License (CC BY 4.0), which means that the manuscript, images, and Supporting Information files will be freely available online, and any third party is permitted to access, download, copy, distribute, and use these materials in any way, even commercially, with proper attribution. For these reasons, we cannot publish previously copyrighted maps or satellite images created using proprietary data, such as Google software (Google Maps, Street View, and Earth). For more information, see our copyright guidelines: http://journals.plos.org/plosone/s/licenses-and-copyright.

Response to comment #4: We obtained the necessary permission from the authority to use the shape file, and a copy of the permission letter has been attached.

Thank you once again for your precious comments and advice. Those comments are all valuable and very helpful for revising and improving our manuscript. We have revised the manuscript accordingly, and our point-by-point responses are presented above. We hope you are satisfied with our answers and the new data we have provided. Our deepest gratitude goes to you for your careful work and thoughtful suggestions that have helped improve this paper substantially.

Sincerely yours

All Authors’

Manuscript number: PONE-D-23-09867R1

Title: Spatiotemporal mapping of rice acreage and productivity growth in Bangladesh

Journal: PLOS ONE

Thank you for the comments concerning our manuscript. We deeply appreciate your positive evaluation of our work. Those comments are valuable and very helpful. We have read through comments carefully and have made corrections. Please see below; all tasks and revisions taken are shown point-by-point.

Response to Academic Editor comments

Comments #1: Fails to show any novelty even after the major revision. Please see the reviewer's comment.

Response to comment #1: Thank you for your thorough review and valuable feedback. While we respect your perspective, we assert that our manuscript offers unique contributions not fully covered in previous studies. Recognizing the significance of novelty in research, we have carefully revised the introduction section to highlight the distinctive aspects of our work. Additionally, we have incorporated relevant citations to address updated references, aiming to enhance the overall value of our manuscript. We hope that upon reevaluation, you will find our revised work to be a meaningful addition to the existing literature.

Regarding your comment on our research paper's publication in PLOS ONE, we appreciate your acknowledgment. In our previous study, we focused on the regional context within 14 specific locations, but we recognize the inherent diversity in Bangladesh's rice-growing ecosystem. To address this, our current study takes a more detailed approach, analyzing data from 64 locations for a comprehensive understanding of rice cultivation dynamics. This shift enhances the statistical validity of our research and facilitates the formulation of more effective, location-specific policies. We have incorporated these refinements, including the research novelty, into the introduction section based on your insightful comments. For detailed insights into the novelty of our research, kindly refer to response of Comment #1 from Reviewer #1.

Our research findings have significant practical implications for informing and guiding future agricultural practices and policies in rice-intensive areas of Bangladesh. The study provides valuable insights into the dynamic changes in the area and production of rice cultivation at a more disaggregated level, representing diverse ecosystems and management constraints. Policymakers can utilize this information to develop targeted strategies and interventions aimed at enhancing rice production in specific regions. For instance, regions with favorable environmental conditions and fertile soil content can be encouraged to prioritize expanding rice cultivation. Efforts can be directed towards providing farmers in these regions with access to modern high-yielding varieties and advanced agricultural technologies to improve productivity. Conversely, areas facing challenges such as salinity or hilly terrain require tailored approaches to overcome these limitations, such as the development of stress-tolerant rice varieties and the implementation of appropriate irrigation systems. Additionally, identifying regions where rice production is declining, or farmers are shifting to alternative crops can guide initiatives aimed at revitalizing rice cultivation through improved irrigation facilities, training programs, and the development of short-duration varieties. The study highlights the areas that require closer attention to overcome challenges and enhance productivity, ultimately contributing to achieving Sustainable Development Goal 2.3 of doubling agricultural productivity and income for smallholder farmers by 2030. In summary, these practical implications can contribute to sustainable agricultural practices, enhance food security, and improve the livelihoods of farmers in Bangladesh.

We express our sincere gratitude for your valuable comments and advice, which have significantly contributed to the improvement of our manuscript. Your thorough review has been instrumental in shaping our revisions, and we trust that our responses and the additional data provided meet your expectations. Once again, thank you for your careful work and thoughtful suggestions.

Sincerely yours

All Authors’

Manuscript number: PONE-D-23-09867

Title: Spatiotemporal mapping of rice acreage and productivity growth in Bangladesh

Journal: PLOS ONE

Dear Reviewer,

Thank you for the comments concerning our manuscript. We deeply appreciate your posi-tive evaluation of our work. Those comments are valuable and very helpful. We have read through comments carefully and have made corrections. Please see below; all tasks and re-visions taken are shown point-by-point.

Response to Reviewer’s comments

Reviewer #1:

Comment #1: In this paper, the authors performed a spatio-temporal mapping of the area, production, and yield of rice from 2006-2007 to 2019-2020 using secondary data for dis-aggregating 64 districts in Bangladesh. They also looked at the adoption rate of high-yielding varieties of rice and did cluster analysis. The authors of this paper has also pub-lished a similar paper (same lead author) in PLOS One with the title “Growth and trend analysis of area, production and yield of rice: A scenario of rice security in Bangladesh (Al Mamun M.A, Nihad SAI, Sarkar M.AR, Aziz M.A, Qayum M.A, Ahmed R, et al. 2021. Growth and trend analysis of area, production and yield of rice: A scenario of rice security in Bangladesh. PLoS ONE 16(12): e0261128. https://doi.org/10.1371/journal.pone.0261128”). The current manuscript is a sub-set of the other published paper. In the published paper, they used data for the period of 1969–70 to 2019–20 at the region level (old district level, there were 20 old districts which are currently named as region, the statistical data until early 2005 were available only at that level). In this paper they are using the current 64 districts (20 regions were divided into 64 districts) for which data are available for the period of 2006 to 2020 which they used in this manuscript). Many things are common between these two papers (such as maps of area, production, yield and adoption rate, statistical parameters, cluster analysis, etc.). There is no new message in this manuscript. So, what is the novel aspect of this paper? 

Response to comment #1: Thank you for your comment. Yes, our research paper on the growth and trend analysis of rice area, production, and yield has been published in PLOS ONE. In the aforementioned publication, our primary focus was on the regional context, aiming to derive meaningful findings within the framework of 14 specific locations. How-ever, it is important to recognize that the rice-growing ecosystem in Bangladesh is charac-terized by a high degree of diversity, driven by factors such as geographical position, socio-economic conditions, and environmental variations. So, the findings of the previous study have limitations in terms of their statistical robustness and generalizability for the formula-tion of region-specific policies. To address these limitations, our current study takes a more disaggregated approach, analyzing data from 64 locations, in order to provide a comprehen-sive understanding of the dynamics of rice cultivation in Bangladesh. We believe that this shift in focus not only enhances the statistical validity of our research but also facilitates the formulation of more effective policies tailored to specific locations. However, we greatly appreciate your insightful comments, and as a result, we have made efforts to incorporate the research novelty into the introduction section of our manuscript. For a detailed elabora-tion of the novelty and distinctiveness of our current research, please refer to the table pro-vided below.

Table 1. Comparison of two manuscripts

Particulars Previous pa-per Current paper Remarks

Study area 14 agricultural regions 64 districts The rice growing ecosystem in Bangladesh exhibits significant diversity, making it essen-tial to initiate strategies and policy formula-tion at the grassroots level. In our previous study, we focused on 14 agricultural regions to gather information at the regional level. However, in our current study, we analyzed data at a disaggregated level, specifically ex-amining the 64 districts of Bangladesh. Ex-panding the study from 14 locations to 64 locations in Bangladesh was justified for sev-eral reasons. Firstly, by including a larger number of locations, we were able to capture a more comprehensive and representative pic-ture of rice cultivation practices and produc-tion across the country. This broader scope allowed us to identify regional variations, un-derstand diverse agricultural practices, and uncover unique challenges specific to each location. More specifically, by studying 64 districts, we gained a deeper understanding of the spatial distribution and dynamics of rice cultivation, which would have been limited in a study confined to only 14 regions. Second-ly, the inclusion of additional locations en-hanced the statistical robustness and generali-zability of our findings. With a larger sample study location, we were able to obtain more reliable estimates and draw more meaningful conclusions about rice cultivation patterns in Bangladesh. The increased geographical cov-erage provided a more accurate representation of the overall situation, minimizing potential biases that could arise from studying a small-er subset of regions. Furthermore, the expan-sion to 64 districts enabled us to better tailor our recommendations and policy implications to the specific needs and challenges faced by a wider range of regions, ensuring that the findings have practical relevance and applica-bility at a national level.

Data period 1969-70 to 2019-20 2006-07 to 2019-20 In order to align with the research objectives of our present study, we utilized 14 years of data (from 2006-07 to 2019-20) pertaining to rice area, production, and yield at the district level. For this purpose, we relied on the Bang-ladesh Bureau of Statistics (BBS) as the sole national statistical data repository in the country. The availability of data at the district level spans from 2006 onwards, while data prior to 2006 was only available at the region-al level. Therefore, in our previous study, we conducted a regional-level analysis using data ranging from 1969-70 to 2019-20. In that study, we utilized national-level data from 1969-70 to 2019-20 and regional-level data from 1984-85 to 2019-20, taking into consid-eration the availability of relevant data for each level of analysis. Thus, the choice of data period in our studies is primarily guided by the study objectives and the availability of data.

Statistical method Durbin-Watson test, Exponential growth

model, Cochrane-Orcutt itera-tion method, and k-means cluster analy-sis Augmented Dickey-Fuller test, Shapiro-Wilk normality test, Exponential growth model, Principal compo-nent analysis, and Hierarchical clus-ter analysis This study nature is a partial continuation of the previous one, employing a similar growth model to estimate growth and conduct trend analysis in both cases. However, there are notable differences in the methodology. In the previous study, we utilized the k-means cluster method, where only average data points were considered for clustering. In con-trast, for the present study, we applied robust multivariate time series clustering techniques, specifically dynamic time warping, which enabled us to use time series data points for more precise cluster identification. Addition-ally, we employed principal component analy-sis and optimal clustering methods to identify the most suitable clusters in our current study.

Study ob-jective Region map-ping via growth, trend and cluster analysis District mapping via growth, trend and cluster analy-sis

Nexus between adoption and pro-duction growth

District-level area and production forecasting The present study aims to provide a compre-hensive understanding of rice cultivation at the district level through growth, trend, and cluster analysis. Specifically, we focus on mapping the growth patterns and trends in rice production within each district, while also exploring the relationship between adop-tion rates of new technologies and the corre-sponding production growth. Additionally, we aim to forecast rice cultivation area and pro-duction at the district level. In contrast, our previous study focused on regional mapping, analyzing growth, trends, and clusters at a broader regional level rather than the district level. By shifting our focus to the district lev-el in the present study, we can provide more granular insights into the spatial dynamics of rice cultivation and better inform policy and decision-making processes.

Study con-text-1 - District-wise de-scriptive statistics of area, produc-tion and yield In the current paper, we employed descriptive statistics to offer a comprehensive overview of rice cultivation area, production, and yield at the district level over a 14-year period. This analysis enables us to summarize data, ex-plore patterns, facilitate comparisons, and support arguments. Moreover, it provides a crucial foundation for subsequent analysis and interpretation while enhancing the clarity and communicability of our research findings.

Study con-text-2 Production contribution by seasons and regions - While the current paper includes comprehen-sive descriptive statistics, it does not present the results regarding production contribution.

Study con-text-3 Periodic trend and growth assessment by regions and seasons Yearly trend and growth by dis-tricts and seasons In the previous study, our focus was on con-ducting trend and growth analyses within a regional context, limited by specific periods, which hindered our ability to uncover region-al heterogeneity. In contrast, in the present study, we expanded our examination to the district level, enabling us to conduct trend and growth analyses that capture a greater degree of regional heterogeneity.

Study con-text-4 Periodic adop-tion of modern varieties by regions and seasons The average adoption rate (%) of modern varie-ties and its tem-poral variation via GIS map In the earlier study, the adoption rate (%) of high-yielding variety (HYV) of rice was pre-sented across three distinct periods; however, the spatial distribution of this adoption was not analyzed. In contrast, our current paper offers a more comprehensive analysis by ex-amining the spatiotemporal variation of HYV adoption at the district level and representing it through GIS mapping. This approach pro-vides a valuable understanding of the geo-graphical patterns and temporal changes in HYV adoption, enhancing our knowledge of this important factor (technological adoption) in rice cultivation.

Study con-text-5 Clustering rice growing re-gions based on the production growth and HYV adoption Spatial clustering and classification of rice-growing districts based on cultivation area, production, and yield In our previous paper, we employed the k-means clustering technique to cluster 14 re-gions based on the growth rates of rice pro-duction and high-yielding variety adoption. However, this approach had limitations in identifying regional heterogeneity as it fo-cused on mean values and had a limited scope. Considering the highly diverse and regionally specific factors influencing the rice growing ecosystem in Bangladesh, we sought to address this limitation in our current study. To achieve more comprehensive and precise results, we utilized time series data of rice cultivation area, production, and yield at the district level as input parameters for cluster-ing. We employed robust multivariate cluster-ing techniques, specifically dynamic time warping (DTW), which takes into account technological advancements, farmers' prefer-ences, government initiatives, and environ-mental interactions associated with rice culti-vation. The cluster analysis resulted in group-ing similar districts, facilitating the creation of a rice zoning map and providing valuable insights for policy implications.

Study con-text-6 - Effect of modern varieties adoption on rice produc-tion by districts and seasons This section represents a new addition to our current paper, focusing on examining the im-pact of high-yielding variety (HYV) adoption on rice production in various districts of Bangladesh, assuming all other factors re-mained constant. Our findings revealed that a 10% increase in HYV adoption led to a vary-ing range of rice production increase, ranging from 0.04% to 5.8% across different seasons, with a few exceptions. In this analysis, HYV adoption served as a proxy for technological advancement. To the best of our knowledge, the extent to which this technological ad-vancement has influenced rice production in specific seasons and districts of Bangladesh has not been previously revealed. This study provides novel evidence in this regard, shed-ding light on the relationship between HYV adoption and rice production at the district level.

Study con-text-7 - District-level pro-jections of rice area and produc-tion changes an-ticipated by the year 2030 This section introduces a new addition to our current paper, with a focus on identifying future challenges that may hinder the achievement of doubling agricultural produc-tivity by 2030, specifically in sustaining rice production. While previous studies have of-fered forecasts for national-level rice area and production, our study takes a step further by providing projections for changes in district-level area and production from 2020 to 2030, visualized through GIS mapping. This ap-proach enables a comprehensive situational analysis, offering valuable insights into the required actions and strategies to effectively implement the 2030 global development agenda.

Comment #2: The paper is mostly the presentation of the secondary data in maps and charts though they are hardly readable. The quality of the figures and graphs are very poor as I mentioned in detail in my specific comments below. 

Response to comment #2: We appreciate your feedback regarding the presentation of the secondary data in maps and charts. We acknowledge that the figures and graphs may have been difficult to read due to their low resolution in the PDF copy. However, we would like to inform you that we have uploaded the figures in TIFF format, ensuring higher resolution and larger file sizes. We kindly request you to refer to the original figures for better clarity and readability.

Comment #3: The discussion section of the paper is mostly not much relevant. Many sig-nificant factors such as rapid urbanization and industrialization on the agricultural lands, varieties of rice grown in different districts, problems of flood, drought and salinity in the coastal region are significant factors in future rice cultivation which should have been dis-cussed. 

Response to comment #3: Thank you for your comment and valuable suggestions. We have taken your feedback into consideration and made revisions to the discussion section accord-ingly. We have included a more comprehensive discussion on significant factors such as rapid urbanization and industrialization affecting agricultural lands, the diversity of rice varieties grown in different districts, and the challenges posed by issues like flood, drought, and salinity in the coastal region. We believe these additions have enhanced the relevance and completeness of our discussion. We appreciate your input and the opportunity to im-prove the paper.

Comment #4: Abstract and introduction and in other places in the manuscript: Spatial – temporal should be spatio-temporal

Response to comment #4: Thank you for your comment. We have made the necessary changes throughout the manuscript by replacing the term "Spatial - temporal" with "Spatio-temporal" to ensure consistency and accuracy.

Comment #5: Lines 59-68: The word rice is used and the statistics for milled-rice is given for the world (787 tons) in lines 59-65. However, in line 66 the authors mentioned about the average paddy yield. Please clarify the average paddy yield. Is this milled-rice or yield at the farm after the harvest, un-milled rice? This always create confusion and it is not clear in the Bangladesh statistics whether the yield reported in for milled rice or un-milled paddy rice.

Response to comment #5: Thank you for bringing this to our attention. Here, paddy means un-milled rice. We have made the necessary clarification by adding the definition of "pad-dy" in the brackets.

Comment #6: Line 74: Projected population growth to 189.9 – provide reference for that.

Response to comment #6: Thank you for bringing this to our attention. We have added a reference to support this projection.

Comment #7: Line 76: Reference 9 given in the list does not have journal name.

Response to comment #7: Thank you for bringing this to our attention. We have made the necessary correction to the reference.

Comment #8: Fig.1 What does inter-district alignment mean? The text in the figure partic-ularly in the right on is not readable.

Response to comment #8: Thank you for your comment. In this context, inter-district alignment refers to the occurrence of agro-ecological zones (AEZs) overlapping in different districts. In Bangladesh, there are 30 AEZs distributed across 64 districts with overlapping zones. We acknowledge that the figures in the PDF copy may have had low resolution, mak-ing them difficult to read. However, we would like to inform you that we have uploaded the figures in TIFF format, ensuring higher resolution and larger file sizes. We kindly request you to refer to the original figures for better clarity and readability.

Comment #9: Line 193: Showing area in has up to 2 decimals is unnecessary. Please re-move decimal places in all.

Response to comment #9: Thank you for your suggestion. We have made the necessary re-visions to this section accordingly.

Comment #10: Lines 205 to 212: Please explain whether this is the yield of milled rice or unmilled rice.

Response to comment #10: Thank you for bringing this to our attention. In fact, this refers to milled rice, and we have already mentioned milled rice in the first line of the paragraph.

Comment #11: Lines 217-218: Among the three seasons Aman season …., respectively – not clear. Please rephrase.

Response to comment #11: Thank you for your suggestion. We have revised the sentence as per your recommendation. Please see the text, “The Aman season exhibited standard error of the mean (SEM) values of 2.17% for cultivation area, 3.87% for production, and 2.91% for yield”.

Comment #12: Line 229-230: “We examined season-wise assessments and their aggregated aspects to determine the impact of the leading season on national rice security in Bangla-desh” - Not clear.

Response to comment #12: Thank you for bringing this to our attention. We have revised the sentence as per your recommendation.

Comment #13: Fig.7: What does this % numbers mean? It is not clear in the text. What are the 10 principal components as mentioned in the fig caption?

Response to comment #13: Thank you for your comment. Regarding the principal compo-nent analysis (PCA), the percentages represent the amount of explained variation by each individual principal component (PC). The PCA technique allows us to specify the number of components to consider. In our analysis, we utilized 10 PCs to capture the variation in the multivariate data. These components serve as indicators of the explanatory power of the da-ta, with the first few components explaining the majority of the variation. Specifically, we observed that the first five PCs accounted for approximately 93.21%, 97.14%, and 96.02% of the total variances in the Aus, Aman, and Boro seasons, respectively.

Comment #14: Figs 8-10: What additional information do they provide? The information presented here can be easily presented in the Fig.2 district wise and then arranging them in ascending or descending order to show the cluster. In addition, there is spatial maps which shows the variations. So new information do they provide? Just another figures. The text in the figures are not readable at all.

Response to comment #14: Thank you for your comment. In our analysis, Figure 2 illus-trates the average performance (mean and standard error) of area, production, and yield for different seasons spanning from 2006-2007 to 2019-2020. It provides an overview of the trends without considering the clustering aspect. Our research focuses on three parameters: area, production, and yield. When dealing with a single parameter, clustering is relatively straightforward. However, when multiple parameters are involved, clustering becomes more challenging, requiring the application of statistical techniques.

For example, if we were to cluster based on ascending order of area using the mean value, the time variation effect would be disregarded, resulting in a regional area cluster. On the other hand, clustering based on production or yield could potentially result a different re-gional cluster. With multiple parameters, it becomes challenging to establish fixed regional clusters as the cluster of area may not be similar to the production cluster or yield cluster. To address this issue, we rely on multivariate statistical techniques to identify unique clus-ters based on the three parameters: area, production, and yield.

To achieve more comprehensive and precise results, we employed robust multivariate time series clustering techniques, specifically dynamic time warping, which enables us to utilize time series data points for more accurate cluster identification. Furthermore, we utilized principal component analysis and optimal clustering methods to identify the most suitable clusters for our study. By incorporating time series data of rice cultivation area, production, and yield at the district level as input parameters, we captured the heterogeneity in patterns across districts, attributable to geographical positions, technological dissemination, farmers' responsiveness, and climatic and edaphic factors. So, the cluster analysis resulted in the grouping of similar districts, facilitating the creation of a rice zoning map and providing valuable insights for policy implications. 

We acknowledge that the text in the figures may have been difficult to read in the PDF copy due to their low resolution. However, we have uploaded the figures in TIFF format, ensuring higher resolution and larger file sizes. We kindly request you to refer to the original figures for better clarity and readability.

Comment #15: Lines 314: What are the characteristics of different clusters? In Fig. 8, C1 to C5 were used as clusters? What C1 to C5 indicates in terms of area, yield, and production?

Response to comment #15: Thank you for your comments. In our study, we utilized long-term data of area, production, and yield as input parameters for identifying clusters using a multivariate clustering technique. The logic behind cluster analysis is to uncover hidden structures or patterns within a dataset and group similar observations together. More specif-ically it helps to identify groups or clusters within a dataset based on similarities or dissimi-larities between observations. In the case of the Aus season (Fig. 8), we identified five clus-ters out of the 64 districts. Each cluster represents a group of districts with similar charac-teristics in terms of cultivation area, production, and yield. Additionally, these clusters cap-ture regional heterogeneity and agro-ecological conditions, enabling effective policy formu-lation and implementation.

For example, the characteristics of Cluster 5 in Fig. 8 includes three districts (Bhola, My-mensingh, and Patuakhali) with similar dynamics in Aus rice cultivation. These districts are located in two coastal and one central region of Bangladesh. The rice cultivation area (high-est among the clusters) in this cluster ranges from 15,000 to 88,000 hectares, with more sta-ble rice production ranging from 40,000 to 170,000 tons and the highest mean rice produc-tion. The average yield is 1.8 tons per hectare, with a minimum of 1.2 tons per hectare and a maximum of 2.8 tons per hectare. Furthermore, six agro-ecological zones (AEZs) namely 8, 9, 13, 18, 28, and 29 have been identified in terms of Aus rice cultivation area, production, and yield within this cluster, indicating similar agro-ecological conditions in those regions for Aus cultivation.

Similarly, each cluster exhibits unique characteristics that provide valuable insights for pol-icymakers and researchers in formulating effective policies and taking immediate initia-tives to sustain rice production in Bangladesh.

Comment #16: Lines 319: Elbow criterion – not readable so difficult to understand.

Response to comment #16: Thank you for your comments. We have included a footnote to provide a clear explanation of the 'elbow criterion'. Please refer to the following text: “The elbow criterion is a method used in cluster analysis to determine the optimal number of clusters in a dataset. It involves plotting the variance explained by the clusters against the number of clusters. The plot typically resembles an arm, and the "elbow" or bend in the plot represents the point where the addition of more clusters does not significantly reduce the variance. This point is considered the optimal number of clusters for the given dataset. The elbow criterion helps in selecting a reasonable number of clusters that balance capturing meaningful patterns in the data while avoiding overfitting.”

Comment #17: Lines: 321- 329: Please see my comments above.

Response to comment #17: Thank you for your comments. We have addressed the im-portance of clustering in response to comment #14 and provided an explanation of the key characteristics of each cluster in response to comment #15. Please refer to the aforemen-tioned responses for more details.

Comment #18: Line 325: Poorest rice production – what does it mean? I think you wanted to mean lowest total production.

Response to comment #18: Thank you for bringing this to our attention. We have revised the sentence as per your recommendation.

Comment #19: Lines 337 – 365: Comments related to cluster analysis mentioned above applies here as well.

Response to comment #19: Thank you for your comments. We have addressed the im-portance of clustering in response to comment #14 and provided an explanation of the key characteristics of each cluster in response to comment #15. Please refer to the aforemen-tioned responses for more details.

Comment #20: Lines 366 – 370: Increasing trend of what? I understand this could be the area but there is no mention of it in the text or in the figure 11. This should be clear that of the total cultivated area, xx% were on HYV cultivation. This is also country level data. Up to this point, the data was presented at the district level. Here, there is no mention of wheth-er the data and the figure are at country level or district level.

Response to comment #20: Thank you for your comments. We have made the necessary re-visions to this section accordingly.

Comment #21: Fig 11: Adoption (%) - % of what? Should be mentioned. Same applies to Fig. 12.

Response to comment #21: Thank you for your comments. We have made the necessary re-visions to these figures accordingly.

Comment #22: Fig 12: Spatial distribution of adoption rate presented here is very confus-ing. Adoption rate for each district for 14 years were classified into different groups. But this data should have two dimensions on is temporal and the other is spatial. How the au-thors presented these in these maps? What is the point of presenting the previous years? I think the authors can take the adoption rate of the last year and then classify them in differ-ent group (64 district into different group) which then can be presented in a map.

Response to comment #22: Thank you for your comments. We sincerely apologize for any confusion caused by Fig 12 and appreciate the opportunity to clarify it. In Fig 12, we have utilized 14 years of high-yielding variety (HYV) adoption data from 64 districts. This data encompasses two dimensions: temporal and spatial. Our aim is to illustrate the spatial dis-tribution and temporal variations in the adoption rate of HYVs in rice cultivation across Bangladesh.

If we were solely interested in the spatial distribution, it would be sufficient to present the adoption rate of the current year. However, this approach would not capture the dynamic changes and fluctuations in HYV adoption over time. As varietal adoption is subject to dy-namic shifts, relying solely on the current year's data might overlook important temporal variations. Moreover, cultivating a particular variety in a single year does not guarantee its cultivation in the following year.

Given the two-dimensional nature of the data and our objective to track both spatiotemporal variations, a statistical technique is required. For the spatial distribution, we have employed the mean value of the 14-year data for each district, which is represented in the map as one of the legends "HYV adoption (%)." Additionally, we have included another legend in the form of "coefficient of variation (%)" to depict the temporal variability of HYV adoption in each district. The coefficient of variation (CV) was calculated using the 14-year data points for each district and visualized as circles on the map.

Therefore, to fulfill our research objective and ensure clarity, it is essential to consider the multi-year data. We have revised the title of Fig 12 accordingly. We hope that our explana-tion satisfactorily addresses your concerns.

Comment #23: Line 395 -396: Districtwide adaptation scenario – is it adaptation or adop-tion? In the whole section of 3.5, adaptation was used in the text whether in the title it is adoption.

Response to comment #23: Thank you for bringing this to our attention. We sincerely apol-ogize for the typo mistake. The correct term should be "adoption" instead of "adaptation" in section 3.5. We appreciate your feedback, and we have made the necessary revisions to en-sure consistency throughout the section.

Comment #24: Fig 13: Again it is hard to understand the message from the figure caption or the axis headings. What does influence mean here? Increase or decrease of yield or pro-duction?

Response to comment #24: Thank you for your comments. We deeply regret any confusion caused by Fig 13 and we are grateful for the chance to provide clarification. We have re-vised Section 3.5 and included the methodology in Section 2.4 to ensure a clear explanation and improve understanding. We kindly request you to review these sections, and we hope that the revisions will address your concerns adequately.

Comment #25: Lines 432-441: Mostly already described in the introduction (lines 78 to 86).

Response to comment #25: Thank you for your comments. We have taken note of your sug-gestion and have removed the redundant information as per your recommendation. We ap-preciate your feedback in streamlining the content of the manuscript.

Comment #26: Lines 448-452: Repetition of results

Response to comment #26: Thank you for your continued feedback. We have carefully con-sidered your comment and have made the appropriate changes to ensure that the content is concise and avoids repetition. We appreciate your diligence in reviewing our manuscript and helping us enhance its quality.

Comment #27: Line 452-453: “The bulk of the aromatic rice is grown during the Aman season which may lead to high Aman rice production in Dinajpur” what about area. Aro-matic rice nothing to do with the high production it is the area or the yield. There is no dis-cussion on the yield differences and on varieties. In some areas, local varieties or HYV with lower yield are grown which has impact on overall production.

Response to comment #27: Thank you for your comments. We have made the necessary re-visions to this section accordingly.

Comment #28: Lines 454- 462: Any references for this?

Response to comment #28: Thank you for pointing out the need for references in lines 454-462. We apologize for the oversight and have now included the relevant citations as per your suggestion.

Comment #29: Lines 491 – 496: Any data or references?

Response to comment #29: Thank you for pointing out the need for references in lines 491-496. We apologize for the oversight and have now included the relevant citations as per your suggestion. 

Comment #30: Lines 499-502: Vertical and hydroponic farming for rice? Any references?

Response to comment #30: Thank you for bringing this to our attention. We appreciate your comment and have removed the reference line to vertical and hydroponic farming for rice from the manuscript.

Comment #31: Lines 529-532: Repetitive.

Response to comment #31: Thank you for your comments. We have taken them into consid-eration and have removed the mentioned lines from the manuscript.

Thank you once again for your precious comments and advice. Those comments are all val-uable and very helpful for revising and improving our manuscript. We have revised the manuscript accordingly, and our point-by-point responses are presented above. We hope you are satisfied with our answers and the new data we have provided. Our deepest gratitude goes to you for your careful work and thoughtful suggestions that have helped improve this paper substantially.

Sincerely yours

All Authors’

Manuscript number: PONE-D-23-09867

Title: Spatiotemporal mapping of rice acreage and productivity growth in Bangladesh

Journal: PLOS ONE

Dear Reviewer,

Thank you for the comments concerning our manuscript. We deeply appreciate your posi-tive evaluation of our work. Those comments are valuable and very helpful. We have read through comments carefully and have made corrections. Please see below; all tasks and re-visions taken are shown point-by-point.

Response to Reviewer’s comments

Reviewer #2:

Comment #1: The research topic “Spatiotemporal mapping of rice acreage and productivity growth in Bangladesh” is substantive and is within the scope of the journal. In general, the paper is thoroughly researched and well-written. By the way, I have some minor comments for the authors that would help to improve the quality of the manuscript.

Response to comment #1: We would like to express our gratitude for your valuable and in-sightful suggestions and comments regarding the improvement of our manuscript. We firm-ly believe that they have significantly enhanced the scientific value of the manuscript, and we sincerely appreciate your contributions.

Abstract

Comment #2: Line # 35: Authors wrote “… performed a spatial-temporal mapping of the area, production, and yield….”. Is the word “area” sufficient or it is “cultivated/cultivation area”?

Response to comment #2: Thank you for the suggestion. The word “cultivation” has been added.

Comment #3: Line # 37 – 38: Replace “Results show that …” with “Results showed that….”.

Response to comment #3: The word 'show' has been replaced with 'showed'.

Comment #4: Line # 41: “…the rice area in 19 districts, 11 districts, and 13 districts de-clined significantly”. The word “rice area” is not a suitable wording. Please replace it with “rice-cultivation-area” or “cultivation area”.

Response to comment #4: The word "cultivation" has been added before the word "area".

Introduction

Comment #5: Line # 60-63: Please add the information “over 3.5 billion people are solely dependent upon rice for at least 20% of their daily required calories” from Introduction sec-tion of “Alam, M. J., Alamin, M., Sultana, M. H., Ahsan, M. A., Hossain, M. R., Islam, S. S., & Mollah, M. N. H. (2020). Bioinformatics studies on structures, functions and diversifica-tions of rolling leaf related genes in rice (Oryza sativa L.). Plant Genetic Resources, 18(5), 382-395” with the sentence “With over half of the world's population depending on rice for their daily energy, and it supplies approximately 62% carbohydrate, 46% protein, 8% fat, 7% calcium, and 44% phosphorus of the recommended dietary allowance” and cite Alam et al. 2020.

Response to comment #5: We appreciate your suggestion, and we have included the provid-ed information and incorporated the reference you suggested. Thank you for your valuable input.

Comment #6: Line # 65-66: Please add citation for the sentence “China is the leading rice-producing country, followed by India, Bangladesh, and Vietnam”.

Response to comment #6: We appreciate your observation. The citation for the sentence "China is the leading rice-producing country, followed by India, Bangladesh, and Vietnam" has been duly included.

Comment #7: Line #103-104: “This paper analyzed spatiotemporal” is not a standard word-ing. Please reshape the sentence as “In this study, we analyzed spatiotemporal data on culti-vation area, and production and yield of rice to examine/investigate the trends and growth patterns from 2006-2007 to 2019-2020 in Bangladesh. Do not mention the data sources un-der Introduction section, rather write it under Materials and Methods section.

Response to comment #7: Thank you for the suggestion. We have revised the sentence ac-cordingly.

Materials and Methods

Comment #8: Line # 120: Replace the word “area” with “cultivation area”. Also, this is ap-plicable for whole body of this manuscript.

Response to comment #8: We have made the necessary adjustment by replacing the term "area" with "cultivation area" throughout the entire manuscript, as suggested.

Comment #9: Please add a sentence to mention the level of significance considered for dif-ferent statistical tests (e.g., normality test, t-test, etc.) under “Statistical analysis” subsec-tion.

Response to comment #9: As per your recommendation, we have included the significance level of 5% in the analysis. Thank you for the suggestion.

Results

Comment #10: 1. Line # 233: “… seasons in Bangladesh is illustrated in Fig 4”. Replace “is” with “are”.

Response to comment #10: The word 'is' has been replaced with 'are'.

Comment #11: 2. Line # 293: Add unit of measurement for area, production and yield with-in brackets.

Response to comment #11: We have incorporated the percentage as a unit of measurement for the growth rate.

Comment #12: 3. Line # 425: Replace “will” with “would” in line # 425.

Response to comment #12: The word 'will' have been replaced with 'would'.

Discussion

Comment #13: Line # 484: Add an “and” before the phrase “declined over time”.

Response to comment #13: We have corrected it accordingly.

Comment #14: Line # 485: Replace “rice area” with “rice cultivation areas”.

Response to comment #14: We have corrected it accordingly.

Comment #15: Line # 555: Add “were” verb before the phrase “revealed in five ….” in line no. 555.

Response to comment #15: We have added ‘were’ verb before the phrase “revealed in five ….” accordingly.

Comment #16: Line # 561: Replace “will” with “would” in line no. 561.

Response to comment #16: The word 'will' have been replaced with 'would'.

Comment #17: I would suggest to add practical implications of this research under the Dis-cussion section.

Response to comment #17: Thank you for the suggestion. We have made an effort to incor-porate the practical implications of this research into the discussion section. Please see the text, “The research findings have significant practical implications for informing and guid-ing future agricultural practices and policies in rice-intensive areas of Bangladesh. The study provides valuable insights into the dynamic changes in the area and production of rice cultivation at a more disaggregated level, representing diverse ecosystems and man-agement constraints. Policymakers can utilize this information to develop targeted strate-gies and interventions aimed at enhancing rice production in specific regions. For instance, regions with favorable environmental conditions and fertile soil content can be encouraged to prioritize expanding rice cultivation. Efforts can be directed towards providing farmers in these regions with access to modern high-yielding varieties and advanced agricultural technologies to improve productivity. Conversely, areas facing challenges such as salinity or hilly terrain require tailored approaches to overcome these limitations, such as the de-velopment of stress-tolerant rice varieties and the implementation of appropriate irrigation systems. Additionally, identifying regions where rice production is declining, or farmers are shifting to alternative crops can guide initiatives aimed at revitalizing rice cultivation through improved irrigation facilities, training programs, and the development of short-duration varieties. The study highlights the areas that require closer attention to overcome challenges and enhance productivity, ultimately contributing to achieving Sustainable De-velopment Goal 2.3 of doubling agricultural productivity and income for smallholder farm-ers by 2030. In summary, these practical implications can contribute to sustainable agricul-tural practices, enhance food security, and improve the livelihoods of farmers in Bangla-desh.”

Comment #18: In the Discussion section limitation of this study and future scope could be added.

Response to comment #18: Thank you for the suggestion. We have made an effort to incor-porate the limitation and future research scope into the discussion section. Please see the text, “Despite the valuable insights provided by this study, there are certain limitations that should be acknowledged. First, this research focused primarily on the quantitative analysis of rice cultivation area and production trends, and did not delve into the underlying socio-economic factors that may influence these trends. Additionally, this study relied on second-ary data sources, which may be subject to limitations such as data accuracy and availabil-ity. Conducting primary data collection through field surveys and interviews could enhance the accuracy and reliability of the findings. Furthermore, the study's scope was limited to a specific timeframe (2006-2019), and it is important to recognize that future changes in cli-mate, technology, and agricultural practices could impact rice cultivation in unforeseen ways.

Building on the findings and limitations of this study, there are several areas for future re-search that can further contribute to the understanding of rice cultivation in Bangladesh. Firstly, investigating the socio-economic factors that influence farmers' decision-making processes regarding rice cultivation, including factors such as market conditions, govern-ment policies, and farmers' preferences, would provide valuable insights into the drivers of rice production. Secondly, exploring the impact of climate change on rice cultivation, in-cluding the effects of rising temperatures, changing rainfall patterns, and increased occur-rences of extreme weather events, is crucial for developing climate-resilient agricultural strategies. Additionally, examining the adoption and effectiveness of specific interventions aimed at improving rice productivity, such as the dissemination of high-yielding varieties, the implementation of irrigation systems, and the provision of training and extension ser-vices, would help in identifying best practices and areas for improvement. Finally, integrat-ing remote sensing and geospatial analysis techniques can enhance the accuracy and time-liness of monitoring rice cultivation dynamics, allowing for more precise and up-to-date assessments of cultivation area and production trends.”

Conclusion

Comment #19: Line 545-550: First four sentences of Conclusion section are repetition that are already told in objective and method section. Rewrite these into one introductory sen-tence of Conclusion and write the conclusion with more focus on main finding and recom-mendation and policy making for government. 

Response to comment #19: We greatly appreciate your valuable suggestion, which has prompted us to thoroughly revise the conclusion section and incorporate essential policy recommendations. 

Thank you once again for your precious comments and advice. Those comments are all val-uable and very helpful for revising and improving our manuscript. We have revised the manuscript accordingly, and our point-by-point responses are presented above. We hope you are satisfied with our answers and the new data we have provided. Our deepest gratitude goes to you for your careful work and thoughtful suggestions that have helped improve this paper substantially.

Sincerely yours

All Authors’

Manuscript number: PONE-D-23-09867R1

Title: Spatiotemporal mapping of rice acreage and productivity growth in Bangladesh

Journal: PLOS ONE

Dear Reviewer,

Thank you for the comments concerning our manuscript. We deeply appreciate your positive evaluation of our work. Those comments are valuable and very helpful. We have read through comments carefully and have made corrections. Please see below; all tasks and re-visions taken are shown point-by-point.

Response to Reviewer’s comments

Reviewer #3:

Comment #1: If the authors have adequately addressed your comments raised in a previous round of review and you feel that this manuscript is now acceptable for publication, you may indicate that here to bypass the “Comments to the Author” section, enter your conflict of interest statement in the “Confidential to Editor” section, and submit your "Accept" recommendation.

Reviewer response: All comments have been addressed.

Response to comment #1: Thank you for your comment. We appreciate your positive feedback.

Comment #2: Is the manuscript technically sound, and do the data support the conclusions?

Reviewer response: Yes

Response to comment #2: Thank you for your valuable insights.

Comment #3: Has the statistical analysis been performed appropriately and rigorously?

Reviewer response: Yes

Response to comment #3: Thank you for your positive feedback.

Comment #4: Have the authors made all data underlying the findings in their manuscript fully available?

Reviewer response: Yes

Response to comment #4: Thank you for your valuable feedback.

Comment #5: Is the manuscript presented in an intelligible fashion and written in standard English?

Reviewer response: Yes

Response to comment #5: Thank you for your comment.

Comment #6: Review Comments to the Author

Reviewer response: The authors have fairly addressed all the comments. In this regard, this article can be published in this journal. Good luck!

Response to comment #6: Thank you once more for your valuable comments and feedback.

Sincerely yours

All Authors’

Manuscript number: PONE-D-23-09867R1

Title: Spatiotemporal mapping of rice acreage and productivity growth in Bangladesh

Journal: PLOS ONE

Dear Reviewer,

Thank you for the comments concerning our manuscript. We deeply appreciate your positive evaluation of our work. Those comments are valuable and very helpful. We have read through comments carefully and have made corrections. Please see below; all tasks and re-visions taken are shown point-by-point.

Response to Reviewer’s comments

Reviewer #4:

Comment #1: If the authors have adequately addressed your comments raised in a previous round of review and you feel that this manuscript is now acceptable for publication, you may indicate that here to bypass the “Comments to the Author” section, enter your conflict of interest statement in the “Confidential to Editor” section, and submit your "Accept" recommendation.

Reviewer response: All comments have been addressed

Response to comment #1: Thank you for your comment. We appreciate your positive feedback.

Comment #2: Is the manuscript technically sound, and do the data support the conclusions?

Reviewer response: No

Response to comment #2: Thank you for your comment. Indeed, our primary emphasis was on delving into the regional context, striving to extract meaningful insights within the framework of 64 specific locations. It's crucial to acknowledge that the rice-growing ecosystem in Bangladesh exhibits considerable diversity, influenced by factors such as geographical position, socio-economic conditions, and environmental variations. Consequently, the previous study's findings have limitations in terms of statistical robustness and generalizability for shaping region-specific policies. In response to these limitations, our current study adopts a more granular approach by analyzing data from 64 locations. This approach aims to offer a comprehensive understanding of the dynamics of rice cultivation in Bangladesh. We believe that this shift not only enhances the statistical validity of our research but also facilitates the formulation of more effective policies tailored to specific locations. For the technical soundness and scientific evidence, please refer to results and discussion sections for details.

In addressing the data support aspect, we employed information related to rice cultivation area, production, and yield over a 14-year period across 64 districts, covering the Aus, Aman, and Boro seasons from 2006-2007 to 2019-2020. The data were sourced from the Bangladesh Bureau of Statistics (BBS), a government department with national authorization, which consistently collects rice area, production, and yield data for all 64 districts annually since 2006. Consequently, our study's sample initiation was from 2006, extending up to the latest available data, ensuring the utilization of a nationally representative data source in Bangladesh. Regarding sample size, our time series dataset comprises 2688 observations or data points for each variable, establishing a comprehensive and nationally representative sample.

In the context of replicating our study, our methodology can be applied to datasets in other rice-growing countries and diverse crops. Despite revealing potential threats to agricultural sustainability, such as temporal variations and regional disparities in rice production, these aspects have received limited exploration. Consequently, other nations cultivating rice can replicate our methodology to investigate the dynamics of regional rice cultivation in their respective contexts.

Our study's conclusions, succinctly presented in the conclusion and abstract sections, highlight significant findings from the analyzed data. Notably, we like to refers the following text for your reference, “To ensure the sustainability of rice production in Bangladesh, it is imperative to have a comprehensive understanding of the spatiotemporal distribution of rice cultivation area, production, and yield. This knowledge will enable the formulation of region-specific policies tailored to the specific needs of different areas. The study has introduced novel methodological approaches for trend analysis and spatial clustering. Our findings showed that 14 years averages of rice cultivation area, production, and yield for three major seasons, Aus, Aman, and Boro, differ significantly among the study districts in Bangladesh. The Aus season has the highest temporal variability of rice production determinants, followed by the Aman and Boro seasons. Regional disparities in production were revealed in five cluster groups for the Aus season, seven for the Aman, and six for the Boro season. The share of HYV adoption significantly increased for most of the season. A significant increasing trend in Aus (0.007-0.521%), Aman (0.004-0.039%), and Boro (0.013-0.584%) were observed in 28, 34, and 36 districts, respectively, with an increase of 1% adoption of HYV. Predictions revealed that more than 5% of rice production would be increased in 28 districts in the Aus season, and for Aman and Boro, more than 5% would be increased in Narail and Bogura, respectively. Moreover, a 1-5% increase will be found in 50, 54, and 41 districts in Aus, Aman, and Boro seasons, respectively. These findings underscore the importance of formulating tailored and targeted policies at the regional level to effectively enhance rice productivity in Bangladesh”.

We sincerely appreciate your insightful comments, and in light of your input, we have taken steps to incorporate the novelty of our research into the introduction section of our manuscript. We hope you are satisfied with our response, and if necessary, we are happy to provide further clarification.

Comment #3: Has the statistical analysis been performed appropriately and rigorously?

Reviewer response: No

Response to comment #3: We applied a robust and rigorous statistical method for data analysis. Augmented Dickey-Fuller test and Shapiro-Wilk normality test were applied to check the stationarity of time series and normality test of the respective data series. In growth analysis, we employed exponential growth model to estimate growth and trend nature. To achieve more comprehensive and precise results, we utilized time series data of rice cultivation area, production, and yield at the district level as input parameters for clustering. We employed robust multivariate clustering techniques, specifically dynamic time warping (DTW). This approach had overcome the limitations in identifying regional heterogeneity as it focused on mean values and had a limited scope. Additionally, we employed principal component analysis and optimal clustering methods to identify the most suitable clusters in our study. The cluster analysis resulted in grouping similar districts, facilitating the creation of a rice zoning map and providing valuable insights for policy implications. Moreover, we employed Ordinary Least Squares (OLS) regression to investigate the adoption rate and rice production in Bangladesh, aiming to evaluate the influence of adopting high-yielding varieties on rice production. Please see the details in the “Materials and Methods” section.

Comment #4: Have the authors made all data underlying the findings in their manuscript fully available?

Reviewer response: No

Response to comment #4: We have uploaded the minimal anonymized dataset and subsequently adjusted the data availability statement according to the journal data availability guideline. Please see the PLOS ONE data availability guideline.

Comment #5: Is the manuscript presented in an intelligible fashion and written in standard English?

Reviewer response: No

Response to comment #5: We acknowledge the reviewer's valuable feedback regarding the need for a thorough check for grammatical and typographical errors throughout the manuscript. We have carefully addressed this concern and made the necessary revisions to ensure the quality and accuracy of the text. Thank you for your thoughtful input.

Comment #6: Review Comments to the Author

Reviewer response: Sorry to say that there is nothing new in the manuscript. A lot of similar studies have been performed. These cited studies are missing in this paper. I suggest rejecting this paper due to lack of novelty.

Response to comment #6: Thank you for your comment. While we appreciate your perspective, we believe there are unique contributions in our manuscript that may not have been fully captured in the previous studies. We acknowledge the importance of novelty in research and have carefully revised our manuscript to emphasize the distinctive aspects of our work in the introduction section. We have also incorporated relevant citations to address the updated references. We hope that upon reevaluation, you may find the revised manuscript to be a valuable addition to the existing literature. For detailed insights into the novelty of our research, kindly refer to response of Comment #1 from Reviewer #1 and Introduction section of our manuscript. Your insights have been instrumental in shaping our improvements, and we appreciate your time and consideration.

Sincerely yours

All Authors’

---

## [Decision Letter · Decision Letter 2]

1 Mar 2024

Spatiotemporal mapping of rice acreage and productivity growth in Bangladesh

PONE-D-23-09867R2

Dear Dr. Md. Abdullah Al Mamun,

We’re pleased to inform you that your manuscript has been judged scientifically suitable for publication and will be formally accepted for publication once it meets all outstanding technical requirements.

Kind regards,

Vishal Ahuja

Academic Editor

PLOS ONE

Reviewers' comments:

**Comments to the Author**

Reviewer #5: All comments have been addressed

---

## [Editor Report · Acceptance letter]

6 Mar 2024

PONE-D-23-09867R2 

PLOS ONE

Dear Dr. Al Mamun, 

I'm pleased to inform you that your manuscript has been deemed suitable for publication in PLOS ONE. Congratulations! Your manuscript is now being handed over to our production team.

Kind regards, 

on behalf of

Dr. Vishal Ahuja 

Academic Editor

PLOS ONE